# Sample, estimate, aggregate:
# A recipe for causal discovery foundation models

**Menghua Wu**                                        *rmwu{at}mit.edu*
*Department of Computer Science, Massachusetts Institute of Technology*

**Yujia Bao**                                        *yujia.bao{at}accenture.com*
*Center for Advanced AI, Accenture*

**Regina Barzilay**                                        *regina{at}csail.mit.edu*
*Department of Computer Science, Massachusetts Institute of Technology*

**Tommi S. Jaakkola**                                        *tommi{at}csail.mit.edu*
*Department of Computer Science, Massachusetts Institute of Technology*

**Reviewed on OpenReview:** *https://openreview.net/forum?id=h434zx5SX0*

## Abstract

Causal discovery, the task of inferring causal structure from data, has the potential to uncover mechanistic insights from biological experiments, especially those involving perturbations. However, causal discovery algorithms over larger sets of variables tend to be brittle against misspecification or when data are limited. For example, single-cell transcriptomics measures thousands of genes, but the nature of their relationships is not known, and there may be as few as tens of cells per intervention setting. To mitigate these challenges, we propose a foundation model-inspired approach: a supervised model trained on large-scale, synthetic data to predict causal graphs from summary statistics — like the outputs of classical causal discovery algorithms run over subsets of variables and other statistical hints like inverse covariance. Our approach is enabled by the observation that typical errors in the outputs of a discovery algorithm remain comparable across datasets. Theoretically, we show that the model architecture is well-specified, in the sense that it can recover a causal graph consistent with graphs over subsets. Empirically, we train the model to be robust to misspecification and distribution shift using diverse datasets. Experiments on biological and synthetic data confirm that this model generalizes well beyond its training set, runs on graphs with hundreds of variables in seconds, and can be easily adapted to different underlying data assumptions.[1]

## 1 Introduction

A fundamental aspect of scientific research is to discover and validate causal hypotheses involving variables of interest. Given observations of these variables, the goal of causal discovery algorithms is to extract such hypotheses in the form of directed graphs, in which edges denote causal relationships (Spirtes et al., 2001). There are several challenges to their widespread adoption in basic science. The core issue is that the correctness of these algorithms is tied to their assumptions on the data-generating processes, which are unknown in real applications. In principle, one could circumvent this issue by exhaustively running discovery algorithms with different assumptions and comparing their outputs with surrogate measures that reflect graph quality (Faller et al., 2023). However, this search would be costly: current algorithms must be optimized from scratch each time, and they scale poorly to the graph and dataset sizes present in modern scientific big data (Replogle et al., 2022).

---

[1]Our code is available at `https://github.com/rmwu/sea`.

Causal discovery algorithms follow two primary approaches that differ in their treatment of the causal graph. Discrete search algorithms explore the super-exponential space of graphs by proposing and evaluating changes to a working graph (Glymour et al., 2019). While these methods are fast on small graphs, the combinatorial space renders them intractable for exploring larger structures. An alternative is to frame causal discovery as a continuous optimization over weighted adjacency matrices. These algorithms either fit a generative model to the data and extract the causal graph as a parameter (Zheng et al., 2018), or train a supervised learning model on simulated data (Petersen et al., 2023). However, these methods may be less robust beyond simple settings, and their optimization can be nontrivial (Ng et al., 2024).

In this work, we present SEA: Sample, Estimate, Aggregate, a supervised causal discovery framework that aims to perform well even when data-generating processes are unknown, and to easily incorporate prior knowledge when it is available. We train a deep learning model to predict causal graphs from two types of statistical descriptors: the estimates of classical discovery algorithms over small subsets, and graph-level statistics. Each classical discovery algorithm outputs a representation of a graph's equivalency class, and the types of errors that it makes may be comparable across datasets. On the other hand, statistics like correlation or inverse covariance are fast to compute, and strong indicators for a graph's overall connectivity. Theoretically, we illustrate a simple algorithm for recovering larger causal graphs that are consistent with estimates over subsets, and we show that there exists a set of model parameters that can map sets of subgraph estimates to the correct global graph. Empirically, our training procedure forces the model to predict causal graphs across diverse synthetic data, including on datasets that are misaligned with the discovery algorithms' assumptions, or when insufficient subsets are provided.

Our experiments probe three qualities that we view a foundation model should fulfill, with thorough comparison to three classical baselines and five deep learning approaches. Specifically, we assess the framework's ability to generalize to unseen and out-of-distribution data; to steer predictions based on prior knowledge; and to perform well in low-data regimes. SEA attains the state-of-the-art results on synthetic and real causal discovery tasks, while providing 10-1000x faster inference. To incorporate domain knowledge, we show that it is possible to swap classic discovery algorithms at inference time, for significant improvements on datasets that match the assumptions of the new algorithm. Our models can also be finetuned at a fraction of the training cost to accommodate new graph-level statistics that capture different (e.g. nonlinear) relationships. We extensively analyze SEA in terms of low-data performance, scalability, causal identifiability, and other design choices. To conclude, while our experiments focus on specific algorithms and architectures, this work presents a blueprint for designing causal discovery foundation models, in which sampling heuristics, classical causal discovery algorithms, and summary statistics are the fundamental building blocks.

## 2 Background and related work

### 2.1 Causal structure learning

A **causal graphical model** is a directed graph $G = (V, E)$, where each node $i \in V$ corresponds to a random variable $X_i \in X$ and each edge $(i, j) \in E$ represents a causal relationship from $X_i \rightarrow X_j$. There are a number of assumptions that relate data distribution $P_X$ to $G$ (Spirtes et al., 2001; Hauser & Bühlmann, 2012), which determine whether $G$ is uniquely identifiable, or identifiable up to an equivalence class. In this work, we assume causal sufficiency – that is, $V$ contains all the parents $\pi_i$ of every node $i$. Causal graphical models can be used, along with other information, to compute the downstream consequences of interventions. An intervention on node $i$ refers to setting conditional $P(X_i \mid X_{\pi_i})$ to a different distribution $\tilde{P}(X_i \mid X_{\pi_i})$. Our experiments cover the observational case (no interventions) and the case with perfect interventions on each node, i.e. for all nodes $i$, we have access to data where we set $P(X_i \mid X_{\pi_i}) \leftarrow \tilde{P}(X_i)$.

Given a dataset $D \sim P_X$, the goal of **causal structure learning** (causal discovery) is to recover $G$. There are two main challenges. First, the number of possible graphs is super-exponential in the number of nodes $N$, so algorithms must navigate this combinatorial search space efficiently. Second, depending on data availability and the underlying data generation process, causal discovery algorithms may or may not be able to recover $G$ in practice. In fact, many algorithms are only analyzed in the infinite-data regime and require at least thousands of data samples for reasonable empirical performance (Spirtes et al., 2001; Brouillard et al., 2020).

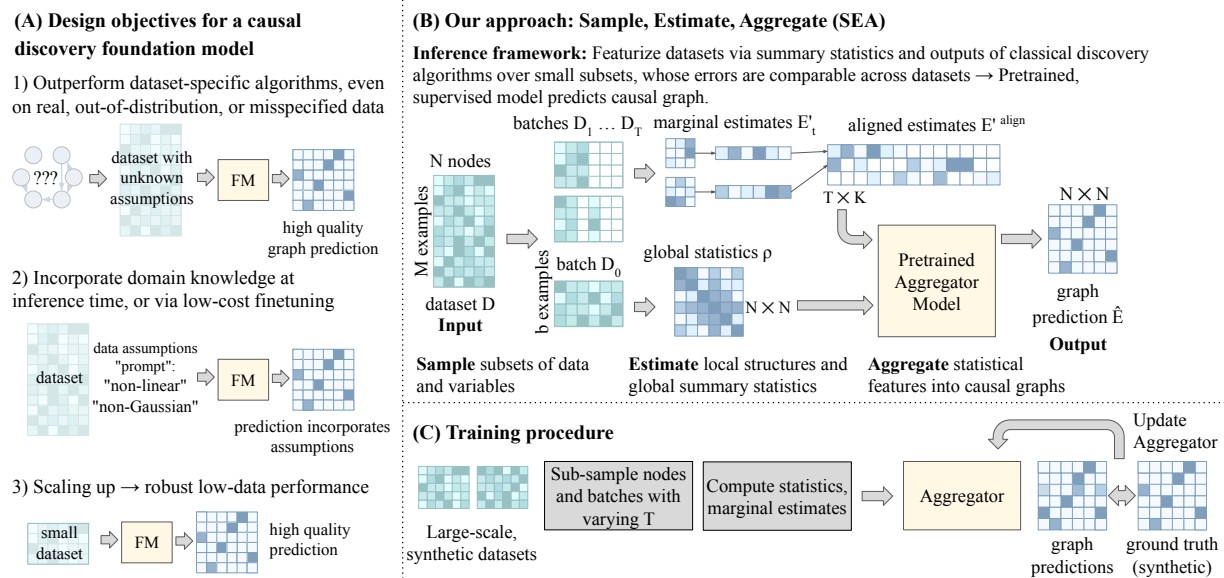

Figure 1: Overview of our goals and approach. (A) Criteria we aim to fulfill. (B-C) Inference and training procedure. Green: raw data. Blue: graph / features. Yellow: Learned. Gray: Stochastic, but not learned.

**Discrete search algorithms** encompass diverse strategies for traversing the combinatorial space of possible graphs. Constraint-based algorithms are based on conditional independence tests, whose discrete results inform of the presence or absence of edges, and whose statistical power depends directly on dataset size (Glymour et al., 2019). These include the observational FCI and PC algorithms (Spirtes et al., 1995), and the interventional JCI algorithm (Mooij et al., 2020). Score-based methods define a continuous score that guides the search through the discrete space of valid graphs, where the true graph lies at the optimum. Examples include GES (Chickering, 2002), GIES (Hauser & Bühlmann, 2012), CAM (Bühlmann et al., 2014), and IGSP (Wang et al., 2017). Finally, semi-parametric methods such as LiNGAM (Shimizu et al., 2006) or additive noise models (Hoyer et al., 2008) exploit asymmetries implied by the model class to identify graph connectivity and causal ordering.

**Continuous optimization approaches** recast the combinatorial space of graphs into a continuous space of weighted adjacency matrices. Many of these works train a generative model to learn the empirical data distribution, which is parameterized through the adjacency matrix (Zheng et al., 2018; Lachapelle et al., 2020; Brouillard et al., 2020). Others focus on properties related to the empirical data distribution, such as a relationship between the underlying graph and the Jacobian of the learned model (Reizinger et al., 2023), or between the Hessian of the data log-likelihood and the topological order (Sanchez et al., 2023). Finally, amortized inference approaches (Ke et al., 2023; Lorch et al., 2022; Petersen et al., 2023) frame causal discovery as a supervised learning problem, where a neural network is trained to predict (synthetic) graphs from (synthetic) datasets. To incorporate new information, current supervised methods must simulate new datasets and re-train. In addition, these models that operate on raw observations scale poorly to larger datasets. Since this direction is most similar to our own, we include further comparisons in B.2.

## 2.2 Foundation models

The concept of foundation models has revolutionized the machine learning workflow in a variety of disciplines: instead of training domain-specific models from scratch, we start from a pretrained, general-purpose model (Bommasani et al., 2021). This work describes a blueprint for designing "foundation models" in the context of causal discovery. The precise definition of a foundation model varies by application, but we aim to fulfill the following properties (Figure 1A), enjoyed by modern text and image foundation models (Radford et al., 2021; Brown et al., 2020).

1. A foundation model should enable us to outperform domain-specific models trained from scratch, even if the former has never seen similar tasks during training (Radford et al., 2019). In the context

of causal discovery, we would like to train a model that outperforms any individual algorithm on real, misspecified, and/or out-of-distribution datasets.

2. It should be possible to explicitly steer a foundation model's behavior towards better performance on new tasks, either directly at inference time, e.g. "prompting" (Reynolds & McDonell, 2021), or at low cost compared to pretraining (Ouyang et al., 2022). Here, we would like to easily change our causal discovery algorithm's "assumptions" regarding the data, e.g. by incorporating the knowledge of non-linearity, non-Gaussianity.

3. Scaling up a foundation model should lead to improved performance in few-shot or data-poor regimes (Brown et al., 2020). This aspect we analyze empirically.

In the following sections, we will revisit these desiderata from both the design and experimental perspectives.

## 3 Methods

Sample, Estimate, Aggregate (SEA) is a supervised causal discovery framework built upon the intuition that summary statistics and marginal estimates (outputs of a classical causal discovery algorithm on subsets of nodes) provide useful hints towards global causal structure. In the following sections, we describe how these statistics are efficiently estimated from data (Section 3.1), and how we train a neural network to predict causal graphs from these inputs (Section 3.2). We expand upon the model architecture in Section 3.3, and conclude with theoretical motivation for this architecture in Section 3.4.

### 3.1 Inference procedure

Given a new dataset $D \in \mathbb{R}^{M \times N}$, we sample small batches of nodes and observations; estimate global summary statistics and local subgraphs; and aggregate these information with a trained neural network (Figure 1B).

**Sample:** takes as input dataset $D$; and outputs data batches $\{D_0, D_1, \ldots, D_T\}$ and node subsets $\{S_1, \ldots, S_T\}$.

1. Sample $T + 1$ batches of $b \ll M$ observations uniformly at random from $D$.
2. Initialize selection scores $\alpha \in \mathbb{R}^{N \times N}$ (e.g. correlation or inverse covariance, computed over $D_0$ or $D$).
3. Sample $T$ node subsets of size $k$. Each subset $S_t$ is constructed iteratively as follows.
   (a) The initial node is sampled with probability proportional to $\sum_{j \in V} \alpha_{i,j}$.
   (b) Each subsequent node is added with probability proportional to $\sum_{j \in S_t} \alpha_{i,j}$ (prioritizing connections to nodes that have already been sampled), until $\|S_t\| = k$.
   (c) We update $\alpha$, down-weighting $\alpha_{i,j}$ proportional to the number of times $i, j$ have been selected.

We include further details and analyze alternative strategies for sampling nodes in B.6.

**Estimate:** takes as inputs data batches, node subsets, and (optionally) intervention targets; and outputs global statistics $\rho$ and marginal estimates $\{E'_1, \ldots, E'_T\}$.

1. Compute global statistics $\rho \in \mathbb{R}^{N \times N}$ over $D_0$ (e.g. correlation or inverse covariance).
2. Run discovery algorithm $f$ to obtain marginal estimates $f(D_t[S_t]) = E'_t$ for $t = 1 \ldots T$.

We use $D_t[S_t]$ to denote the observations in $D_t$ that correspond only to the variables in $S_t$. Each estimate $E'_t$ is a $k \times k$ adjacency matrix, corresponding to the $k$ nodes in $S_t$.

**Aggregate:** takes as inputs global statistics, marginal estimates, and node subsets. A trained aggregator model outputs the predicted global causal graph $\hat{E} \in (0, 1)^{N \times N}$ (Section 3.3).

### 3.2 Training procedure

The training procedure mirrors the inference procedure (Figure 1C). We assume access to pairs of simulated datasets and graphs, $(D, G)$, where each dataset $D$ is generated by a parametric model, whose dependencies are given by graph $G$ (Section 4.1). Datasets $D$ are summarized into global statistics and marginal estimates via the sampling and estimation steps. The resultant features are input to the aggregator (neural network), which is supervised by graphs $G$.

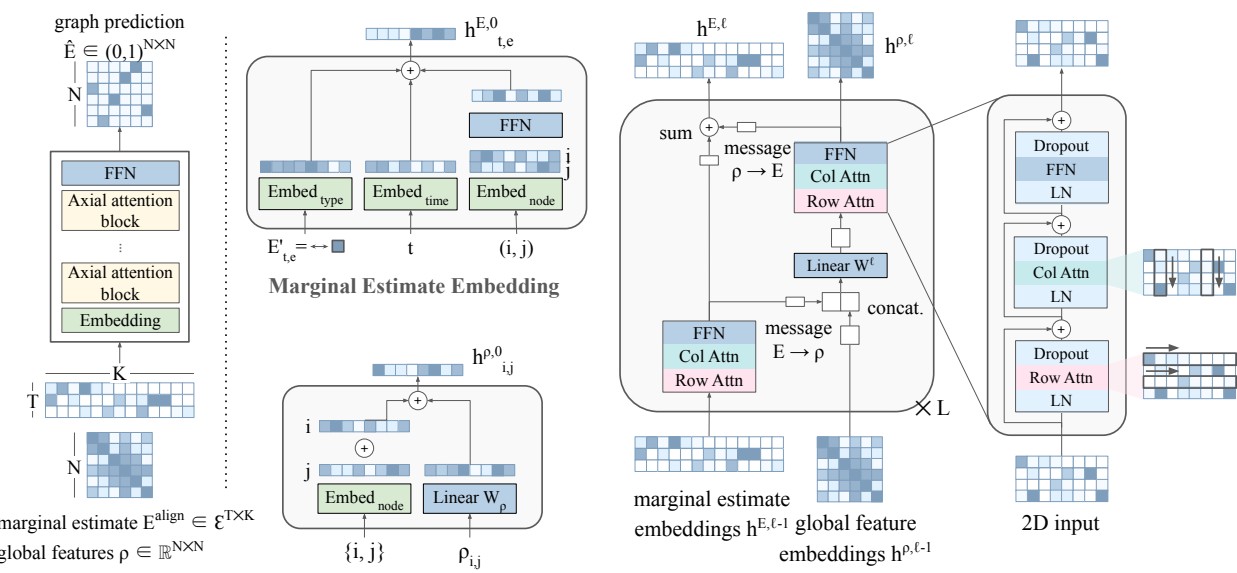

Figure 2: Aggregator architecture. Marginal graph estimates and global statistics are embedded into the model dimension. 1D positional embeddings are added along both rows and columns. Embedded features pass through a series of axial attention blocks, which attend to the marginal and global features. Final layer global features pass through a feedforward network to predict the causal graph.

We trained two aggregator models, which employed the FCI algorithm with the Fisherz test and GIES algorithm with the Bayesian information criterion (Schwarz, 1978). Both estimation algorithms were chosen for speed, but they differ in their assumptions, discovery strategies, and output formats. Though we only trained two models, alternate estimation algorithms that produce the same output type[2] may be used at inference time (experiments in Section 5.2). The training dataset contains both data that are correctly and incorrectly specified (Section 4.1), so the aggregator is forced to predict the correct graph regardless. In addition, each training instance samples a random number of marginal estimates, which might not cover every edge. As a result, the aggregator must extrapolate to unseen edges using the available estimates and the global statistics. For example, if two variables have low correlation, and they are located in different neighborhoods of the already-identified graph, it may be reasonable to assume that they are unconnected.

## 3.3 Model architecture

The aggregator is a neural network that takes as input: global statistics $\rho \in \mathbb{R}^{N \times N}$, marginal estimates $E'_{1 \ldots T} \in \mathcal{E}^{T \times k \times k}$, and node subsets $S_{1 \ldots T} \in [N]^{T \times k}$ (Figure 2), where $\mathcal{E}$ is the set of output edge types for the causal discovery algorithm $f$.[3]

We project global statistics into the model dimension via a learned linear projection matrix $W_\rho : \mathbb{R} \to \mathbb{R}^d$, and we embed edge types via a learned embedding $\mathrm{ebd}_\mathcal{E} : \mathcal{E} \to \mathbb{R}^d$. To collect estimates of the same edge over all subsets, we align entries of $E'_{1 \ldots T}$ into $E_T^{'\mathrm{align}} \in \mathcal{E}^{T \times K}$

$$E_{t,e=(i,j)}^{'\mathrm{align}} = \begin{cases} E'_{t,i,j} & \text{if } i \in S_t, j \in S_t \\ 0 & \text{otherwise} \end{cases} \tag{1}$$

where $t$ indexes into the subsets, $e = 1 \ldots K$ indexes into the set of unique edges, i.e. the union of pairs $(i, j)$ over all $E'_t$. We add learned 1D positional embeddings along both dimensions of each input,

$$\text{pos-ebd}(\rho_{i,j}) = \mathrm{ebd}_{\mathrm{node}}(i') + \mathrm{ebd}_{\mathrm{node}}(j')$$

$$\text{pos-ebd}(E_{t,e}^{'\mathrm{align}}) = \mathrm{ebd}_{\mathrm{time}}(t) + \mathrm{FFN}([\mathrm{ebd}_{\mathrm{node}}(i'), \mathrm{ebd}_{\mathrm{node}}(j')])$$

---

[2]PC (Spirtes et al., 2001) also predicts CPDAGs (Andersson et al., 1997), so its outputs may be input to the GIES model, while FCI's PAG outputs cannot.

[3]E.g. "no relationship," "$X$ causes $Y$," "$X$ is not a descendent of $Y$"

where $i', j'$ index into a random permutation on $V$ for invariance to node permutation and graph size.[4] Due to the (a)symmetries of their inputs, pos-ebd$(\rho_{i,j})$ is symmetric, while pos-ebd$(E_{t,e}^{'\text{align}})$ considers the node ordering. In summary, the inputs to our axial attention blocks are

$$h_{i,j}^{\rho} = (W_\rho \rho)_{i,j} + \text{pos-ebd}(\rho_{i,j}) \tag{2}$$

$$h_{t,e}^{E} = \text{ebd}_{\mathcal{E}}(E_{t,e}^{'\text{align}}) + \text{pos-ebd}(E_{t,e}^{'\text{align}}) \tag{3}$$

for $i, j \in [N]^2$, $t \in [T]$, $e \in [K]$. Note that attention is permutation invariant, so positional embeddings are *required* for the model to know which edges belong to the same subset, or what each edge's endpoints endpoints are.

**Axial attention**   An axial attention block contains two axial attention layers (marginal estimates, global statistics) and a feed-forward network (Figure 2, right). Given a 2D input, an axial attention layer attends first along the rows, then along the columns. For example, on a matrix of size $(\texttt{R},\texttt{C},\texttt{d})$, one pass of the axial attention layer is equivalent to running standard self-attention along $\texttt{C}$ with batch size $\texttt{R}$, followed by the reverse. For marginal estimates, $\texttt{R}$ is the number of subsets $T$, and $\texttt{C}$ is the number of unique edges $K$. For global statistics, $\texttt{R}$ and $\texttt{C}$ are both the total number of vertices $N$.

Following Rao et al. (2021), each self-attention mechanism is preceded by layer normalization and followed by dropout, with residual connections to the input,

$$x = x + \text{Dropout}(\text{Attn}(\text{LayerNorm}(x))). \tag{4}$$

We pass messages between the marginal and global layers to propagate information. Let $\phi_{E,\ell}$ be marginal layer $\ell$, let $\phi_{\rho,\ell}$ be global layer $\ell$, and let $h^{\cdot,\ell}$ denote the hidden representations out of layer $\ell$. The marginal to global message $m^{E\to\rho} \in \mathbb{R}^{N\times N\times d}$ contains representations of each edge averaged over subsets,

$$m_{i,j}^{E\to\rho,\ell} = \begin{cases} \frac{1}{T_e} \sum_t h_{t,e=(i,j)}^{E,\ell} & \text{if } \exists S_t, i, j \in S_t \\ \epsilon & \text{otherwise.} \end{cases} \tag{5}$$

where $T_e$ is the number of $S_t$ containing $e$, and missing entries are padded to learned constant $\epsilon$. The global to marginal message $m^{\rho\to E} \in \mathbb{R}^{K\times d}$ is simply the hidden representation itself,

$$m_{t,e=(i,j)}^{\rho\to E,\ell} = h_{i,j}^{\rho,\ell}. \tag{6}$$

We update representations based on these messages as follows.

$$h^{E,\ell} = \phi_{E,\ell}(h^{E,\ell-1}) \qquad\qquad \text{(marginal feature)} \tag{7}$$

$$h^{\rho,\ell-1} \leftarrow W^\ell \left[ h^{\rho,\ell-1}, m^{E\to\rho,\ell} \right] \qquad\qquad \text{(marginal to global)} \tag{8}$$

$$h^{\rho,\ell} = \phi_{\rho,\ell}(h^{\rho,\ell-1}) \qquad\qquad \text{(global feature)} \tag{9}$$

$$h^{E,\ell} \leftarrow h^{E,\ell} + m^{\rho\to E,\ell} \qquad\qquad \text{(global to marginal)} \tag{10}$$

$W^\ell \in \mathbb{R}^{2d\times d}$ is a learned linear projection, and $[\cdot]$ denotes concatenation.

**Graph prediction**   For each pair of vertices $i \neq j \in V$, we predict $e = 0, 1$, or $2$ for no edge, $i \to j$, and $j \to i$. We do not additionally enforce that our predicted graphs are acyclic, similar in spirit to Lippe et al. (2022). Given the output of the final axial attention block $h^\rho$, we compute logits

$$z_{\{i,j\}} = \text{FFN}\left( \left[ h_{i,j}^\rho, h_{j,i}^\rho \right] \right) \in \mathbb{R}^3 \tag{11}$$

which correspond to probabilities after softmax normalization. The overall output $\hat{E} \in \{0,1\}^{N\times N}$ is supervised by the ground truth $E$. Our model is trained with cross entropy loss and L2 regularization.

---

[4]The random permutation $i' = \sigma(V)_i$ allows us to avoid updating positional embeddings of lower order positions more than higher order ones, due to the mixing of graph sizes during training.

**Implementation details**   Unless otherwise noted, inverse covariance is used for the global statistic and selection score, due to its relationship to partial correlation. We sample batches of size $b = 500$ over $k = 5$ nodes each (analysis in 5.4). Our model was implemented with 4 layers with 8 attention heads and hidden dimension 64. Our model was trained using the AdamW optimizer with a learning rate of 1e-4 (Loshchilov et al., 2017). See B.4 for additional details about hyperparameters.

**Complexity**   The aggregator should be be invariant to node labeling, while maintaining the order of sampled subsets, so attention-based architectures were a natural choice (Vaswani et al., 2017). If we concatenated $\rho$ and $E'_{1\ldots T}$ into a length $N^2 T$ input, quadratic-scaling attention would cost $O(N^4 T^2)$. Instead, we opted for axial attention blocks, which attend along each of the three axes separately in $O(N^3 T + N^2 T^2)$. Both are parallelizable on GPU, but the latter is more efficient, especially on larger $N$.

### 3.4   Theoretical interpretation

**Marginal graph resolution**   It is well-established that estimates of causal graphs over subsets of variables can be "merged" into consistent graphs over their union (Faller et al., 2023; Tillman et al., 2008; Huang et al., 2020). In A.1 and A.2, we describe a simple algorithm towards this task, based on the intuition that edges absent from the global graph should be absent from at least one marginal estimate, and that v-structures present in the global graph are present in the marginal estimates. We then prove Theorem 3.1, which states that the axial attention architecture is well-specified as a model class. That is, there exists a setting of its weights that can map marginal estimates into global graphs. Our construction in A.3 follows the same reasoning steps as the simple algorithm in A.2, and it provides realistic bounds on the model size.

**Theorem 3.1.** *Let $G = (V, E)$ be a directed acyclic graph with maximum degree $d$. For $S \subseteq V$, let $E'_S$ denote the marginal estimate over $S$. Let $\mathcal{S}_d$ denote the superset that contains all subsets $S \subseteq V$ of size at most $d$. Given $\{E'_S\}_{S \in \mathcal{S}_{d+2}}$, a stack of $L$ axial attention blocks has the capacity to recover $G$'s skeleton and v-structures in $O(N)$ width, and propagate orientations on paths of $O(L)$ length.*

There are two practical considerations that motivate a framework like SEA, instead of directly running classic reconciliation algorithms. First, many of these algorithms rely on specific characterizations of the data-generating process, e.g. linear non-Gaussian (Huang et al., 2020). While our proof does not constrain the causal mechanisms or exogenous noise, it assumes that the marginal estimates are correct. These assumptions may not hold on real data. However, the failure modes of any particular causal discovery algorithm may be similar across datasets and can be corrected using statistics that capture richer information. For example, an algorithm that assumes linearity will make (predictably) asymmetric mistakes on non-linear data and underestimate the presence of edges. However, we may be able to recover nonlinear relationships with statistics like distance correlation (Sz'ekely et al., 2007). By training a deep learning model to reconcile marginal estimates and interpret global statistics, we are less sensitive to artifacts of sampling and discretization (e.g. p-value thresholds, statistics $\lesssim 0$). The second consideration is that checking a combinatorial number of subsets is wasteful on smaller graphs and infeasible on larger graphs. In fact, if we only leverage marginal estimates, we must check at least $O(N^2)$ subsets to cover each edge at least once. To this end, the classical Independence Graph algorithm (Spirtes et al., 2001) motivates statistics such as inverse covariance to initialize the undirected skeleton and reduce the number of independence tests required. This allows us to use marginal estimates more efficiently, towards answering orientation questions. We verify this latter consideration in Section 5.4, where we empirically quantify the number of estimates a global statistic is "worth."

**On identifiability**   The primary goal of this paper is to develop a practical framework for causal discovery, especially when data assumptions are unknown. Instead of focusing on the identifiability of any particular setting, we provide these interpretations of our model's outputs, and show empirically that our model respects classic identifiability theory (Section 5.3). The model will always output an orientation for all edges, but the graph can be interpreted as one member of an equivalence class. Metrics can be computed with respect to either the ground truth graph (if identifiable) or the inferred equivalence class, e.g. the implied CPDAG. When data do not match the estimation algorithm's assumptions, performance is inherently an empirical question, and we show empirically that our model still does well (Section 5.1).

# 4 Experimental setup

Our experiments aim to address the three desiderata proposed in Section 2.2 – namely, generalization, adaptability, and emergent few-shot behavior. These experiments span both real and synthetic data. Real experiments quantify the practical utility of this framework, while synthetic experiments allow us to probe and compare each design choice in a controlled setting.

## 4.1 Datasets

We pretrained SEA models on 6,480 synthetic datasets, which constitute approximately 280 million individual observations, each of 10-100 variables.[5] To assess generalization and robustness, we evaluate on unseen in-distribution and out-of-distribution synthetic datasets, as well as two real biological datasets (Sachs et al., 2005; Replogle et al., 2022), using the versions from Wang et al. (2017); Chevalley et al. (2025). To probe for emergent few-shot behavior, we down-sample both the training and testing sets. We also include experiments on simulated mRNA datasets with unseen datasets in Appendix C.3 (Dibaeinia & Sinha, 2020).

The training datasets were constructed by 1) sampling Erdős-Rényi and scale free graphs with $N = 10, 20, 100$ nodes and $E = N, 2N, 3N, 4N$ expected edges; 2) sampling random instantiations of causal mechanisms (Linear, NN with additive/non-additive Gaussian noise); and 3) iteratively sampling observations in topological order (details in Appendix B.1). From every causal graphical model (steps 1-2), we generated two datasets, each with $1000N$ points: either all observational, or split equally among regimes (observational and perfect single-node interventions on all nodes). All models that can accommodate interventions were run on the interventional datasets, with complete knowledge of the intervention target identities. The remaining were run on the observational datasets. We generated 90 training, 5 validation, and 5 testing datasets for each combination. For testing, we also sampled out-of-distribution datasets with 1) Sigmoid and Polynomial mechanisms with Gaussian noise; and 2) Linear with additive non-Gaussian noise.

## 4.2 Metrics

We report standard causal discovery metrics. These include both discrete and continuous metrics, as neural networks can be notoriously uncalibrated (Guo et al., 2017), and arbitrary discretization thresholds may impact the robustness of findings (Ng et al., 2024; Schaeffer et al., 2023). For all continuous metrics, we exclude the diagonal since several baselines manually set it to zero (Brouillard et al., 2020; Lopez et al., 2022).

**SHD:** Structural Hamming distance is the minimum number of edge edits required to match two graphs (Tsamardinos et al., 2006) (predicted and true DAGs, or the implied CPDAGs). Discretization thresholds are as published or default to 0.5.

**mAP:** Mean average precision computes the area under precision-recall curve per edge and averages over the graph (predicted and true DAGs, or undirected skeletons). The random guessing baseline depends on the positive rate.

**AUC:** Area under the ROC curve (Bradley, 1997) computed per edge (binary prediction) and averaged over the graph. For each edge, 0.5 indicates random guessing, while 1 indicates perfect performance.

**Orientation accuracy:** We compute the accuracy of edge orientations as

$$\text{OA} = \frac{\sum_{(i,j) \in E} \mathbb{1}\{P(i,j) > P(j,i)\}}{\|E\|}. \tag{12}$$

Since OA is normalized by $\|E\|$, it is invariant to the positive rate. In contrast to orientation F1 (Geffner et al., 2024), it is also invariant to the assignment of forward/reverse edges as 1/0.

---

[5]3 mechanisms, 3 graph sizes, 4 sparsities, 2 topologies, $1000N$ examples, 90 datasets $\rightarrow$ 280,800,000 examples. For a sense of scale, single cell foundation models are trained on 300K (Rosen et al., 2024) to 30M cells (Cui et al., 2024).

### 4.3 Baselines

We compare against several deep learning and classical baselines. All baselines were trained and/or run from scratch on each testing dataset using their published code and hyperparameters, except Avici (their recommended checkpoint was trained on their synthetic train and test sets after publication, Appendix B.2).

**DCDI** (Brouillard et al., 2020) extracts the causal graph as a parameter of a generative model. The G and Dsf variants use Gaussian or deep sigmoidal flow likelihoods, respectively. **DCD-FG** (Lopez et al., 2022) follows DCDI-G, but factorizes the graph into a product of two low-rank matrices for scalability. **DiffAn** (Sanchez et al., 2023) uses the trained model's Hessian to obtain a topological ordering, followed by a classical pruning algorithm. **AVICI** (Lorch et al., 2022) uses an amortized inference approach to estimate $P(G \mid D)$ over a class of data-generating mechanisms via variational inference. **VarSort** (a.k.a. "sort and regress") (Reisach et al., 2021) sorts nodes by marginal variance and sparsely regresses nodes based on their predecessors. This naive baseline is intended to reveal artifacts of synthetic data generation. **FCI, GIES** quantify the predictive power of FCI and GIES estimates, when run over all nodes. VarSort, Fci, and Gies were run using non-parametric bootstrapping (Friedman et al., 1999), with 100 estimates of the full graph (based on 1000 examples each), where the final prediction for each edge is its frequency of appearing as directed. Since these methods are treated as oracles, the bootstrapping strategy was selected to maximize test performance. Visualizations of the bootstrapped graph can be found in Figures 7 and 8.

## 5 Results

We highlight representative results in each section, with additional experiments and analyses in Appendix C.

1. Section 5.1 examines the case where we have no prior knowledge about the data. Our models achieve high performance out-of-the-box, even when the data are misspecified or out-of-domain.
2. Section 5.2 focuses on the case where we do know (or can estimate) the class of causal mechanisms or exogenous noise. We show that adapting our pretrained models with this information at zero/low cost leads to substantial improvement and exceeds the best baseline trained from scratch.
3. Section 5.3 analyzes Sea predictions in context of classic identifiability theory. In particular, we focus on the linear Gaussian case, and show that Sea approaches "oracle" performance (with respect to the MEC), while simply running a classic discovery algorithm cannot, on our finite datasets.
4. Section 5.4 contains a variety of ablation studies. In particular, Sea exhibits impressive low-data performance, requiring only 400 samples to perform well on $N = 100$ datasets. We also ablate estimation hyperparameters and the contribution of marginal/global features.

### 5.1 SEA generalizes to out-of-distribution, misspecified, and real datasets

Table 1 summarizes our controlled experiments on synthetic data. Sea exceeds all baselines in the Linear case, which matches the models' assumptions exactly (causal discovery algorithms and inverse covariance). In the misspecified (NN) or misspecified *and* out-of-distribution settings (Sigmoid, Polynomial), Sea also attains the best performance in the vast majority of cases, even though Dcdi and Avici both have access to the raw data. Furthermore, our models outperform VarSort in every single setting, while most baselines are unable to do so consistently. This indicates that our models do not simply overfit to spurious features of the synthetic data generation process.

Table 2 illustrates that we exceed baselines on single cell gene expression data from CausalBench (Chevalley et al., 2025; Replogle et al., 2022). Furthermore, when we increase the subset size to $b = 2000$, we achieve very high precision (0.838) over 2834 predicted edges. Sea runs within 5s on this dataset of 162k cells and $N = 622$ genes, while the fastest naive baseline takes 5 minutes and the slowest deep learning baseline takes 9 hours (run in parallel on subsets of genes).

### 5.2 SEA adapts to new data assumptions with zero to minimal finetuning

We illustrate two strategies that allow us to use pretrained Sea models with different implicit assumptions. First, if two causal discovery algorithms share the same output format, they can be used interchangeably

Table 1: Synthetic experiments. Mean/std over 5 distinct Erdős-Rényi graphs, with metrics relative to ground truth DAG. DiffAn, VarSort, Fci, Sea(Fci) run on observational data only. Evaluation w.r.t. CPDAG and undirected skeleton in Tables 14 and 15, with additional baselines and ablations in Appendix C. † indicates o.o.d. setting. ∗ indicates non-parametric bootstrapping. Runtimes with 1 CPU and 1 V100 GPU.

| N | E | Model | Linear | | NN add. | | Sigmoid† | | Polynomial† | | Overall |
|---|---|---|---|---|---|---|---|---|---|---|---|
| | | | mAP ↑ | SHD ↓ | mAP ↑ | SHD ↓ | mAP ↑ | SHD ↓ | mAP ↑ | SHD ↓ | Time(s) ↓ |
| 20 | 20 | Dcdi-G | $0.59_{\pm.12}$ | $6.4_{\pm.9}$ | $0.78_{\pm.07}$ | $\mathbf{3.0}_{\pm.7}$ | $0.36_{\pm.06}$ | $42.7_{\pm.3}$ | $0.42_{\pm.08}$ | $10.4_{\pm.4}$ | 4735.7 |
| | | Dcdi-Dsf | $0.66_{\pm.16}$ | $5.2_{\pm.3}$ | $0.69_{\pm.18}$ | $4.2_{\pm.5}$ | $0.37_{\pm.04}$ | $43.2_{\pm.4}$ | $0.26_{\pm.08}$ | $15.7_{\pm.2}$ | 3569.1 |
| | | DiffAn | $0.19_{\pm.09}$ | $40.2_{\pm4.4}$ | $0.16_{\pm.10}$ | $38.6_{\pm3.1}$ | $0.29_{\pm.11}$ | $19.2_{\pm.6}$ | $0.09_{\pm.03}$ | $49.7_{\pm4.6}$ | 434.3 |
| | | Avici | $0.48_{\pm.17}$ | $17.2_{\pm.1}$ | $0.59_{\pm.09}$ | $10.8_{\pm.1}$ | $0.42_{\pm.13}$ | $17.2_{\pm.8}$ | $0.24_{\pm.08}$ | $18.4_{\pm.1}$ | 2.0 |
| | | VarSort* | $0.81_{\pm.08}$ | $10.0_{\pm.4}$ | $0.81_{\pm.15}$ | $6.6_{\pm.7}$ | $0.50_{\pm.13}$ | $16.1_{\pm.7}$ | $0.33_{\pm.13}$ | $17.1_{\pm.1}$ | 0.4 |
| | | Fci* | $0.66_{\pm.07}$ | $19.0_{\pm.3}$ | $0.42_{\pm.19}$ | $17.4_{\pm.2}$ | $0.56_{\pm.08}$ | $18.5_{\pm.5}$ | $0.41_{\pm.14}$ | $18.9_{\pm.3}$ | 22.2 |
| | | Gies* | $0.84_{\pm.08}$ | $7.4_{\pm.0}$ | $0.79_{\pm.07}$ | $9.0_{\pm.1}$ | $0.71_{\pm.10}$ | $12.5_{\pm.7}$ | $0.62_{\pm.09}$ | $13.7_{\pm.7}$ | 482.1 |
| | | Sea (Fci) | $\mathbf{0.96}_{\pm.03}$ | $3.2_{\pm.6}$ | $0.91_{\pm.04}$ | $5.0_{\pm.8}$ | $\mathbf{0.85}_{\pm.09}$ | $\mathbf{6.7}_{\pm.1}$ | $\mathbf{0.69}_{\pm.09}$ | $\mathbf{9.8}_{\pm.2}$ | 4.2 |
| | | Sea (Gies) | $\mathbf{0.97}_{\pm.02}$ | $\mathbf{3.0}_{\pm.9}$ | $\mathbf{0.94}_{\pm.03}$ | $3.4_{\pm.4}$ | $0.84_{\pm.07}$ | $8.1_{\pm.8}$ | $\mathbf{0.69}_{\pm.12}$ | $10.1_{\pm.9}$ | 3.0 |
| 100 | 400 | Dcd-Fg | $0.05_{\pm.00}$ | $3068_{\pm131}$ | $0.07_{\pm.00}$ | $3428_{\pm154}$ | $0.13_{\pm.02}$ | $3601_{\pm272}$ | $0.12_{\pm.03}$ | $3316_{\pm698}$ | 1838.2 |
| | | Avici | $0.12_{\pm.02}$ | $391_{\pm8}$ | $0.17_{\pm.01}$ | $407_{\pm19}$ | $0.10_{\pm.02}$ | $398_{\pm11}$ | $0.03_{\pm.00}$ | $402_{\pm19}$ | 9.3 |
| | | VarSort* | $0.80_{\pm.02}$ | $224_{\pm10}$ | $0.18_{\pm.03}$ | $1139_{\pm269}$ | $0.51_{\pm.05}$ | $350_{\pm15}$ | $0.27_{\pm.04}$ | $380_{\pm17}$ | 5.1 |
| | | Sea (Fci) | $0.84_{\pm.02}$ | $162_{\pm12}$ | $0.04_{\pm.00}$ | $403_{\pm16}$ | $0.63_{\pm.03}$ | $247_{\pm17}$ | $0.34_{\pm.04}$ | $\mathbf{325}_{\pm22}$ | 19.2 |
| | | Sea (Gies) | $\mathbf{0.91}_{\pm.01}$ | $\mathbf{116}_{\pm7}$ | $\mathbf{0.27}_{\pm.10}$ | $\mathbf{364}_{\pm34}$ | $\mathbf{0.69}_{\pm.03}$ | $\mathbf{218}_{\pm21}$ | $\mathbf{0.38}_{\pm.04}$ | $328_{\pm22}$ | 3.1 |

Table 2: Results on K562 single cell data, with STRING database (physical) as ground truth. Baselines taken from Chevalley et al. (2025).

| Model | P ↑ | R ↑ | F1 ↑ | Time(s) ↓ |
|---|---|---|---|---|
| GRNboost | 0.070 | **0.710** | 0.127 | 316 |
| Gies | 0.190 | 0.020 | 0.036 | 2350 |
| NoTears | 0.080 | 0.620 | 0.142 | 32883 |
| Dcdi-G | 0.180 | 0.030 | 0.051 | 16561 |
| Dcdi-Dsf | 0.140 | 0.040 | 0.062 | 5709 |
| Dcd-Fg | 0.110 | 0.070 | 0.086 | 6368 |
| Sea (G)+Corr | 0.491 | 0.109 | **0.179** | **4** |
| with $b = 2000$ | **0.838** | 0.093 | 0.167 | 5 |

Table 3: Performance on Sachs (C.5) varies depending on implicit (Avici training set) and explicit (Sea variants) assumptions.

| Model | mAP ↑ | AUC ↑ | SHD ↓ |
|---|---|---|---|
| Dcdi-Dsf | 0.20 | 0.59 | 20.0 |
| Avici-L | 0.35 | 0.78 | 20.0 |
| Avici-R | 0.29 | 0.65 | 18.0 |
| Avici-L+R | **0.59** | **0.83** | 14.0 |
| Sea (F) | 0.23 | 0.54 | 24.0 |
| +Kci | 0.33 | 0.63 | 14.0 |
| +Corr | 0.41 | 0.70 | 15.0 |
| +Kci+Corr | 0.49 | 0.71 | **13.0** |

for marginal estimation. On observational, linear *non*-Gaussian data, replacing the Ges algorithm with Lingam (Shimizu et al., 2006) is beneficial without any other change (Table 4). The same improvement can be observed on Polynomial and Sigmoid non-additive data, when running Fci with a polynomial kernel conditional independence test (Kci, Zhang et al. (2011)) instead of the Fisherz test, which assumes linearity (Table 5). In principle, different algorithms might make different mistakes, so this strategy could lead to out-of-distribution inputs for the pretrained aggregator. While comparing across algorithms is not a primary focus of this work and requires further theoretical study, we notice similar performance for alternate estimation algorithms with linear Gaussian assumptions (Table 7), regardless of discovery strategy (GRaSP (Lam et al., 2022)). The gap is larger for LiNGAM, which assumes non-Gaussianity (Table 25).

Another strategy is to "finetune" the aggregator, either fully or using low-cost methods like LoRA (Hu et al., 2022). Specifically, we keep the same training set and classification objective, while changing the input's featurization, e.g. a different global statistic. Here, we show that finetuning our models for distance correlation (Dcor) is beneficial in both Tables 4 and 5, and the combination of strategies results in the highest performance overall, surpassing the best baseline trained from scratch (Dcdi-Dsf).

Table 4: Adapting Sea to linear non-Gaussian (Uniform) noise. Lingam run without finetuning; Sea(G) finetuned for distance correlation.

| Model | N=10, E=10 | | N=20, E=20 | |
|---|---|---|---|---|
| | mAP ↑ | SHD ↓ | mAP ↑ | SHD ↓ |
| Dcdi-Dsf | 0.34 | 22.3 | 0.32 | 63.0 |
| Lingam* | 0.34 | 7.2 | 0.30 | 18.8 |
| Sea (G) | 0.26 | 12.7 | 0.12 | 46.6 |
| +lingam | 0.52 | 10.1 | 0.22 | 39.7 |
| +dcor | 0.44 | 8.0 | 0.21 | 33.1 |
| +ling+dcor | **0.76** | **4.6** | **0.67** | **14.2** |

Table 5: Adapting Sea to polynomial, sigmoid non-additive (N=10, E=10). Fci run with Kci test; Sea(F) finetuned for distance correlation.

| Model | Polynomial | | Sigmoid | |
|---|---|---|---|---|
| | mAP ↑ | SHD ↓ | mAP ↑ | SHD ↓ |
| Dcdi-Dsf | 0.39 | 9.8 | 0.81 | 13.6 |
| Fci* | 0.12 | 10.6 | 0.53 | 8.1 |
| Sea (F) | 0.22 | 10.6 | 0.59 | 4.8 |
| +kci | 0.30 | 10.6 | 0.59 | 5.5 |
| +dcor | 0.45 | 9.6 | **0.90** | **2.1** |
| +kci+dcor | **0.52** | **8.2** | 0.86 | 3.4 |

Table 6: Sea respects identifiability theory. Observational setting, *standardized* (-std) $N = 10, E = 10$ linear Gaussian test datasets with $> 1$ graph in Markov equivalence class (MEC). Top: oracle performance based on true MEC (see left). Bottom: trained Sea approaches oracle performance, while FCI is very noisy.

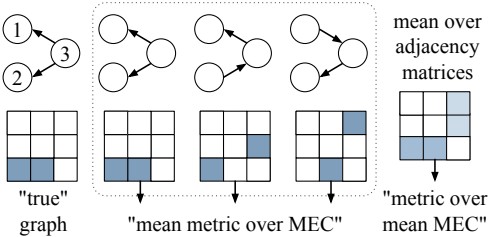

| Model | mAP(↑) | AUC(↑) | SHD(↓) | OA(↑) |
|---|---|---|---|---|
| metric over mean MEC | 0.88 ±.10 | 0.98 ±.03 | 2.0 ±1.0 | 0.74 ±.22 |
| mean metric over MEC | 0.74 ±.21 | 0.91 ±.07 | 1.2 ±.69 | 0.84 ±.13 |
| Sea(Fci)-std | 0.83 ±.16 | 0.97 ±.04 | 3.3 ±2.3 | 0.69 ±.21 |
| Sea(Fci)+Corr-std | 0.84 ±.14 | 0.96 ±.03 | 5.0 ±4.5 | 0.85 ±.14 |
| Fci-std | 0.49 ±.28 | 0.75 ±.16 | 9.3 ±2.8 | 0.49 ±.29 |

On real data from unknown distributions, these two strategies enable the ability to run causal discovery with different assumptions, which may be coupled with unsupervised methods for model selection (Faller et al., 2023). Table 3 illustrates this idea using the Sachs proteomics dataset. Sea can be run directly with a different estimation algorithm (FCI with polynomial kernel "Kci"), or finetuned for around 4-6 hours on 1 A6000 and $< 4$ GB of GPU memory (correlation "Corr"). In contrast, methods like Avici must simulate new datasets based on each new assumption and re-train/finetune on these data (reportedly around 4 days).

### 5.3 SEA respects identifiability theory

While the identifiability of specific causal models is not a primary focus of this work, we show that Sea still respects classic identifiability theory. Specifically, while linear Gaussian models are known to be unidentifiable, Table 1 might suggest that both Sea and Dcdi perform quite well on these data – better than would be expected if graphs were only identifiable up to their Markov equivalence classes. This empirical "identifiability" may be the consequence of two findings. Common synthetic data generation schemes tend to result in marginal variances that reflect topological order (Reisach et al., 2021), and in additive noise models, it has been shown that marginal variances that are the "same or weakly monotone increasing in the [topological] ordering" result in uniquely identifiable graphs (Park, 2020). Data standardization can eliminate these artifacts of synthetic data generation. In Table 5.3, we see that after standardizing linear Gaussian data, our model performs no better than randomly selecting a graph from the Markov equivalence class (enumerated via `pcalg` (Kalisch et al., 2012)). The classic FCI algorithm is unable to reach this upper bound, suggesting that the amortized inference framework allows us to perform better in finite datasets.

### 5.4 Ablation studies

In addition to high performance and flexibility, one of the hallmarks of foundation models is their ability to act as few-shot learners when scaling up (Brown et al., 2020). We first confirm that Sea is indeed data-efficient, requiring only around 300-400 examples for performance to converge on datasets of $N = 100$ variables, and outperforms inverse covariance (computed with 500 examples) at only 200 examples (Figure 4A). To probe

Table 7: SEA is generally insensitive to swapping estimation other algorithms with linear Gaussian assumptions, at *inference* time. Results on $N = 10$ observational setting. FCI cannot be used with SEA(G) since FCI outputs a PAG, not a CPDAG.

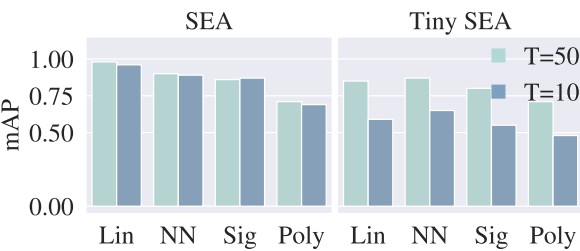

| Inference estimator | SEA (FCI) | | | | SEA (GIES) | | | |
|---|---|---|---|---|---|---|---|---|
| | Lin. | NN | Sig. | Poly. | Lin. | NN | Sig. | Poly. |
| FCI | 0.98 | 0.88 | 0.83 | 0.62 | — | — | — | — |
| PC | 0.93 | 0.85 | 0.86 | 0.64 | 0.96 | 0.89 | 0.82 | 0.58 |
| GES | 0.94 | 0.85 | 0.80 | 0.60 | 0.95 | 0.88 | 0.81 | 0.57 |
| GRaSP | 0.93 | 0.85 | 0.80 | 0.61 | 0.95 | 0.88 | 0.81 | 0.57 |

Figure 3: Few-shot learning behavior emerges as training set increases. "Tiny" SEA trained on 1/4 of the data is comparable to the full model on $N = 10$ datasets when given $T = 50$ batches, but is less robust with only $T = 10$.

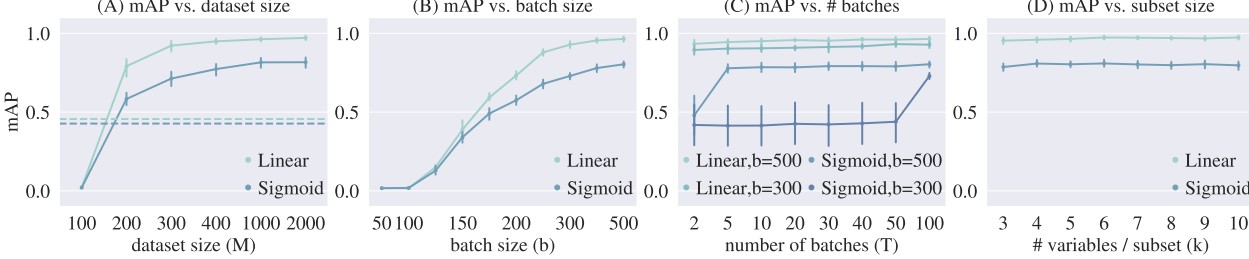

Figure 4: Ablations with SEA (GIES) for estimation parameters on $N = 100, E = 100$. Error bars indicate 95% confidence interval across the 5 i.i.d. datasets of each setting. All parameters are set to the defaults (Section 3.3) unless otherwise noted. (A) Dashed: inverse covariance at $M = 500$. (C) Variance is unusually high for Sigmoid $b = 300$ until $T = 100$, indicating that larger batches result in more stable results.

for how this behavior emerges, we trained a "tiny" version of SEA (GIES) on approximately a quarter of the training data ($N = 10, 20$ datasets, 64.8 million examples). The tiny model performs nearly as well as the original on $N = 10$ datasets when provided $T = 50$ batches, but exhibits much poorer few-shot behavior with only $T = 10$ batches (Figure 3). This demonstrates that SEA is able to ingest large amounts of data, leading to promising few-shot behavior.[6]

We also ablate each parameter of the estimation step to inform best practices. The trade-off between the number and size of batches may be relevant to estimation algorithms that scale poorly with the number of examples, e.g. kernel-based methods (Zhang et al., 2011). When given $T = 100$ batches, SEA reaches reasonable performance at around 250-300 examples per batch (Figure 4B). Figure 4C further illustrates that on the harder Sigmoid datasets, 5 batches of size $b = 500$ are roughly equivalent to 100 batches of size $b = 300$. Finally, increasing the number of variables in each subset has minimal impact (Figure 4D), which is encouraging, as there is no need to incur the exponentially-scaling runtimes associated with larger subsets.

Finally, we analyze the impact of removing marginal estimates or global statistics (Table 8). First, we take a fully pretrained SEA (GIES) and set the corresponding hidden representations to 0. Performance drops more when $h^\rho$ is set to 0, indicating that our *pretrained* aggregator relies more on global statistics, though a sizable gap emerges in both situations. Then, we *re-train* SEA (GIES) on the $N = 10$ datasets, with and without global statistics, so that lack of $\rho$ is in-distribution for the latter model, and the training sets are comparable. Here, we see that the "no $\rho$" version with $T = 50$ estimates is on par with the original architecture with $T = 10$ estimates, so the global statistic is equivalent to $\sim 40$ estimates. This roughly aligns with the theory that global statistics can expedite the skeleton discovery process (Section 3.4), as the number of estimates required to discover the skeleton of a $N = 10$ graph is approximately $\binom{10}{2} = 45$ (Prop. A.9).

---

[6]Due to computational limitations, we were unable to train larger models, as our existing training set requires several hundred GB in memory, and our file system does not support fast dynamic loading.

Table 8: Ablating marginal and global features on SEA (GIES). Top: We set marginal and global representations to 0 (lack of $E'/\rho$ is out-of-distribution) and observe that the pretrained model is more robust to removing $E'$, perhaps since we sample varying $T$ during training. Bottom: Re-train SEA (GIES) on $N = 10$ datasets, with and without global features (lack of $\rho$ is in-distribution). We observe that global features are "worth" $T \approx 40$ estimates of $k = 5$ variables each.

| Model | Linear | | NN add. | | NN. | | Sigmoid | | Polynomial | |
|---|---|---|---|---|---|---|---|---|---|---|
| | mAP ↑ | SHD ↓ | mAP ↑ | SHD ↓ | mAP ↑ | SHD ↓ | mAP ↑ | SHD ↓ | mAP ↑ | SHD ↓ |
| SEA (GIES) | $0.99_{\pm.01}$ | $1.2_{\pm.7}$ | $0.94_{\pm.06}$ | $2.6_{\pm.8}$ | $0.91_{\pm.07}$ | $3.2_{\pm.3}$ | $0.85_{\pm.12}$ | $4.0_{\pm.5}$ | $0.70_{\pm.11}$ | $5.8_{\pm.6}$ |
| $h^\rho \leftarrow 0$ | $0.30_{\pm.17}$ | $29.2_{\pm.4}$ | $0.27_{\pm.18}$ | $29.4_{\pm.8}$ | $0.19_{\pm.09}$ | $29.0_{\pm.0}$ | $0.35_{\pm.17}$ | $27.4_{\pm.4}$ | $0.31_{\pm.15}$ | $27.1_{\pm.9}$ |
| $h^E \leftarrow 0$ | $0.85_{\pm.09}$ | $6.4_{\pm.7}$ | $0.82_{\pm.11}$ | $10.2_{\pm.9}$ | $0.78_{\pm.07}$ | $13.2_{\pm.2}$ | $0.63_{\pm.21}$ | $10.2_{\pm.8}$ | $0.55_{\pm.19}$ | $13.4_{\pm.6}$ |
| $T = 2$ | $0.20_{\pm.04}$ | $31.2_{\pm.5}$ | $0.25_{\pm.06}$ | $29.2_{\pm.4}$ | $0.33_{\pm.10}$ | $27.8_{\pm.1}$ | $0.19_{\pm.07}$ | $30.2_{\pm.9}$ | $0.24_{\pm.09}$ | $28.1_{\pm.9}$ |
| $T = 10$ | $0.62_{\pm.16}$ | $8.0_{\pm.8}$ | $0.69_{\pm.11}$ | $9.6_{\pm.6}$ | $0.66_{\pm.13}$ | $11.2_{\pm.9}$ | $0.62_{\pm.20}$ | $9.2_{\pm.6}$ | $0.50_{\pm.22}$ | $8.9_{\pm.2}$ |
| $T = 50$, no $\rho$ | $0.63_{\pm.13}$ | $6.8_{\pm.1}$ | $0.53_{\pm.07}$ | $6.2_{\pm.9}$ | $0.68_{\pm.20}$ | $6.0_{\pm.1}$ | $0.58_{\pm.15}$ | $7.1_{\pm.1}$ | $0.50_{\pm.14}$ | $7.1_{\pm.5}$ |

## 6 Discussion

Interventional experiments have formed the basis of scientific discovery throughout history, and in recent years, advances in the life sciences have led to datasets of unprecedented scale and resolution (Replogle et al., 2022; Nadig et al., 2024). The goal of these perturbation experiments is to extract causal relationships between biological entities, such as genes or proteins. However, the sheer size, sparsity, and noise level of these data pose significant challenges to existing causal discovery algorithms. Moreover, these real datasets do not fit cleanly into causal frameworks that are designed around fixed sets of data assumptions, either explicit (Spirtes et al., 1995) or implicit (Lorch et al., 2022). In this work, we approached these challenges through a causal discovery "foundation model." Central to this concept were three goals. First, this model should generalize to unseen datasets whose data-generating mechanisms are unknown, and potentially out-of-distribution. Second, it should be easy to steer the model's predictions with inductive biases about the data. Finally, scaling up the model should lead to data-efficiency. We proposed SEA, a framework that yields causal discovery foundation models. SEA was motivated by the idea that classical statistics and discovery algorithms provide powerful descriptors of data that are fast to compute and robust across datasets. Given these statistics, we trained a deep learning model to reproduce faithful causal graphs. Theoretically, we demonstrated that it is possible to infer causal graphs consistent with correct marginal estimates, and that our model is well-specified with respect to this task. Empirically, we implemented two proofs of concept of SEA that perform well across a variety of causal discovery tasks, easily incorporate inductive biases when they are available, and exhibit excellent few-shot behavior when scaled up.

While SEA provides a high-level framework for supervised causal discovery, there are several empirical limitations of the two implementations describe in this paper. These include: 1) an arbitrary, hard-coded maximum of 1000 variables, 2) poor generalization to synthetic cyclic data, 3) erring on the side of sparsity on real data, 4) numeric instability of inverse covariance on larger graphs, and as a result, 5) training requires full precision. The first three aspects may be addressed by modifying the architecture and/or synthetic training datasets, while the latter two can be addressed with more numerically stable statistics, like correlation.

More broadly, the success of supervised causal discovery algorithms derives from the fidelity of the data simulation procedure. In this paper, we present proofs of concept for the modeling framework, but we do not solve the problem of simulating realistic data. Of the data that may exist in the real world, the training data used here represent only a small, perhaps unrealistic, fraction. To achieve any semblance of trustworthiness on real applications, it is crucial to study the characteristics of each domain in detail – including common graph topologies and functional forms (Aguirre et al., 2024); the degree and nature of missing data (Hicks et al., 2018); and sources of measurement error or other covariates (Tran et al., 2020). When possible, we recommend that any insights be triangulated with other sources of knowledge. For example, while this work does not directly provide an inference framework, the predicted structure could be used to parametrize a generative model, whose likelihood could be evaluated on held-out interventions (Hägele et al., 2023). It is

also important to check whether the inferred relationships compound upon any existing biases in the data. This is particularly important for sensitive domains like healthcare or legal applications.

This work also opens several directions for further investigation. The framework we describe utilizes a single causal discovery algorithm and a single global statistic. Classic causal discovery algorithms leverage diverse insights for identifying causal structure, e.g. the non-Gaussianity of noise (Shimizu et al., 2006) vs. conditional independence (Spirtes et al., 1995). Thus, different discovery algorithms or summary statistics may reveal different aspects of the causal structure. Learning to resolve these potentially conflicting views remains unexplored, both from experimental and theoretical perspectives. Furthermore, this work only shows that the axial attention architecture is well-specified as a model class and probes generalization empirically. This motivates theoretical studies into supervised causal discovery with regards to what information can be provably learned, e.g. in the style of of PAC (probably approximately correct) learning (Allen-Zhu et al., 2019).

In summary, we hope that this work will inspire a new avenue of research into causal discovery algorithms that are applicable to and informed by real applications.

## Acknowledgements

We thank Bowen Jing, Felix Faltings, Sean Murphy, and Wenxian Shi for helpful discussions regarding the writing; as well as Jiaqi Zhang, Romain Lopez, Caroline Uhler, and Stephen Bates for helpful feedback regarding the framing of this project. Finally, we thank our action editor Bryon Aragam and our anonymous reviewers for their invaluable suggestions towards improving this paper during the review process.

This material is based upon work supported by the National Science Foundation Graduate Research Fellowship under Grant No. 1745302. We would like to acknowledge support from the NSF Expeditions grant (award 1918839: Collaborative Research: Understanding the World Through Code), Machine Learning for Pharmaceutical Discovery and Synthesis (MLPDS) consortium, and the Abdul Latif Jameel Clinic for Machine Learning in Health.

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

# A Theoretical motivations

Our theoretical contributions focus on two primary directions.

1. We formalize the notion of marginal estimates used in this paper, and prove that given sufficient marginal estimates, it is possible to recover a pattern faithful to the global causal graph. We provide lower bounds on the number of marginal estimates required for such a task, and motivate global statistics as an efficient means to reduce this bound.
2. We show that our proposed axial attention has the capacity to recapitulate the reasoning required for marginal estimate resolution. We provide realistic, finite bounds on the width and depth required for this task.

Before these formal discussions, we start with a toy example to provide intuition regarding marginal estimates and constraint-based causal discovery algorithms.

## A.1 Toy example: Resolving marginal graphs

Consider the Y-shaped graph with four nodes in Figure 5. Suppose we run the PC algorithm on all subsets of three nodes, and we would like to recover the result of the PC algorithm on the full graph. We illustrate how one might resolve the marginal graph estimates. The PC algorithm consists of the following steps (Spirtes et al., 2001).

1. Start from the fully connected, undirected graph on $N$ nodes.
2. Remove all edges $(i, j)$ where $X_i \perp\!\!\!\perp X_j$.
3. For each edge $(i, j)$ and subsets $S \subseteq [N] \setminus \{i, j\}$ of increasing size $n = 1, 2, \ldots, d$, where $d$ is the maximum degree in $G$, and all $k \in S$ are connected to either $i$ or $j$: if $X_i \perp\!\!\!\perp X_j \mid S$, remove edge $(i, j)$.
4. For each triplet $(i, j, k)$, such that only edges $(i, k)$ and $(j, k)$ remain, if $k$ was not in the set $S$ that eliminated edge $(i, j)$, then orient the "v-structure" as $i \rightarrow k \leftarrow j$.
5. (Orientation propagation) If $i \rightarrow j$, edge $(j, k)$ remains, and edge $(i, k)$ has been removed, orient $j \rightarrow k$. If there is a directed path $i \rightsquigarrow j$ and an undirected edge $(i, j)$, then orient $i \rightarrow j$.

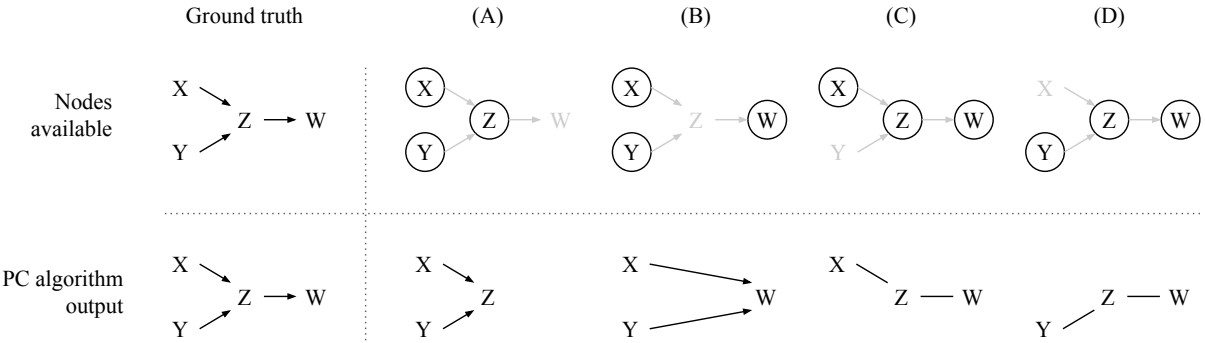

Figure 5: Resolving marginal graphs. Subsets of nodes revealed to the PC algorithm (circled in row 1) and its outputs (row 2).

In each of the four cases, the PC algorithm estimates the respective graphs as follows.

(A) We remove edge $(X, Y)$ via (2) and orient the v-structure.
(B) We remove edge $(X, Y)$ via (2) and orient the v-structure.
(C) We remove edge $(X, W)$ via (3) by conditioning on $Z$. There are no v-structures, so the edges remain undirected.
(D) We remove edge $(Y, W)$ via (3) by conditioning on $Z$. There are no v-structures, so the edges remain undirected.

The outputs (A-D) admit the full PC algorithm output as the only consistent graph on four nodes.

- $X$ and $Y$ are unconditionally independent, so no subset will reveal an edge between $(X, Y)$.

- There are no edges between $(X, W)$ and $(Y, W)$. Otherwise, (C) and (D) would yield the undirected triangle.

- $X, Y, Z$ must be oriented as $X \to Z \leftarrow Y$. Paths $X \to Z \to Y$ and $X \leftarrow Z \leftarrow Y$ would induce an $(X, Y)$ edge in (B). Reversing orientations $X \leftarrow Z \to Y$ would contradict (A).

- $(Y, Z)$ must be oriented as $Y \to Z$. Otherwise, (A) would remain unoriented.

## A.2 Resolving marginal estimates into global graphs

Classical results have characterized the Markov equivalency class of directed acyclic graphs. Two graphs are observationally equivalent if they have the same skeleton and v-structures (Verma & Pearl, 1990). Thus, a pattern $P$ is *faithful* to a graph $G$ if and only if they share the same skeletons and v-structures (Spirtes et al., 1990).

**Definition A.1.** Let $G = (V, E)$ be a directed acyclic graph. A *pattern $P$* is a set of directed and undirected edges over $V$.

**Definition A.2** (Theorem 3.4 from Spirtes et al. (2001))**.** If pattern $P$ is *faithful* to some directed acyclic graph, then $P$ is faithful to $G$ if and only if

1. for all vertices $X, Y$ of $G$, $X$ and $Y$ are adjacent if and only if $X$ and $Y$ are dependent conditional on every set of vertices of $G$ that does not include $X$ or $Y$; and
2. for all vertices $X, Y, Z$, such that $X$ is adjacent to $Y$ and $Y$ is adjacent to $Z$ and $X$ and $Z$ are not adjacent, $X \to Y \leftarrow Z$ is a subgraph of $G$ if and only if $X, Z$ are dependent conditional on every set containing $Y$ but not $X$ or $Z$.

Given data faithful to $G$, a number of classical constraint-based algorithms produce patterns that are faithful to $G$. We denote this set of algorithms as $\mathcal{F}$.

**Theorem A.3** (Theorem 5.1 from Spirtes et al. (2001))**.** *If the input to the PC, SGS, PC-1, PC-2, PC\*, or IG algorithms faithful to directed acyclic graph $G$, the output is a pattern that represents the faithful indistinguishability class of $G$.*

The algorithms in $\mathcal{F}$ are sound and complete *if* there are no unobserved confounders.

Let $P_V$ be a probability distribution that is Markov, minimal, and faithful to $G$. Let $D \in \mathbb{R}^{M \times N} \sim P_V$ be a dataset of $M$ observations over all $N = |V|$ nodes.

Consider a subset $S \subseteq V$. Let $D[S]$ denote the subset of $D$ over $S$,

$$D[S] = \{x_{i,v} : v \in S\}_{i=1}^{N}, \tag{13}$$

and let $G[S]$ denote the subgraph of $G$ induced by $S$

$$G[S] = (S, \{(i, j) : i, j \in S, (i, j) \in E\}. \tag{14}$$

If we apply any $f \in \mathcal{F}$ to $D[S]$, the results are *not* necessarily faithful to $G[S]$, as now there may be latent confounders in $V \setminus S$ (by construction). We introduce the term *marginal estimate* to denote the resultant pattern that, while not faithful to $G[S]$, is still informative.

**Definition A.4** (Marginal estimate)**.** A pattern $E'$ is a *marginal estimate* of $G[S]$ if and only if

1. for all vertices $X, Y$ of $S$, $X$ and $Y$ are adjacent if and only if $X$ and $Y$ are dependent conditional on every set of vertices of $S$ that does not include $X$ or $Y$; and
2. for all vertices $X, Y, Z$, such that $X$ is adjacent to $Y$ and $Y$ is adjacent to $Z$ and $X$ and $Z$ are not adjacent, $X \to Y \leftarrow Z$ is a subgraph of $S$ if and only if $X, Z$ are dependent conditional on every set containing $Y$ but not $X$ or $Z$.

---

**Algorithm 1** Resolve marginal estimates of $f \in \mathcal{F}$

---

1: **Input:** Data $\mathcal{D}_G$ faithful to $G$
2: Initialize $E' \leftarrow K_N$ as the complete undirected graph on $N$ nodes.
3: **for** $S \in \mathcal{S}_{d+2}$ **do**
4:     Compute $E'_S = f(\mathcal{D}_{G[S]})$
5:     **for** $(i, j) \notin E'_S$ **do**
6:         Remove $(i, j)$ from $E'$
7:     **end for**
8: **end for**
9: **for** $E'_S \in \{E'_S\}_{\mathcal{S}_{d+2}}$ **do**
10:     **for** v-structure $i \rightarrow j \leftarrow k$ in $E'_S$ **do**
11:         **if** $\{i, j\}, \{j, k\} \in E'$ and $\{i, k\} \notin E'$ **then**
12:             Assign orientation $i \rightarrow j \leftarrow k$ in $E'$
13:         **end if**
14:     **end for**
15: **end for**
16: Propagate orientations in $E'$ (optional).

---

**Proposition A.5.** *Let $G = (V, E)$ be a directed acyclic graph with maximum degree $d$. For $S \subseteq V$, let $E'_S$ denote the marginal estimate over $S$. Let $\mathcal{S}_d$ denote the superset that contains all subsets $S \subseteq V$ of size at most $d$. Algorithm 1 maps $\{E'_S\}_{S \in \mathcal{S}_{d+2}}$ to a pattern $E'$ faithful to $G$.*

On a high level, lines 3-8 recover the undirected "skeleton" graph of $E^*$, lines 9-15 recover the v-structures, and line 16 references step 5 in Section A.1.

*Remark* A.6. In the PC algorithm (Spirtes et al. (2001), A.1), its derivatives, and Algorithm 1, there is no need to consider separating sets with cardinality greater than maximum degree $d$, since the maximum number of independence tests required to separate any node from the rest of the graph is equal to number of its parents plus its children (due to the Markov assumption).

**Lemma A.7.** *The undirected skeleton of $E^*$ is equivalent to the undirected skeleton of $E'$*

$$C^* := \{\{i, j\} \mid (i, j) \in E^* \text{ or } (j, i) \in E^*\} = \{\{i, j\} \mid (i, j) \in E' \text{ or } (j, i) \in E'\} := C'. \tag{15}$$

*That is, $\{i, j\} \in C^* \iff \{i, j\} \in C'$.*

*Proof.* It is equivalent to show that $\{i, j\} \notin C^* \iff \{i, j\} \notin C'$

$\Rightarrow$ If $\{i, j\} \notin C*$, then there must exist a separating set $S$ in $G$ of at most size $d$ such that $i \perp\!\!\!\perp j \mid S$. Then $S \cup \{i, j\}$ is a set of at most size $d + 2$, where $\{i, j\} \notin C'_{S \cup \{i, j\}}$. Thus, $\{i, j\}$ would have been removed from $C'$ in line 6 of Algorithm 1.

$\Leftarrow$ If $\{i, j\} \notin C'$, let $S$ be a separating set in $\mathcal{S}_{d+2}$ such that $\{i, j\} \notin C'_{S \cup \{i, j\}}$ and $i \perp\!\!\!\perp j \mid S$. $S$ is also a separating set in $G$, and conditioning on $S$ removes $\{i, j\}$ from $C^*$. $\qquad \square$

**Lemma A.8.** *A v-structure $i \rightarrow j \leftarrow k$ exists in $E^*$ if and only if there exists the same v-structure in $E'$.*

*Proof.* V-structures are oriented $i \rightarrow j \leftarrow k$ in $E^*$ if there is an edge between $\{i, j\}$ and $\{j, k\}$ but not $\{i, k\}$; and if $j$ was not in the conditioning set that removed $\{i, k\}$. Algorithm 1 orients v-structures $i \rightarrow j \leftarrow k$ in $E'$ if they are oriented as such in any $E'_S$; and if $\{i, j\}, \{j, k\} \in E', \{i, k\} \notin E'$

$\Rightarrow$ Suppose for contradiction that $i \rightarrow j \leftarrow k$ is oriented as a v-structure in $E^*$, but not in $E'$. There are two cases.

1. No $E'_S$ contains the undirected path $i - j - k$. If either $i - j$ or $j - k$ are missing from any $E'_S$, then $E^*$ would not contain $(i, j)$ or $(k, j)$. Otherwise, if all $S$ contain $\{i, k\}$, then $E^*$ would not be missing $\{i, k\}$ (Lemma A.7).

2. In every $E'_S$ that contains $i-j-k$, $j$ is in the conditioning set that removed $\{i,k\}$, i.e. $i \perp\!\!\!\perp k \mid S, S \ni j$. This would violate the faithfulness property, as $j$ is neither a parent of $i$ or $k$ in $E^*$, and the outputs of the PC algorithm are faithful to the equivalence class of $G$ (Theorem 5.1 Spirtes et al. (2001)).

$\Leftarrow$ Suppose for contradiction that $i \to j \leftarrow k$ is oriented as a v-structure in $E'$, but not in $E^*$. By Lemma A.7, the path $i - j - k$ must exist in $E^*$. There are two cases.

1. If $i \to j \to k$ or $i \leftarrow j \leftarrow k$, then $j$ must be in the conditioning set that removes $\{i,k\}$, so no $E'_S$ containing $\{i,j,k\}$ would orient them as v-structures.
2. If $j$ is the root of a fork $i \leftarrow j \to k$, then as the parent of both $i$ and $k$, $j$ must be in the conditioning set that removes $\{i,k\}$, so no $E'_S$ containing $\{i,j,k\}$ would orient them as v-structures.

Therefore, all v-structures in $E'$ are also v-structures in $E^*$. □

*Proof of Proposition A.5.* Given data that is faithful to $G$, Algorithm 1 produces a pattern $E'$ with the same connectivity and v-structures as $E^*$. Any additional orientations in both patterns are propagated using identical, deterministic procedures, so $E' = E^*$. □

This proof presents a deterministic but inefficient algorithm for resolving marginal subgraph estimates. In reality, it is possible to recover the undirected skeleton and the v-structures of $G$ without checking all subsets $S \in \mathcal{S}_{d+2}$.

**Proposition A.9** (Skeleton bounds). *Let $G = (V, E)$ be a directed acyclic graph with maximum degree $d$. It takes $O(N^2)$ marginal estimates over subsets of size $d + 2$ to recover the undirected skeleton of $G$.*

*Proof.* Following Lemma A.7, an edge $(i, j)$ is not present in $C$ if it is not present in any of the size $d + 2$ estimates. Therefore, every pair of nodes $\{i, j\}$ requires only a single estimate of size $d + 2$, so it is possible to recover $C$ in $\binom{N}{2}$ estimates. □

**Proposition A.10** (V-structures bounds). *Let $G = (V, E)$ be a directed acyclic graph with maximum degree $d$ and $\nu$ v-structures. It is possible to identify all v-structures in $O(\nu)$ estimates over subsets of at most size $d + 2$.*

*Proof.* Each v-structure $i \to j \leftarrow k$ falls under two cases.

1. $i \perp\!\!\!\perp k$ unconditionally. Then an estimate over $\{i, j, k\}$ will identify the v-structure.
2. $i \perp\!\!\!\perp k \mid S$, where $j \notin S \subset V$. Then an estimate over $S \cup \{i, j, k\}$ will identify the v-structure. Note that $|S| \le d + 2$ since the degree of $i$ is at least $|S| + 1$.

Therefore, each v-structure only requires one estimate, and it is possible to identify all v-structures in $O(\nu)$ estimates. □

There are three takeaways from this section.

1. If we exhaustively run a constraint-based algorithm on all subsets of size $d + 2$, it is trivial to recover the estimate of the full graph. However, this is no more efficient than running the causal discovery algorithm on the full graph.
2. In theory, it is possible to recover the undirected graph in $O(N^2)$ estimates, and the v-structures in $O(\nu)$ estimates. However, we may not know the appropriate subsets ahead of time.
3. In practice, if we have a surrogate for connectivity, such as the global statistics used in SEA, then we can vastly reduce the number of estimates used to eliminate edges from consideration, and more effectively focus on sampling subsets for orientation determination.

## A.3   Model specification

**Model specification and universality**   In classical statistics, a statistical model can be expressed as a pair $(S, P_\theta)$ for $\theta \in \Theta$, where $S$ is the sample space, $P$ is the family of distributions on $S$, and $\Theta$ is the space of parameters (McCullagh, 2002). Let $x_1, \ldots, x_N$ be observations of i.i.d. random variables in $S$, and let $P^*$ denote their common distribution. A statistical model is *well-specified* if $P^* = P_\theta$ for some $\theta \in \Theta$. Identifying the most appropriate class of models has long been an area of interest in statistics (Akaike, 1973). For simple models like linear regression, there are many ways to test for model specification, where the alternative may be that the data follow a non-linear relationship (Davidson & MacKinnon, 1981).

In machine learning, however, even the simplest architectures vary immensely. Whether a neural network is "well-specified" depends on many aspects, such as its width (number of hidden units), depth (number of layers), activation functions (sources of non-linearity), and more. Thus, instead of testing whether each neural network architecture is well-specified for each experiment, it is more common to show universality (or lack thereof). That is, given a class of functions $\mathcal{F}$ and the space of parameters $\Theta$, for every $f \in \mathcal{F}$ does there exist a $\theta \in \Theta$ that allows the neural network to approximate $f$ to arbitrary accuracy? For example, the most well-cited work for the universality of multi-layer perceptrons (Hornik et al., 1989) showed that:

> "standard multilayer feedforward networks with as few as one hidden layer using arbitrary **squashing functions** are capable of approximating any Borel measurable function from one finite dimensional space to another to any desired degree of accuracy, **provided sufficiently many hidden units are available**."

Note that this definition *excludes* neural networks that use rectified linear units (ReLU is unbounded), whose variants are ubiquitous today, as well as neural networks with bounded width. Universality under alternate assumptions has been addressed by many later works, e.g. Maiorov & Pinkus (1999); Shen et al. (2022).

With respect to modern Transformer architectures (Vaswani et al., 2017), the existing literature are similarly subject to constraints. For example, Pérez et al. (2019) shows that Transformers equipped with positional encodings are Turing complete, given *infinite precision*. Alternatively, Yun et al. (2020) proves that Transformers can approximate arbitrary continuous "sequence-to-sequence" functions, but requires in the worst case, *exponential depth*. Finally, Sanford et al. (2024) shows that one-dimensional self attention is *unable* to detect arbitrary triples along a sequence, unless the depth scales linearly as the input size (sequence length). This last case is particularly relevant to our setting, as v-structures are relations of three nodes, but even modern large language models "only" contain $\sim 100$ layers (e.g. GPT-4 is reported to have 120 layers). In general, these works present constructive proofs of either $\theta$ (Pérez et al., 2019; Yun et al., 2020) or pathological cases that cannot be handled (Sanford et al., 2024).

In this paper, we do *not* consider universality in a general sense, but rather limit our scope to the algorithms presented in Section A.2. This will allow us to make more realistic assumptions regarding the model class.

**Causal identifiability for continuous discovery algorithms**   Identifiability is central to causal discovery, as it is important to understand the degree to which a causal model can or cannot be inferred from data. In contrast to classic approaches, causal discovery algorithms that rely on continuous optimization must consider several additional aspects with respect to identifiability – namely, their objective; optimization dynamics; and model specification. We discuss representative works in light of these considerations.

First, the optima of the objectives must correspond to true graphs. NoTears (Zheng et al., 2018) focuses on the linear SEM case with least-squares loss, and they cite earlier literature regarding the identifiability of this setup (van de Geer & Bühlmann, 2013; Loh & Bühlmann, 2014; Aragam et al., 2015). DcDi (Brouillard et al., 2020) proves that the graph that maximizes their proposed score is $\mathcal{I}$-Markov equivalent to the ground truth, subject to faithfulness and other regularity conditions. Since amortized causal discovery algorithms tend to be trained on data from a variety of data-generating processes, the identifiability of an arbitrary graph is less clear. Ke et al. (2023) cites Eberhardt et al. (2006) and claims that any graph is identifiable "in the limit of an infinite amount of [single-node hard] interventional data samples." In our interventional setting, this also holds. Avici (Lorch et al., 2022) focuses solely on inferring graphs from data, without addressing whether

they are identifiable. If we suppose that the graphs are identifiable, then in all three cases, the objectives can be written in terms of the KL divergence between the true and predicted edge distributions, which reaches its minimum when the predicted graph matches the true graph.

Second, the optimization process must not only converge to an optimum, but also (for amortized models) generalize to unseen data. This is, in general, difficult to show. Both NoTears and DcDI acknowledge that due to the non-convexity of this optimization problem, the optimizer may converge to a stable point, which is not necessarily the global optimum. Avici includes an explicit acyclicity constraint, so they additionally cite Nandwani et al. (2019); Jin et al. (2020) for inspiration in constrained optimization of a neural network. Regarding generalization, existing results on what can be "provably" learned by neural networks are limited to simple architectures and/or algorithms (Allen-Zhu et al., 2019; Shao et al., 2022). Instead, both Avici and our work assess generalization empirically, by holding out certain causal mechanisms from training.

Finally, the implementation of each model must be well-specified. Since NoTears assumes linearity, it directly optimizes the (weighted) adjacency, and the only concern is whether linearity holds. DcDI cites that deep sigmoidal flows are universal density approximators (Huang et al., 2018), which in turn invokes the classic result that a multilayer perceptron with sigmoid activations is a universal approximator (Cybenko, 1989). Avici does not discuss this aspect of their architecture, which is also based on sparse attention, but is otherwise quite different from ours. The following section proves by construction that our model is well-specified.

**Axial attention architecture is well-specified**   In the context of Section A.2, we show that three axial attention blocks (model depth) are sufficient to recover the skeleton and v-structures in $O(N)$ width, and we require $O(L)$ to propagate orientations along paths of length $L$. In the following section, we first formalize the notion of a neural network architecture's capacity to "implement" an algorithm. Then we prove Theorem 3.1 by construction.

**Definition A.11.** Let $f$ be a map from finite sets $Q$ to $F$, and let $\phi$ be a map from finite sets $Q_\Phi$ to $F_\Phi$. We say $\phi$ *implements* $f$ if there exists injection $g_{\text{in}} : Q \to Q_\Phi$ and surjection $g_{\text{out}} : F_\Phi \to F$ such that

$$\forall q \in Q, g_{\text{out}}(\phi(g_{\text{in}}(q))) = f(q). \tag{16}$$

**Definition A.12.** Let $Q, F, Q_\Phi, F_\Phi$ be finite sets. Let $f$ be a map from $Q$ to $F$, and let $\Phi$ be a finite set of maps $\{\phi : Q_\Phi \to F_\Phi\}$. We say $\Phi$ has the *capacity* to implement $f$ if and only if there exists at least one element $\phi \in \Phi$ that implements $f$.

That is, a single model *implements* an algorithm $f$ if for every input to $f$, the model outputs the corresponding output. A class of models parametrized by $\phi \in \Phi$ has the *capacity* to implement an algorithm if there exists at least one $\phi$ that implements $f$.

**Theorem 3.1.** *Let $G = (V, E)$ be a directed acyclic graph with maximum degree $d$. For $S \subseteq V$, let $E'_S$ denote the marginal estimate over $S$. Let $\mathcal{S}_d$ denote the superset that contains all subsets $S \subseteq V$ of size at most $d$. Given $\{E'_S\}_{S \in \mathcal{S}_{d+2}}$, a stack of $L$ axial attention blocks has the capacity to recover $G$'s skeleton and v-structures in $O(N)$ width, and propagate orientations on paths of $O(L)$ length.*

*Proof.* We consider axial attention blocks with dot-product attention and omit layer normalization from our analysis, as is common in the Transformer universality literature Yun et al. (2020). Our inputs $X \in \mathbb{R}^{d \times R \times C}$ consist of $d$-dimension embeddings over $R$ rows and $C$ columns. Since our axial attention only operates over one dimension at a time, we use $X_{\cdot,c}$ to denote a 1D sequence of length $R$, given a fixed column $c$, and $X_{r,\cdot}$ to denote a 1D sequence of length $C$, given a fixed row $r$. A single axial attention layer (with one head) consists of two attention layers and a feedforward network,

$$\text{Attn}_{\text{row}}(X_{\cdot,c}) = X_{\cdot,c} + W_O W_V X_{\cdot,c} \cdot \sigma \left[ (W_K X_{\cdot,c})^T W_Q X_{\cdot,c} \right], \tag{17}$$
$$X \leftarrow \text{Attn}_{\text{row}}(X)$$
$$\text{Attn}_{\text{col}}(X_{r,\cdot}) = X_{r,\cdot} + W_O W_V X_{r,\cdot} \cdot \sigma \left[ (W_K X_{r,\cdot})^T W_Q X_{r,\cdot} \right], \tag{18}$$
$$X \leftarrow \text{Attn}_{\text{col}}(X)$$
$$\text{FFN}(X) = X + W_2 \cdot \text{ReLU}(W_1 \cdot X + b_1 \mathbf{1}_L^T) + b_2 \mathbf{1}_L^T, \tag{19}$$

where $W_O \in \mathbb{R}^{d \times d}, W_V, W_K, W_Q \in \mathbb{R}^{d \times d}, W_2 \in \mathbb{R}^{d \times m}, W_1 \in \mathbb{R}^{m \times d}, b_2 \in \mathbb{R}^d, b_1 \in \mathbb{R}^m$, and $m$ is the hidden layer size of the feedforward network. For concision, we have omitted the $r$ and $c$ subscripts on the $W$s, but the row and column attentions use different parameters. Any row or column attention can take on the identity mapping by setting $W_O, W_V, W_K, W_Q$ to $d \times d$ matrices of zeros.

A single axial attention *block* consists of two axial attention layers $\phi_E$ and $\phi_\rho$, connected via messages (Section 3.3)

$$h^{E,\ell} = \phi_{E,\ell}(h^{E,\ell-1})$$
$$h^{\rho,\ell-1} \leftarrow W_{\rho,\ell}\left[h^{\rho,\ell-1}, m^{E \to \rho,\ell}\right]$$
$$h^{\rho,\ell} = \phi_{\rho,\ell}(h^{\rho,\ell-1})$$
$$h^{E,\ell} \leftarrow h^{E,\ell} + m^{\rho \to E,\ell}$$

where $h^\ell$ denote the hidden representations of $E$ and $\rho$ at layer $\ell$, and the outputs of the axial attention block are $h^{\rho,\ell}, h^{E,\ell}$.

We construct a stack of $L \geq 3$ axial attention blocks that implement Algorithm 1.

**Model inputs**  Consider edge estimate $E'_{i,j} \in \mathcal{E}$ in a graph of size $N$. Let $e_i, e_j$ denote the endpoints of $(i,j)$. Outputs of the PC algorithm can be expressed by three endpoints: $\{\varnothing, \bullet, \blacktriangleright\}$. A directed edge from $i \to j$ has endpoints $(\bullet, \blacktriangleright)$, the reversed edge $i \leftarrow j$ has endpoints $(\blacktriangleright, \bullet)$, an undirected edge has endpoints $(\bullet, \bullet)$, and the lack of any edge between $i, j$ has endpoints $(\varnothing, \varnothing)$.

Let one-hot$_N(i)$ denote the $N$-dimensional one-hot column vector where element $i$ is 1. We define the embedding of $(i,j)$ as a $d = 2N + 6$ dimensional vector,

$$g_{\text{in}}(E_{t,(i,j)}) = h^{E,0}_{(i,j)} = \begin{bmatrix} \text{one-hot}_3(e_i) \\ \hline \text{one-hot}_3(e_j) \\ \hline \text{one-hot}_N(i) \\ \hline \text{one-hot}_N(j) \end{bmatrix}. \tag{20}$$

To recover graph structures from $h^E$, we simply read off the indices of non-zero entries ($g_{\text{out}}$). We can set $h^{\rho,0}$ to any $\mathbb{R}^{d \times N \times N}$ matrix, as we do not consider its values in this analysis and discard it during the first step.

**Claim A.13.** *(Consistency) The outputs of each step*

1. *are consistent with (20), and*
2. *are equivariant to the ordering of nodes in edges.*

For example, if $(i,j)$ is oriented as $(\blacktriangleright, \bullet)$, then we expect $(j,i)$ to be oriented $(\bullet, \blacktriangleright)$.

**Step 1: Undirected skeleton**  We use the first axial attention block to recover the undirected skeleton $C'$. We set all attentions to the identity, set $W_{\rho,1} \in \mathbb{R}^{2d \times d}$ to a $d \times d$ zeros matrix, stacked on top of a $d \times d$ identity matrix (discard $\rho$), and set $\text{FFN}_E$ to the identity (inputs are positive). This yields

$$h^{\rho,0}_{i,j} = m^{E \to \rho,1}_{i,j} = \begin{bmatrix} P_{e_i}(\varnothing) \\ P_{e_i}(\bullet) \\ P_{e_i}(\blacktriangleright) \\ \vdots \\ \hline \text{one-hot}_N(i) \\ \hline \text{one-hot}_N(j) \end{bmatrix}, \tag{21}$$

where $P_{e_i}(\cdot)$ is the frequency that endpoint $e_i = \cdot$ within the subsets sampled. FFNs with 1 hidden layer are universal approximators of continuous functions (Hornik et al., 1989), so we use $\text{FFN}_\rho$ to map

$$\text{FFN}_\rho(X_{i,u,v}) = \begin{cases} 0 & i \leq 6 \\ 0 & i > 6, X_{1,u,v} = 0 \\ -X_{i,u,v} & \text{otherwise,} \end{cases} \tag{22}$$

where $i \in [2N + 6]$ indexes into the feature dimension, and $u, v$ index into the rows and columns. This allows us to remove edges not present in $C'$ from consideration:

$$m^{\rho \to E, 1} = h^{\rho, 1}$$

$$h_{i,j}^{E,1} \leftarrow h_{i,j}^{E,1} + m_{i,j}^{\rho \to E, 1} = \begin{cases} 0 & (i, j) \notin C' \\ h_{i,j}^{E,0} & \text{otherwise.} \end{cases} \tag{23}$$

This yields $(i, j) \in C'$ if and only if $h_{i,j}^{\rho, 1} \neq \mathbf{0}$. We satisfy A.13 since our inputs are valid PC algorithm outputs for which $P_{e_i}(\varnothing) = P_{e_j}(\varnothing)$.

**Step 2: V-structures** The second and third axial attention blocks recover v-structures. We run the same procedure twice, once to capture v-structures that point towards the first node in an ordered pair, and one to capture v-structures that point towards the latter node.

We start with the first row attention over edge estimates, given a fixed subset $t$. We set the key and query attention matrices

$$W_K = k \cdot \begin{bmatrix} 0 & 0 & 1 & & & \\ & & & 0 & 1 & 0 \\ & \vdots & & & & \\ & & & & I_N & \\ & & & & & -I_N \end{bmatrix} \qquad W_Q = k \cdot \begin{bmatrix} 0 & 0 & 1 & & & \\ & & & 0 & 1 & 0 \\ & \vdots & & & & \\ & & & & I_N & \\ & & & & & I_N \end{bmatrix} \tag{24}$$

where $k$ is a large constant, $I_N$ denotes the size $N$ identity matrix, and all unmarked entries are 0s.

Recall that a v-structure is a pair of directed edges that share a target node. We claim that two edges $(i, j), (u, v)$ form a v-structure in $E'$, pointing towards $i = u$, if this inner product takes on the maximum value

$$\langle (W_K h^{E, 1})_{i,j}, (W_Q h^{E, 1})_{u,v} \rangle = 3. \tag{25}$$

Suppose both edges $(i, j)$ and $(u, v)$ still remain in $C'$. There are two components to consider.

1. If $i = u$, then their shared node contributes $+1$ to the inner product (prior to scaling by $k$). If $j = v$, then the inner product accrues $-1$.
2. Nodes that do not share the same endpoint contribute 0 to the inner product. Of edges that share one node, only endpoints that match ▶ at the starting node, or ● at the ending node contribute $+1$ to the inner product each. We provide some examples below.

| $(e_i, e_j)$ | $(e_u, e_v)$ | contribution | note |
|---|---|---|---|
| (▶, ●) | (●, ▶) | 0 | no shared node |
| (●, ▶) | (●, ▶) | 0 | wrong endpoints |
| (●, ●) | (●, ●) | 1 | one correct endpoint |
| (▶, ●) | (▶, ●) | 2 | v-structure |

All edges with endpoints $\varnothing$ were "removed" in step 1, resulting in an inner product of zero, since their node embeddings were set to zero. We set $k$ to some large constant (empirically, $k^2 = 1000$ is more than enough) to ensure that after softmax scaling, $\sigma_{e,e'} > 0$ only if $e, e'$ form a v-structure.

Given ordered pair $e = (i, j)$, let $V_i \subset V$ denote the set of nodes that form a v-structure with $e$ with shared node $i$. Note that $V_i$ excludes $j$ itself, since setting of $W_K, W_Q$ exclude edges that share both nodes. We set $W_V$ to the identity, and we multiply by attention weights $\sigma$ to obtain

$$(W_V h^{E, 1} \sigma)_{e=(i,j)} = \begin{bmatrix} \vdots \\ \hline \text{one-hot}_N(i) \\ \hline \alpha_j \cdot \text{binary}_N(V_j) \end{bmatrix} \tag{26}$$

where $\text{binary}_N(S)$ denotes the $N$-dimensional binary vector with ones at elements in $S$, and the scaling factor

$$\alpha_j = (1/\|V_j\|) \cdot \mathbb{1}\{\|V_j\| > 0\} \in [0, 1] \tag{27}$$

results from softmax normalization. We set

$$W_O = \begin{bmatrix} \mathbf{0}_{N+6} & \\ & 0.5 \cdot I_N \end{bmatrix} \tag{28}$$

to preserve the original endpoint values, and to distinguish between the edge's own node identity and newly recognized v-structures. To summarize, the output of this row attention layer is

$$\text{Attn}_{\text{row}}(X_{\cdot,c}) = X_{\cdot,c} + W_O W_V X_{\cdot,c} \cdot \sigma,$$

which is equal to its input $h^{E,1}$ plus additional positive values $\in (0, 0.5)$ in the last $N$ positions that indicate the presence of v-structures that exist in the overall $E'$.

Our final step is to "copy" newly assigned edge directions into all the edges. We set the $\phi_E$ column attention, $\text{FFN}_E$ and the $\phi_\rho$ attentions to the identity mapping. We also set $W_{\rho,2}$ to a $d \times d$ zeros matrix, stacked on top of a $d \times d$ identity matrix. This passes the output of the $\phi_E$ row attention, aggregated over subsets, directly to $\text{FFN}_{\phi,2}$.

For endpoint dimensions $\mathbf{e} = [6]$, we let $\text{FFN}_{\phi,2}$ implement

$$\text{FFN}_{\rho,2}(X_{\mathbf{e},u,v}) = \begin{cases} [0, 0, 1, 0, 1, 0]^T - X_{\mathbf{e},u,v} & 0 < \sum_{i>N+6} X_{i,u,v} < 0.5 \\ 0 & \text{otherwise.} \end{cases} \tag{29}$$

Subtracting $X_{\mathbf{e},u,v}$ "erases" the original endpoints and replaces them with $(\blacktriangleright, \bullet)$ after the update

$$h_{i,j}^{E,1} \leftarrow h_{i,j}^{E,1} + m_{i,j}^{\rho \to E,1}.$$

The overall operation translates to checking whether *any* v-structure points towards $i$, and if so, assigning edge directions accordingly. For dimensions $i > 6$,

$$\text{FFN}_{\rho,2}(X_{i,u,v}) = \begin{cases} -X_{i,u,v} & X_{i,u,v} \leq 0.5 \\ 0 & \text{otherwise,} \end{cases} \tag{30}$$

effectively erasing the stored v-structures from the representation and remaining consistent to (20).

At this point, we have copied all v-structures once. However, our orientations are not necessarily symmetric. For example, given v-structure $i \to j \leftarrow k$, our model orients edges $(j, i)$ and $(j, k)$, but not $(i, j)$ or $(k, j)$.

The simplest way to symmetrize these edges (for the writer and the reader) is to run another axial attention block, in which we focus on v-structures that point towards the second node of a pair. The only changes are as follows.

- For $W_K$ and $W_Q$, we swap columns 1-3 with 4-6, and columns 7 to $N + 6$ with the last $N$ columns.

- $(h^{E,2}\sigma)_{i,j}$ sees the third and fourth blocks swapped.

- $W_O$ swaps the $N \times N$ blocks that correspond to $i$ and $j$'s node embeddings.

- $\text{FFN}_{\rho,3}$ sets the endpoint embedding to $[0, 1, 0, 0, 0, 1]^T - X_{\mathbf{e},u,v}$ if $i = 7, ..., N + 6$ sum to a value between 0 and 0.5.

The result is $h^{E,3}$ with all v-structures oriented symmetrically, satisfying A.13.

**Step 3: Orientation propagation** To propagate orientations, we would like to identify cases $(i, j), (i, k) \in E', (j, k) \notin E'$ with shared node $i$ and corresponding endpoints $(\blacktriangleright, \bullet), (\bullet, \bullet)$. We use $\phi_E$ to identify triangles, and $\phi_\rho$ to identify edges $(i, j), (i, k) \in E'$ with the desired endpoints, while ignoring triangles.

**Marginal layer** The row attention in $\phi_E$ fixes a subset $t$ and varies the edge $(i,j)$.

Given edge $(i,j)$, we want to extract all $(i,k)$ that share node $i$. We set the key and query attention matrices to

$$W_K, W_Q = k \cdot \begin{bmatrix} 0 & 1 & 1 & 0 & 1 & 1 \\ \vdots & & & & & \\ & & & & I_N & \\ & & & & & \pm I_N \end{bmatrix}. \tag{31}$$

We set $W_V$ to the identity to obtain

$$(W_V h^E \sigma)_{e=(i,k)} = \begin{bmatrix} \vdots \\ \hline \vdots \\ \hline \text{one-hot}_N(i) \\ \alpha_k \cdot \text{binary}_N(V_k) \end{bmatrix}, \tag{32}$$

where $V_k$ is the set of nodes $k$ that share any edge with $i$. To distinguish between $k$ and $V_k$, we again set $W_o$ to the same as in (28). Finally, we set $\text{FFN}_E$ to the identity and pass $h^E$ directly to $\phi_\rho$. To summarize, we have $h^E$ equal to its input, with values $\in (0, 0.5)$ in the last $N$ locations indicating 1-hop neighbors of each edge.

**Global layer** Now we would like to identify cases $(i,k), (j,k)$ with corresponding endpoints $(\bullet, \blacktriangleright), (\bullet, \bullet)$. We set the key and query attention matrices

$$W_K = k \cdot \begin{bmatrix} 0 & 0 & 1 \\ \vdots & & \\ & & I_N \\ & & & I_N \end{bmatrix} \qquad W_Q = k \cdot \begin{bmatrix} 0 & 1 & -1 & 0 & 1 & -1 \\ \vdots & & & & & \\ & & & & I_N & \\ & & & & & -I_N \end{bmatrix}. \tag{33}$$

The key allows us to check that endpoint $i$ is directed, and the query allows us to check that $(i,k)$ exists in $C'$, and does not already point elsewhere. After softmax normalization, for sufficiently large $k$, we obtain $\sigma_{(i,j),(i,k)} > 0$ if and only if $(i,k)$ should be oriented $(\bullet, \blacktriangleright)$, and the inner product attains the maximum possible value

$$\langle (W_K h^\rho)_{i,j}, (W_Q h^\rho)_{i,k} \rangle = 2. \tag{34}$$

We consider two components.

1. If the endpoints match our desired endpoints, we gain a $+1$ contribution to the inner product.
2. A match between the first nodes contributes $+1$. If the second node shares any overlap (either same edge, or a triangle), then a negative value would be added to the overall inner product.

Therefore, we can only attain the maximal inner product if only one edge is directed, and if there exists no triangle.

We set $W_o$ to the same as in (28), and we add $h^\rho$ to the input of the next $\phi_E$. To summarize, we have $h^\rho$ equal to its input, with values $\in (0, 0.5)$ in the last $N$ locations indicating incoming edges.

**Orientation assignment** Our final step is to assign our new edge orientations. Let the column attention take on the identity mapping. For endpoint dimensions $\mathbf{e} = (4, 5, 6)$, we let $\text{FFN}_\rho$ implement

$$\text{FFN}_\rho(X_{\mathbf{e},u,v}) = \begin{cases} [0,0,1]^T - X_{\mathbf{e},u,v} & 0 < \sum_{i>N+6} X_{i,u,v} < 0.5 \\ 0 & \text{otherwise.} \end{cases} \tag{35}$$

This translates to checking whether any incoming edge points towards $v$, and if so, assigning the new edge direction accordingly. For dimensions $i > 6$,

$$\text{FFN}_\rho(X_{i,u,v}) = \begin{cases} 0 & X_{i,u,v} \le 0.5 \\ X_{i,u,v} & \text{otherwise,} \end{cases} \tag{36}$$

effectively erasing the stored assignments from the representation. Thus, we are left with $h^{E,\ell}$ that conforms to the same format as the initial embedding in (20).

To symmetrize these edges, we run another axial attention block, in which we focus on paths that point towards the second node of a pair. The only changes are as follows.

- For $\phi_E$ layer $W_K$ and $W_Q$ (31), we swap $I_N$ and $\pm I_N$.

- For $\phi_\rho$ layer $W_K$ and $W_Q$ (33), we swap $I_N$ and $\pm I_N$.

- $W_O$ swaps the $N \times N$ blocks that correspond to $i$ and $j$'s node embeddings.

- For FFN$_\rho$ (35), we let $\mathbf{e} = (1, 2, 3)$ instead.

The result is $h^E$ with symmetric 1-hop orientation propagation, satisfying A.13. We may repeat this procedure $k$ times to capture $k$-hop paths.

To summarize, we used axial attention block 1 to recover the undirected skeleton $C'$, blocks 2-3 to identify and copy v-structures in $E'$, and all subsequent $L - 3$ layers to propagate orientations on paths up to $\lfloor (L-3)/2 \rfloor$ length. Overall, this particular construction requires $O(N)$ width for $O(L)$ paths.

$\square$

**Final remarks**  Information theoretically, it should be possible to encode the same information in $\log N$ space, and achieve $O(\log N)$ width. For ease of construction, we have allowed for wider networks than optimal. On the other hand, if we increase the width and encode each edge symmetrically, e.g. $(e_i, e_j, e_j, e_i \mid i, j, j, i)$, we can reduce the number of blocks by half, since we no longer need to run each operation twice. However, attention weights scale quadratically, so we opted for an asymmetric construction.

Finally, a strict limitation of our model is that it only considers 1D pairwise interactions. In the graph layer, we cannot compare different edges' estimates at different times in a single step. In the feature layer, we cannot compare $(i, j)$ to $(j, i)$ in a single step either. However, the graph layer does enable us to compare all edges at once (sparsely), and the feature layer looks at a time-collapsed version of the whole graph. Therefore, though we opted for this design for computational efficiency, we have shown that it is able to capture significant graph reasoning.

### A.4   Robustness and stability

We discuss the notion of stability informally, in the context of Spirtes et al. (2001). There are two cases in which our framework may receive erroneous inputs: low/noisy data settings, and functionally misspecified situations. We consider our framework's empirical robustness to these cases, in terms of recovering the skeleton and orienting edges.

In the case of noisy data, edges may be erroneously added, removed, or misdirected from marginal estimates $E'$. Our framework provides two avenues to mitigating such noise.

1. We observe that global statistics can be estimated reliably in low data scenarios. For example, Figure 4 suggests that 300 examples suffice to provide a robust estimate over 100 variables in our synthetic settings. Therefore, even if the marginal estimates are erroneous, the neural network can learn the skeleton from the global statistics.
2. Most classical causal discovery algorithms are not stable with respect to edge orientation assignment. That is, an error in a single edge may propagate throughout the graph. Empirically, we observe that the majority vote of GIES achieves reasonable accuracy even without any training, while FCI suffers in this assessment (Table 10). However both SEA (GIES) and SEA (FCI) achieve high edge accuracy. Therefore, while the underlying algorithms may not be stable with respect to edge orientation, our pretrained aggregator seems to be robust.

It is also possible that our global statistics and marginal estimates make misspecified assumptions regarding the data generating mechanisms. The degree of misspecification can vary case by case, so it is hard to

provide any broad guarantees about the performance of our algorithm, in general. However, we can make the following observation.

If two variables are independent, $X_i \perp\!\!\!\perp X_j$, they are independent, e.g. under linear Gaussian assumptions. If $X_i, X_j$ exhibit more complex functional dependencies, they may be erroneously deemed independent. Therefore, any systematic errors are necessarily one-sided, and the model can learn to recover the connectivity based on global statistics.

## B  Experimental details

### B.1  Synthetic data generation

Synthetic datasets were generated using code from DcDI (Brouillard et al., 2020), which extended the Causal Discovery Toolkit data generators to interventional data (Kalainathan et al., 2020).

We considered the following causal mechanisms. Let $y$ be the node in question, let $X$ be its parents, let $E$ be an independent noise variable (details below), and let $W$ be randomly initialized weight matrices.

- Linear: $y = XW + E$.

- Polynomial: $y = W_0 + XW_1 + X^2 W_2 + \times E$

- Sigmoid: $y = \sum_{i=1}^{d} W_i \cdot \text{sigmoid}(X_i) + \times E$

- Randomly initialized neural network (NN): $y = \text{Tanh}((X, E)W_{\text{in}})W_{\text{out}}$

- Randomly initialized neural network, additive (NN additive): $y = \text{Tanh}(XW_{\text{in}})W_{\text{out}} + E$

Root causal mechanisms, noise variables, and interventional distributions maintained the DcDI defaults.

- Root causal mechanisms were set to $\text{Uniform}(-2, 2)$.

- Noise was set to $E \sim 0.4 \cdot \mathcal{N}(0, \sigma^2)$ where $\sigma^2 \sim \text{Uniform}(1, 2)$.

- Interventions were applied to all nodes (one at a time) by setting their causal mechanisms to $\mathcal{N}(0, 1)$.

Ablation datasets with $N > 100$ nodes contained 100,000 points each (same as $N = 100$). We set random seeds for each dataset using the hash of the output filename.

### B.2  Related work and baselines

We considered the following baselines. All baselines were run using official implementations published by the authors.

**AVICI** (Lorch et al., 2022) is the most similar method to this work, though there are significant differences in both the causal discovery strategy and the implementation. Both works simulate diverse datasets for training, which differ in graph topology, causal mechanism, and type of exogenous noise; and both are attention-based architectures (Vaswani et al., 2017). The primary difference is that AVICI operates over raw data, while we operate over summary statistics. While access to raw data may allow for richer modeling of relationships, it also increases the computational cost on large datasets. In fact, AVICI's model complexity scales quadratically as the number of samples *and* quadratically as the number of nodes. Our model does not explicitly depend on the number of data samples, but scales cubically as the number of nodes, as we attend over richer, pairwise features.

Both AVICI and SEA are equivariant to the ordering of nodes. AVICI is additionally invariant to the ordering of samples. Due to the two-track design of our aggregator, we also include message passing operations between the two input types (global statistics, marginal features), while AVICI directly stacks Transformer layers.

Finally, AVICI explicitly regularizes for acyclicity. We follow the ENCO (Lippe et al., 2022) formulation (edges $i \to j$ and $j \to i$ cannot co-exist), since it is easier to optimize. Empirically, our predictions are still 99% acyclic (Table 18).

AVICI was run on all test datasets using the authors' pretrained `scm-v0` model, recommended for "arbitrary real-valued data." Note that this model is different from the models described in their paper (denoted AVICI-L and AVICI-R), as it was trained on *all* of their synthetic data, including test sets. We sampled 1000 observations per dataset uniformly at random, with their respective interventions (the maximum number of synthetic samples used in their original paper), except for Sachs, which used the entire dataset (as in their paper). Though the authors provided separate weights for synthetic mRNA data, we were unable to use it since we did not preserve the raw gene counts in our simulated mRNA datasets.

**Dcdi** (Brouillard et al., 2020) was trained on each of the $N = 10, 20$ datasets using their published hyperparameters. We denote the Gaussian and Deep Sigmoidal Flow versions as DCDI-G and DCDI-DSF respectively. DCDI could not scale to graphs with $N = 100$ due to memory constraints (did not fit on a 32GB V100 GPU).

**DCD-FG** (Lopez et al., 2022) was trained on all of the test datasets using their published hyperparameters. We set the number of factors to $5, 10, 20$ for each of $N = 10, 20, 100$, based on their ablation studies. Due to numerical instability on $N = 100$, we clamped augmented Lagrangian multipliers $\mu$ and $\gamma$ to 10 and stopped training if elements of the learned adjacency matrix reached `NaN` values. After discussion with the authors, we also tried adjusting the $\mu$ multiplier from 2 to 1.1, but the model did not converge within 48 hours.

**DECI** (Geffner et al., 2024) was trained on all of the test datasets using their published hyperparameters. However, on all $N = 100$ cases, the model failed to produce any meaningful results (adjacency matrices nearly all remained 0s with AUCs of 0.5). Thus, we only report results on $N = 10, 20$.

**DiffAN** (Sanchez et al., 2023) was trained on the each of the $N = 10, 20$ datasets using their published hyperparameters. The authors write that "most hyperparameters are hard-coded into [the] constructor of the DIFFAN class and we verified they work across a wide set of datasets." We used the original, non-approximation version of their algorithm by maintaining `residue=True` in their codebase. We were unable to consistently run DIFFAN with both R and GPU support within a Docker container, and the authors did not respond to questions regarding reproducibility, so all models were trained on the CPU only. We observed approximately a 10x speedup in the $< 5$ cases that were able to complete running on the GPU.

**InvCov** computes inverse covariance over 1000 examples. This does *not* orient edges, but it is a strong connectivity baseline. We discretize based on ground truth positive rate.

**Corr** and **D-Corr** are computed similarly, using global correlation and distance correlation, respectively (See C.1 for details).

### B.3 Metrics

Throughout our evaluation, we compute metrics with respect to the ground truth graph. This means that in the observational setting, the "oracle" value of each metric will vary depending on the size of the equivalence class (e.g. if multiple graphs are observationally equivalent, the expected SHD is $> 0$; see Section 5.3 for more analysis). While it is more correct to evaluate (observational) models with respect to the CPDAG that is implied by the predicted DAG, there were several reasons we also chose to evaluate with respect to the ground truth DAG.

- This practice is quite common among continuous causal discovery algorithms, regardless of whether interventional data are available (Lorch et al., 2022; Brouillard et al., 2020; Hägele et al., 2023) or not (Sanchez et al., 2023). This may be primarily because their predicted graphs contain continuous probabilities, and metrics that reflect uncertainty (mAP, AUROC) are difficult to translate to the equivalence class.

- It has been reported that common simulators produce "identifiable" linear Gaussian datasets, if the data are unstandardized (hence the VarSort baseline, Reisach et al. (2021)). This would mean

that the MEC is actually smaller than expected. While it is possible to normalize all the data, it is unknown whether there are additional artifacts of the data generating process that may influence the (empirical) identifiability of the graphs, and thus, the "true" size of the equivalence class.

- Our models and baselines are evaluated on the same graphs, but not all can incorporate interventional information. For comparability, we evaluated all predictions against the same ground truth, while noting that the equivalence class is larger in the observational case.

- FCI, GIES, and VARSORT don't necessarily produce acyclic graphs, due to bootsrapping. In Appendix C, we also include the symmetric summary statistics as baselines, to quantify the "cost" of not directing edges, and of not using marginal estimates. In these cases, we cannot compute a corresponding CPDAG.

- Practically, it was somewhat expensive to compute the CPDAG, especially for larger dense graphs ($N = 100, E = 400$ and beyond).

## B.4 Training and hardware details

Hyperparameters and architectural choices were selected by training the model on 20% of the the training and validation data for approximately 50k steps (several hours). We considered the following parameters in sequence.

- learned positional embedding vs. sinusoidal positional embedding

- number of layers $\times$ number of heads: $\{4, 8\} \times \{4, 8\}$

- learning rate $\eta = \{1e - 4, 5e - 5, 1e - 5\}$

For our final model, we selected learned positional embeddings, 4 layers, 8 heads, and learning rate $\eta = 1e - 4$.

The models were trained across 2 NVIDIA RTX A6000 GPUs and 60 CPU cores. We used the GPU exclusively for running the aggregator, and retained all classical algorithm execution on the CPUs (during data loading). The total pretraining time took approximately 14 hours for the final FCI model and 16 hours for the final GIES model.

For finetuning, we used rank $r = 2$ adapters on the axial attention model's key, query, and feedforward weights (Hu et al., 2022). We trained until convergence on the validation set (no improvement for 100 epochs), which took 4-6 hours with 40 CPUs and around 10 hours with 20 CPUs. We used a single NVIDIA RTX A6000 GPU, but the bottleneck was CPU availability.

For the scope of this paper, our models and datasets are fairly small. We did not scale further due to hardware constraints. Our primary bottlenecks to scaling up lay in availability of CPU cores and networking speed across nodes, rather than GPU memory or utilization. The optimal CPU:GPU ratio for SEA ranges from 20:1 to 40:1.

We are able to run inference comfortably over $N = 500$ graphs with $T = 500$ subsets of $k = 5$ nodes each, on a single 32GB V100 GPU. For runtime analysis, we used a batch size of 1, with 1 data worker per dataset. Our runtime could be further improved if we amortized the GPU utilization across batches.

## B.5 Choice of classical causal discovery algorithm

For training, we selected FCI (Spirtes et al., 1995) as the underlying discovery algorithm in the observational setting over GES (Chickering, 2002), GRaSP (Lam et al., 2022), and LiNGAM (Shimizu et al., 2006) due to its speed and superior downstream performance. We hypothesize this may be due to its richer output (ancestral graph) providing more signal to the Transformer model. We also tried Causal Additive Models (Bühlmann et al., 2014), but its runtime was too slow for consistent GPU utilization. Observational algorithm implementations were provided by the causal-learn library (Zheng et al., 2024). The code for running these alternative classical algorithms is available in our codebase.

We selected GIES as the discovery algorithm in the interventional setting because an efficient Python implementation was readily available at `https://github.com/juangamella/gies`.

We tried incorporating implementations from the Causal Discovery Toolbox via a Docker image (Kalainathan et al., 2020), but there was excessive overhead associated with calling an R subroutine and reading/writing the inputs/results from disk.

Finally, we considered other independence tests for richer characterization, such as kernel-based methods. However, due to speed, we chose to remain with the default Fisherz conditional independence test for FCI, and BIC for GIES (Schwarz, 1978).

### B.6 Sampling procedure

**Selection scores:** We consider three strategies for computing selection scores $\alpha$. We include an empirical comparison of these strategies in Table 9.

1. Random selection: $\alpha$ is an $N \times N$ matrix of ones.
2. Global-statistic-based selection: $\alpha = \rho$.
3. Uncertainty-based selection: $\alpha = \hat{H}(E_t)$, where $H$ denotes the information entropy

$$\alpha_{i,j} = - \sum_{e \in \{0,1,2\}} p(e) \log p(e). \tag{37}$$

Let $c_{i,j}^t$ be the number of times edge $(i,j)$ was selected in $S_1 \ldots S_{t-1}$, and let $\alpha^t = \alpha / \sqrt{c_{i,j}^t}$. We consider two strategies for selecting $S_t$ based on $\alpha_t$.

**Greedy selection:** Throughout our experiments, we used a greedy algorithm for subset selection. We normalize probabilities to 1 before constructing each Categorical. Initialize

$$S_t \leftarrow \{i : i \sim \text{Categorical}(\alpha_1^t \ldots \alpha_N^t)\}. \tag{38}$$

where $\alpha_i^t = \sum_{j \neq i \in V} \alpha_{i,j}^t$. While $|S_t| < k$, update

$$S_t \leftarrow S_t \cup \{j : j \sim \text{Categorical}(\alpha_{1,S_t}^t \ldots \alpha_{N,S_t}^t)) \tag{39}$$

where

$$\alpha_{j,S_t} = \begin{cases} \sum_{i \in S_t} \alpha_{i,j}^t & j \notin S_t \\ 0 & \text{otherwise.} \end{cases} \tag{40}$$

**Subset selection:** We also considered the following subset-level selection procedure, and observed minor performance gain for significantly longer runtime (linear program takes around 1 second per batch). Therefore, we opted for the greedy method instead.

We solve the following integer linear program to select a subset $S_t$ of size $k$ that maximizes $\sum_{i \in S_t} \alpha_{i,j}^t$. Let $\nu_i \in \{0,1\}$ denote the selection of node $i$, and let $\epsilon_{i,j} \in \{0,1\}$ denote the selection of edge $(i,j)$. Our objective is to

$$
\begin{aligned}
\text{maximize} \quad & \sum_{i,j} a_{i,j}^t \cdot \epsilon_{i,j} \\
\text{subject to} \quad & \sum_i \nu_i = k && \text{subset size} \\
& \epsilon_{i,j} \geq \nu_i + \nu_j - 1 && \text{node-edge consistency} \\
& \epsilon_{i,j} \leq \nu_i \\
& \epsilon_{i,j} \leq \nu_j, \\
& \nu_i \in \{0,1\} \\
& \epsilon_{i,j} \in \{0,1\}
\end{aligned}
$$

for $i, j \in V \times V$, $i \in V$. $S_t$ is the set of non-zero indices in $\nu$.

The final algorithm used the greedy selection strategy, with the first half of batches sampled according to global statistics, and the latter half sampled randomly, with visit counts shared. This strategy was selected

Table 9: Comparison between heuristics-based sampler (random and inverse covariance) vs. model confidence-based sampler. The suffix -L indicates the greedy confidence-based sampler. Each setting encompasses 5 distinct Erdős-Rényi graphs. The symbol † indicates that SEA was not pretrained on this setting. Bold indicates best of all models considered (including baselines not pictured).

| N E | Model | Linear | | | NN add. | | | NN non-add. | | | Sigmoid† | | | Polynomial† | | |
|---|---|---|---|---|---|---|---|---|---|---|---|---|---|---|---|---|
| | | mAP↑ | OA↑ | shd↓ | mAP↑ | OA↑ | shd↓ | mAP↑ | OA↑ | shd↓ | mAP↑ | OA↑ | shd↓ | mAP↑ | OA↑ | shd↓ |
| 10 10 | SEA-F | 0.97 | 0.92 | 1.6 | **0.95** | **0.92** | 2.4 | **0.92** | 0.94 | 2.8 | 0.83 | 0.76 | 3.7 | 0.69 | 0.71 | 6.7 |
| | SEA-G | **0.99** | **0.94** | 1.2 | 0.94 | 0.88 | 2.6 | 0.91 | 0.93 | 3.2 | 0.85 | 0.84 | 4.0 | 0.70 | 0.79 | **5.8** |
| | SEA-F-L | 0.97 | 0.93 | **1.0** | 0.95 | 0.87 | 2.4 | **0.92** | **0.98** | 3.4 | 0.84 | 0.77 | 3.9 | **0.70** | 0.79 | 5.8 |
| | SEA-G-L | 0.98 | 0.93 | 1.4 | 0.94 | 0.91 | 2.8 | 0.91 | 0.94 | 4.0 | **0.88** | **0.84** | **3.6** | 0.70 | **0.80** | 5.8 |
| 10 40 | SEA-F | 0.90 | 0.87 | 14.4 | 0.91 | 0.94 | 11.2 | 0.87 | 0.86 | 16.0 | **0.81** | 0.85 | 22.7 | 0.81 | 0.92 | 33.4 |
| | SEA-G | **0.94** | **0.91** | **12.8** | 0.91 | **0.95** | 10.4 | **0.89** | **0.89** | 17.2 | 0.81 | **0.87** | 24.5 | 0.89 | 0.93 | 29.5 |
| | SEA-F-L | 0.91 | 0.90 | 15.6 | **0.91** | 0.92 | 15.8 | 0.88 | 0.86 | 14.2 | 0.81 | 0.84 | 23.2 | 0.82 | 0.93 | 33.8 |
| | SEA-G-L | 0.93 | 0.91 | 13.4 | 0.91 | 0.93 | **10.4** | 0.88 | 0.85 | 16.2 | 0.79 | 0.83 | 25.5 | **0.90** | **0.94** | 28.3 |
| 20 20 | SEA-F | 0.97 | 0.92 | 3.2 | 0.94 | 0.97 | 3.2 | 0.84 | 0.93 | 7.2 | 0.84 | 0.85 | 7.6 | **0.71** | **0.80** | 10.2 |
| | SEA-G | 0.97 | 0.89 | 3.0 | **0.94** | 0.95 | 3.4 | 0.83 | 0.94 | 7.8 | 0.84 | 0.83 | 8.1 | 0.69 | 0.78 | 10.1 |
| | SEA-F-L | 0.97 | **0.92** | 2.8 | 0.93 | 0.95 | 3.8 | **0.85** | 0.94 | 6.8 | **0.85** | 0.85 | **7.5** | 0.67 | 0.78 | **9.9** |
| | SEA-G-L | **0.97** | 0.90 | **2.6** | **0.94** | **0.98** | 3.4 | 0.83 | **0.97** | 7.0 | 0.84 | 0.84 | 7.9 | 0.67 | 0.79 | 10.6 |
| 20 80 | SEA-F | 0.86 | **0.93** | 29.6 | 0.55 | 0.90 | 73.6 | 0.72 | **0.93** | 51.8 | **0.77** | 0.85 | 42.8 | 0.61 | 0.89 | 61.8 |
| | SEA-G | **0.89** | 0.92 | **26.8** | 0.58 | 0.88 | 71.4 | 0.73 | 0.92 | 50.6 | 0.76 | 0.84 | 45.0 | **0.65** | **0.89** | 60.1 |
| | SEA-F-L | 0.86 | 0.92 | 32.0 | 0.55 | **0.90** | 74.0 | 0.74 | 0.93 | 49.2 | 0.76 | **0.87** | **41.8** | 0.59 | 0.88 | 62.3 |
| | SEA-G-L | **0.89** | 0.92 | 28.4 | 0.58 | 0.89 | 71.6 | 0.75 | 0.92 | 49.4 | 0.75 | 0.85 | 45.7 | **0.65** | 0.88 | 60.6 |

heuristically, and we did not observe significant improvements or drops in performance when switching to other strategies (e.g. all greedy statistics-based, greedy uncertainty-based, linear program uncertainty-based, etc.)

Table 9 compares the heuristics-based greedy sampler (inverse covariance + random) with the model uncertainty-based greedy sampler. Runtimes are plotted in Figure 6. The latter was run on CPU only, since it was non-trivial to access the GPU within a PyTorch data loader. We ran a forward pass to obtain an updated selection score every 10 batches, so this accrued over 10 times the number of forward passes, all on CPU. With proper engineering, this model-based sampler is expected to be much more efficient than reported. Still, it is faster than nearly all baselines.

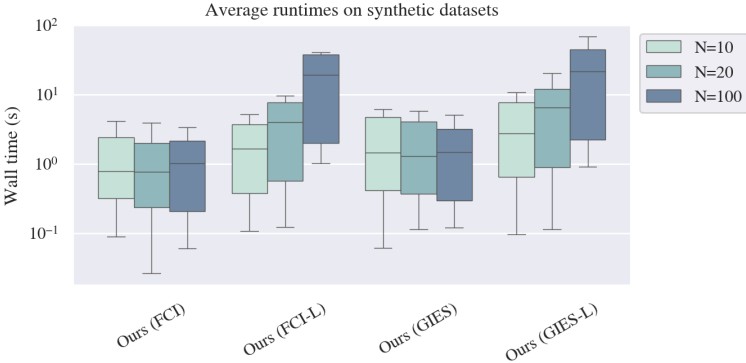

Figure 6: Runtime for heuristics-based greedy sampler vs. model uncertainty-based greedy sampler (suffix -L). For sampling, the model was run on CPU only, due to the difficulty of invoking GPU in the PyTorch data sampler.

## C   Additional analyses

### C.1   Choice of global statistic

We selected inverse covariance as our global feature due to its ease of computation and its relationship to partial correlation. For context, we also provide the performance analysis of several alternatives. Tables 11 and 12 compare the results of different graph-level statistics on our synthetic datasets. Discretization thresholds for SHD were obtained by computing the $p^{\text{th}}$ quantile of the computed values, where $p = 1 - (E/N)$. This is not entirely fair, as no other baseline receives the same calibration, but these ablation studies only seek to compare state-of-the-art causal discovery methods with the "best" possible (oracle) statistical alternatives.

**Corr** refers to global correlation,

$$\rho_{i,j} = \frac{\mathbb{E}\left(X_i X_j\right) - \mathbb{E}\left(X_i\right)\mathbb{E}\left(X_j\right)}{\sqrt{\mathbb{E}\left(X_i^2\right) - \mathbb{E}\left(X_i\right)^2} \cdot \sqrt{\mathbb{E}\left(X_j^2\right) - \mathbb{E}\left(X_j\right)^2}}. \tag{41}$$

**D-Corr** refers to distance correlation, computed between all pairs of variables. Distance correlation captures both linear and non-linear dependencies, and D-CORR$(X_i, X_j) = 0$ if and only if $X_i \perp\!\!\!\perp X_j$. Please refer to Sz'ekely et al. (2007) for the full derivation. Despite its power to capture non-linear dependencies, we opted not to use D-CORR because it is quite slow to compute between all pairs of variables.

**InvCov** refers to inverse covariance, computed globally,

$$\rho = \mathbb{E}\left((X - \mathbb{E}\left(X\right))(X - \mathbb{E}\left(X\right))^T\right)^{-1}. \tag{42}$$

For graphs $N < 100$, inverse covariance was computed directly using NumPy. For graphs $N \geq 100$, inverse covariance was computed using Ledoit-Wolf shrinkage at inference time Ledoit & Wolf (2004). Unfortunately we only realized this after training our models, so swapping to Ledoit-Wolf leads to some distribution shift (and drop in performance) on SEA results for large graphs.

### C.2   Visualization of denoising statistical features

Figure 7 illustrates the input features and our predictions for a $N = 10, E = 10$ linear graph. Compared to the inputs, our method is able to produce a much cleaner graph. GIES may orient edges the wrong direction in some of the bootstrap samples, but SEA can generally identify the right direction. This example also illustrates how SEA can triangulate between GIES and the global statistic, to avoid naive predictions of edges wherever the global statistic has a high value. In the example marked in teal, though the global statistic has a moderate value, the edge is absent from a substantial number of estimates (if we consider both directions). This reflects the theory in A.2 that edges absent from some marginal estimate should be absent from the final skeleton. SEA was able to (rather confidently) reject the edge in the final graph.

In contrast to GIES, FCI tends to identify much fewer edges (Figure 8). Note that this does not hurt performance on metrics, as continuous metrics consider "sliding" thresholds, while discrete metrics are computed with respect to the true edge rate (for the oracle FCI baseline). However, it does highlight that FCI has a much lower signal-to-noise ratio when used alone.

### C.3   Results on simulated mRNA data

We generated mRNA data using the SERGIO simulator Dibaeinia & Sinha (2020). We sampled datasets with the Hill coefficient set to $\{0.25, 0.5, 1, 2, 4\}$ for training, and 2 for testing (2 was default). We set the decay rate to the default 0.8, and the noise parameter to the default of 1.0. We sampled 400 graphs for each of $N = \{10, 20\}$ and $E = \{N, 2N\}$.

These data distributions are quite different from typical synthetic datasets, as they simulate steady-state measurements and the data are lower bounded at 0 (gene counts). Thus, we trained a separate model on these data using the SEA (FCI) architecture. Table 13 shows that SEA performs best across the board.

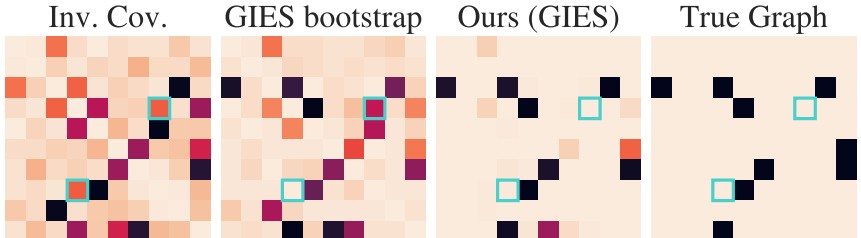

Figure 7: Left to right: magnitude value of global features with diagonals zeroed; GIES run over all variables via non-parametric bootstrapping (frequency of edge); SEA predictions; and the ground truth. SEA is able to denoise the input features much better than naive aggregation schemes.

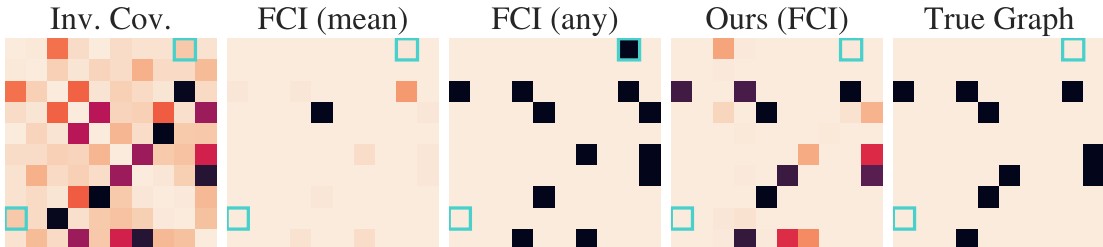

Figure 8: Left to right: magnitude value of global features with diagonals zeroed; FCI run over all variables via non-parametric bootstrapping, aggregation via frequency of edge vs. binarized (appear or not); SEA predictions; and the ground truth. FCI produces sparser outputs than GIES, so FCI alone is less able to distinguish between noisy and genuine edges.

## C.4 Results and ablation studies on synthetic data

For completeness, we include additional results and analysis on the synthetic datasets. Table 14 contains our evaluations in the observational setting, with metrics computed over the inferred CPDAG, rather than the predicted DAG. We find that the SHD varies slightly when evaluating DAGS or converted CPDAGs, but the difference is minimal, compared to the overall gaps in performance between methods.

Tables 19 and 20 compare all baselines across all metrics and graph sizes on Erdős-Rényi graphs. Tables 21 and 22 include the same evaluation on scale-free graphs. Tables 23 and 24 assess $N = 100$ graphs.

Table 17 ablates the contribution of the global and marginal features by setting their hidden representations to zero. Note that our model has never seen this type of input during training, so drops in performance may be conflated with input distributional shift. Overall, removing the joint statistics ($h^\rho \leftarrow 0$) leads to a higher performance drop than removing the marginal estimates ($h^E \leftarrow 0$). However, the gap between these ablation studies and our final performance may be quite large in some cases, so both inputs are important to the prediction.

Table 18 shows that despite omitting the DAG constraint, we find that our predicted graphs (test split) are nearly all acyclic, with a naive discretization threshold of 0.5. Unlike Lippe et al. (2022), which also omits the acyclicity constraint during training but optionally enforces it at inference time, we do not require any post-processing to achieve high performance. Empirically, we found existing DAG constraints to be unstable (Lagrangian) and slow to optimize (Zheng et al., 2018; Brouillard et al., 2020). DAG behavior would not emerge until late in training, when the regularization term is of 1e-8 scale or smaller.

Alternatively, we could quantify the raw information content provided by these two features through the INVCOV, FCI*, and GIES* baselines (Tables 19, 20, 21, 22). Overall, INVCOV and FCI* are comparable to worse-performing baselines. GIES* performs very well, sometimes approaching the strongest baselines.

Table 10: Synthetic experiments, edge direction accuracy (higher is better). All standard deviations were within 0.2. The symbol † indicates that Sea was not pretrained on this setting.

| $N$ | $E$ | Model | Linear | NN add | NN | Sig.† | Poly.† |
|---|---|---|---|---|---|---|---|
| 10 | 10 | Dcdi-G | 0.74 | 0.80 | 0.85 | 0.41 | 0.44 |
| | | Dcdi-Dsf | 0.79 | 0.62 | 0.68 | 0.38 | 0.39 |
| | | Dcd-Fg | 0.50 | 0.47 | 0.70 | 0.43 | 0.54 |
| | | DiffAn | 0.61 | 0.55 | 0.26 | 0.53 | 0.47 |
| | | Deci | 0.50 | 0.43 | 0.62 | 0.63 | 0.75 |
| | | Avici | 0.80 | **0.92** | 0.83 | 0.81 | 0.75 |
| | | Fci* | 0.52 | 0.43 | 0.41 | 0.55 | 0.40 |
| | | Gies* | 0.76 | 0.49 | 0.69 | 0.67 | 0.63 |
| | | Sea (Fci) | 0.92 | **0.92** | **0.94** | 0.76 | 0.71 |
| | | Sea (Gies) | **0.94** | 0.88 | 0.93 | **0.84** | **0.79** |
| 20 | 80 | Dcdi-G | 0.47 | 0.43 | 0.82 | 0.40 | 0.24 |
| | | Dcdi-Dsf | 0.50 | 0.49 | 0.78 | 0.41 | 0.28 |
| | | Dcd-Fg | 0.58 | 0.65 | 0.75 | 0.62 | 0.48 |
| | | DiffAn | 0.46 | 0.28 | 0.36 | 0.45 | 0.21 |
| | | Deci | 0.30 | 0.47 | 0.35 | 0.48 | 0.57 |
| | | Avici | 0.57 | 0.67 | 0.74 | 0.63 | 0.62 |
| | | Fci* | 0.19 | 0.19 | 0.22 | 0.33 | 0.23 |
| | | Gies* | 0.56 | 0.73 | 0.59 | 0.62 | 0.61 |
| | | Sea (Fci) | **0.93** | **0.90** | **0.93** | **0.85** | **0.89** |
| | | Sea (Gies) | 0.92 | 0.88 | 0.92 | 0.84 | **0.89** |
| 100 | 400 | Dcd-Fg | 0.46 | 0.60 | 0.70 | 0.67 | 0.53 |
| | | Avici | 0.61 | 0.68 | 0.72 | 0.54 | 0.42 |
| | | Sea (Fci) | 0.93 | 0.90 | 0.91 | **0.87** | 0.82 |
| | | Sea (Gies) | **0.94** | **0.91** | **0.92** | **0.87** | **0.84** |

Table 11: Comparison of global statistics (continuous metrics). All standard deviations within 0.1.

| $N$ | $E$ | Model | Linear | | NN add. | | NN non-add. | | Sigmoid | | Polynomial | |
|---|---|---|---|---|---|---|---|---|---|---|---|---|
| | | | mAP ↑ | AUC ↑ | mAP ↑ | AUC ↑ | mAP ↑ | AUC ↑ | mAP ↑ | AUC ↑ | mAP ↑ | AUC ↑ |
| 10 | 10 | Corr | 0.45 | 0.87 | 0.41 | 0.86 | 0.41 | 0.85 | 0.46 | 0.86 | 0.45 | 0.85 |
| | | D-Corr | 0.42 | 0.86 | 0.41 | 0.87 | 0.40 | 0.87 | 0.43 | 0.86 | 0.45 | 0.89 |
| | | InvCov | 0.49 | 0.87 | 0.45 | 0.86 | 0.36 | 0.81 | 0.44 | 0.86 | 0.45 | 0.83 |
| 10 | 40 | Corr | 0.47 | 0.53 | 0.47 | 0.52 | 0.46 | 0.52 | 0.48 | 0.53 | 0.48 | 0.54 |
| | | D-Corr | 0.46 | 0.53 | 0.46 | 0.51 | 0.46 | 0.54 | 0.48 | 0.53 | 0.47 | 0.54 |
| | | InvCov | 0.50 | 0.57 | 0.48 | 0.52 | 0.47 | 0.53 | 0.47 | 0.50 | 0.48 | 0.52 |
| 100 | 100 | Corr | 0.42 | 0.99 | 0.25 | 0.94 | 0.25 | 0.93 | 0.42 | 0.98 | 0.35 | 0.91 |
| | | D-Corr | 0.41 | 0.99 | 0.25 | 0.96 | 0.26 | 0.96 | 0.41 | 0.98 | 0.37 | 0.94 |
| | | InvCov | 0.40 | 0.99 | 0.22 | 0.94 | 0.16 | 0.87 | 0.40 | 0.97 | 0.36 | 0.90 |
| 100 | 400 | Corr | 0.19 | 0.80 | 0.10 | 0.63 | 0.14 | 0.72 | 0.27 | 0.84 | 0.20 | 0.72 |
| | | D-Corr | 0.19 | 0.80 | 0.10 | 0.63 | 0.14 | 0.75 | 0.26 | 0.84 | 0.21 | 0.74 |
| | | InvCov | 0.25 | 0.91 | 0.09 | 0.62 | 0.14 | 0.77 | 0.27 | 0.86 | 0.20 | 0.67 |

However, there remains a large gap in performance between these ablations and our method, highlighting the value of learning non-linear transformations of these inputs.

Table 16 and Figure 10 show that the current implementations of Sea can generalize to graphs up to $4\times$ larger than those seen during training. During training, we did not initially anticipate testing on much larger

Table 12: Comparison of global statistics (SHD). Discretization thresholds for SHD were obtained by computing the $p^{\text{th}}$ quantile of the computed values, where $p = 1 - (E/N)$.

| $N$ | $E$ | Model | Linear | NN add. | NN non-add. | Sigmoid | Polynomial |
|---|---|---|---|---|---|---|---|
| 10 | 10 | CORR | $10.6_{\pm2.8}$ | $10.2_{\pm4.6}$ | $12.0_{\pm1.9}$ | $11.1_{\pm4.3}$ | $9.9_{\pm2.8}$ |
|  |  | D-CORR | $10.4_{\pm2.6}$ | $9.8_{\pm4.7}$ | $12.2_{\pm2.6}$ | $10.8_{\pm3.3}$ | $10.2_{\pm3.2}$ |
|  |  | INVCOV | $11.0_{\pm2.8}$ | $11.4_{\pm5.5}$ | $13.6_{\pm2.9}$ | $11.4_{\pm4.1}$ | $10.9_{\pm3.5}$ |
| 10 | 40 | CORR | $39.2_{\pm2.4}$ | $38.0_{\pm1.8}$ | $38.2_{\pm0.7}$ | $38.8_{\pm3.3}$ | $38.2_{\pm2.0}$ |
|  |  | D-CORR | $38.8_{\pm2.0}$ | $38.8_{\pm1.5}$ | $37.0_{\pm0.6}$ | $38.9_{\pm3.2}$ | $38.0_{\pm2.0}$ |
|  |  | INVCOV | $35.8_{\pm2.3}$ | $39.2_{\pm1.5}$ | $37.6_{\pm2.7}$ | $40.7_{\pm2.2}$ | $38.4_{\pm1.2}$ |
| 100 | 100 | CORR | $113.0_{\pm4.9}$ | $132.2_{\pm18.0}$ | $144.6_{\pm5.2}$ | $106.5_{\pm11.5}$ | $110.3_{\pm6.1}$ |
|  |  | D-CORR | $113.8_{\pm5.3}$ | $133.2_{\pm17.9}$ | $144.2_{\pm6.7}$ | $108.5_{\pm11.9}$ | $109.5_{\pm5.7}$ |
|  |  | INVCOV | $124.4_{\pm8.1}$ | $130.0_{\pm17.2}$ | $158.8_{\pm6.2}$ | $112.3_{\pm14.8}$ | $106.3_{\pm4.6}$ |
| 100 | 400 | CORR | $580.4_{\pm24.5}$ | $666.0_{\pm13.5}$ | $626.2_{\pm23.4}$ | $516.5_{\pm18.5}$ | $562.5_{\pm20.1}$ |
|  |  | D-CORR | $578.2_{\pm24.7}$ | $665.4_{\pm15.4}$ | $626.6_{\pm21.9}$ | $522.3_{\pm17.6}$ | $557.2_{\pm20.4}$ |
|  |  | INVCOV | $557.0_{\pm11.7}$ | $667.8_{\pm15.4}$ | $639.0_{\pm9.7}$ | $514.7_{\pm23.1}$ | $539.4_{\pm18.4}$ |

Table 13: Causal discovery results on simulated mRNA data. Each setting encompasses 5 distinct scale-free graphs. Data were generated via SERGIO Dibaeinia & Sinha (2020).

| $N$ | $E$ | Model | mAP ↑ | AUC ↑ | SHD ↓ | OA ↑ |
|---|---|---|---|---|---|---|
| 10 | 10 | DCDI-G | $0.48_{\pm0.1}$ | $0.73_{\pm0.1}$ | $16.1_{\pm3.3}$ | $0.59_{\pm0.2}$ |
|  |  | DCDI-DSF | $0.63_{\pm0.1}$ | $0.84_{\pm0.1}$ | $18.5_{\pm2.7}$ | $0.79_{\pm0.2}$ |
|  |  | DCD-FG | $0.59_{\pm0.2}$ | $0.82_{\pm0.1}$ | $81.0_{\pm0.0}$ | $0.79_{\pm0.2}$ |
|  |  | AVICI | $0.58_{\pm0.2}$ | $0.85_{\pm0.1}$ | $6.4_{\pm4.7}$ | $0.72_{\pm0.2}$ |
|  |  | SEA (FCI) | $\mathbf{0.92}_{\pm0.1}$ | $\mathbf{0.98}_{\pm0.0}$ | $\mathbf{1.9}_{\pm2.0}$ | $\mathbf{0.92}_{\pm0.1}$ |
| 10 | 20 | DCDI-G | $0.32_{\pm0.1}$ | $0.57_{\pm0.1}$ | $26.2_{\pm1.3}$ | $0.47_{\pm0.2}$ |
|  |  | DCDI-DSF | $0.44_{\pm0.1}$ | $0.64_{\pm0.1}$ | $25.7_{\pm1.3}$ | $0.63_{\pm0.1}$ |
|  |  | DCD-FG | $0.43_{\pm0.1}$ | $0.69_{\pm0.1}$ | $73.0_{\pm0.0}$ | $0.67_{\pm0.2}$ |
|  |  | AVICI | $0.22_{\pm0.1}$ | $0.44_{\pm0.2}$ | $16.8_{\pm1.5}$ | $0.27_{\pm0.3}$ |
|  |  | SEA (FCI) | $\mathbf{0.76}_{\pm0.1}$ | $\mathbf{0.90}_{\pm0.1}$ | $\mathbf{8.8}_{\pm1.5}$ | $\mathbf{0.85}_{\pm0.1}$ |
| 20 | 20 | DCDI-G | $0.48_{\pm0.1}$ | $0.86_{\pm0.1}$ | $37.3_{\pm2.8}$ | $0.65_{\pm0.1}$ |
|  |  | DCDI-DSF | $0.45_{\pm0.1}$ | $0.92_{\pm0.0}$ | $51.9_{\pm15.8}$ | $0.81_{\pm0.1}$ |
|  |  | DCD-FG | $0.34_{\pm0.2}$ | $0.87_{\pm0.0}$ | $361_{\pm0}$ | $0.66_{\pm0.2}$ |
|  |  | AVICI | $0.32_{\pm0.2}$ | $0.78_{\pm0.1}$ | $18.7_{\pm4.9}$ | $0.66_{\pm0.2}$ |
|  |  | SEA (FCI) | $\mathbf{0.54}_{\pm0.2}$ | $\mathbf{0.94}_{\pm0.0}$ | $\mathbf{16.6}_{\pm3.3}$ | $\mathbf{0.83}_{\pm0.1}$ |
| 20 | 40 | DCDI-G | $0.31_{\pm0.1}$ | $0.65_{\pm0.1}$ | $54.7_{\pm2.7}$ | $0.49_{\pm0.1}$ |
|  |  | DCDI-DSF | $0.40_{\pm0.1}$ | $0.71_{\pm0.1}$ | $54.6_{\pm4.4}$ | $0.63_{\pm0.1}$ |
|  |  | DCD-FG | $0.36_{\pm0.1}$ | $0.77_{\pm0.1}$ | $343_{\pm0}$ | $0.67_{\pm0.1}$ |
|  |  | AVICI | $0.17_{\pm0.1}$ | $0.54_{\pm0.1}$ | $37.1_{\pm1.9}$ | $0.46_{\pm0.1}$ |
|  |  | SEA (FCI) | $\mathbf{0.50}_{\pm0.1}$ | $\mathbf{0.85}_{\pm0.1}$ | $\mathbf{31.4}_{\pm4.9}$ | $\mathbf{0.78}_{\pm0.1}$ |

graphs. As a result, there are two minor issues with the current implementation with respect to scaling. First, we set an insufficient maximum subset positional embedding size of 500, so it was impossible to encode more subsets. Second, we did not sample random starting subset indices to ensure that higher-order embeddings are updated equally. Since we never sampled up to 500 subsets during training, these higher-order embeddings were essentially random. We anticipate that increasing the limit on the number of subsets and ensuring that all embeddings are sufficiently learned will improve the generalization capacity on larger graphs. Nonetheless, our current model already obtains reasonable performance on larger graphs, out of the box.

Table 14: SHD on synthetic graphs, observational setting, between predicted vs. true DAG and inferred vs. true CPDAG. Mean/std over 5 distinct Erdős-Rényi graphs. † indicates o.o.d. setting. ∗ indicates non-parametric bootstrapping. DAG results from Table 1.

| $N$ | $E$ | Model | Linear | | NN add. | | Sigmoid[†] | | Polynomial[†] | |
|---|---|---|---|---|---|---|---|---|---|---|
| | | | DAG ↓ | CPDAG ↓ | DAG ↓ | CPDAG ↓ | DAG ↓ | CPDAG ↓ | DAG ↓ | CPDAG ↓ |
| 10 | 10 | DiffAn | $14.0_{\pm.4}$ | $16.4_{\pm.4}$ | $13.6_{\pm2.9}$ | $13.6_{\pm3.4}$ | $12.0_{\pm.0}$ | $12.9_{\pm.3}$ | $15.0_{\pm.1}$ | $16.3_{\pm.6}$ |
| | | VarSort* | $6.0_{\pm.4}$ | $7.0_{\pm.3}$ | $4.0_{\pm.8}$ | $4.4_{\pm.2}$ | $7.6_{\pm.5}$ | $8.0_{\pm.1}$ | $9.3_{\pm.5}$ | $9.5_{\pm.8}$ |
| | | Fci* | $10.0_{\pm.3}$ | $11.0_{\pm.8}$ | $8.2_{\pm.8}$ | $8.4_{\pm.8}$ | $9.1_{\pm.5}$ | $10.3_{\pm.3}$ | $10.0_{\pm.3}$ | $10.1_{\pm.2}$ |
| | | Sea (Fci) | $\mathbf{1.0}_{\pm.1}$ | $\mathbf{1.4}_{\pm.7}$ | $\mathbf{3.0}_{\pm.2}$ | $\mathbf{2.6}_{\pm.3}$ | $\mathbf{3.9}_{\pm.9}$ | $\mathbf{3.5}_{\pm.3}$ | $\mathbf{6.1}_{\pm.7}$ | $\mathbf{6.4}_{\pm.1}$ |
| 20 | 20 | DiffAn | $40.2_{\pm4.4}$ | $41.8_{\pm5.7}$ | $38.6_{\pm3.1}$ | $39.8_{\pm5.2}$ | $19.2_{\pm.6}$ | $21.6_{\pm.2}$ | $49.7_{\pm4.6}$ | $52.7_{\pm5.0}$ |
| | | VarSort* | $10.0_{\pm.4}$ | $11.0_{\pm.0}$ | $6.6_{\pm.7}$ | $7.2_{\pm.5}$ | $16.1_{\pm.7}$ | $16.8_{\pm.2}$ | $17.1_{\pm.1}$ | $17.7_{\pm.0}$ |
| | | Fci* | $19.0_{\pm.3}$ | $21.4_{\pm.3}$ | $17.4_{\pm.2}$ | $18.0_{\pm.6}$ | $18.5_{\pm.5}$ | $20.6_{\pm.4}$ | $18.9_{\pm.3}$ | $19.1_{\pm.2}$ |
| | | Sea (Fci) | $\mathbf{3.2}_{\pm.6}$ | $\mathbf{2.6}_{\pm.1}$ | $\mathbf{5.0}_{\pm.8}$ | $\mathbf{4.8}_{\pm.7}$ | $\mathbf{6.7}_{\pm.1}$ | $\mathbf{6.2}_{\pm.1}$ | $\mathbf{9.8}_{\pm.2}$ | $\mathbf{11.0}_{\pm.2}$ |

Table 15: mAP on synthetic graphs, between predicted vs. true DAG and undirected skeletons $E^*$. Mean/std over 5 distinct Erdős-Rényi graphs. † indicates o.o.d. setting. ∗ indicates non-parametric bootstrapping. DAG results from Table 1.

| $N$ | $E$ | Model | Linear | | NN add. | | Sigmoid[†] | | Polynomial[†] | |
|---|---|---|---|---|---|---|---|---|---|---|
| | | | DAG ↑ | $E^*$ ↑ | DAG ↑ | $E^*$ ↑ | DAG ↑ | $E^*$ ↑ | DAG ↑ | $E^*$ ↑ |
| 10 | 10 | Dcdi-G | $0.74_{\pm.16}$ | $0.88_{\pm.10}$ | $0.79_{\pm.12}$ | $0.85_{\pm.12}$ | $0.46_{\pm.24}$ | $0.64_{\pm.20}$ | $0.41_{\pm.13}$ | $0.58_{\pm.15}$ |
| | | Dcdi-Dsf | $0.82_{\pm.20}$ | $0.93_{\pm.09}$ | $0.57_{\pm.24}$ | $0.88_{\pm.11}$ | $0.38_{\pm.21}$ | $0.63_{\pm.19}$ | $0.29_{\pm.13}$ | $0.61_{\pm.20}$ |
| | | DiffAn | $0.25_{\pm.06}$ | $0.54_{\pm.09}$ | $0.32_{\pm.16}$ | $0.62_{\pm.19}$ | $0.24_{\pm.10}$ | $0.63_{\pm.12}$ | $0.20_{\pm.08}$ | $0.50_{\pm.07}$ |
| | | Avici | $0.45_{\pm.14}$ | $0.93_{\pm.06}$ | $0.81_{\pm.15}$ | $\mathbf{0.98}_{\pm.03}$ | $0.52_{\pm.16}$ | $0.86_{\pm.13}$ | $0.31_{\pm.06}$ | $0.75_{\pm.13}$ |
| | | VarSort* | $0.70_{\pm.13}$ | $0.88_{\pm.07}$ | $0.76_{\pm.13}$ | $0.90_{\pm.09}$ | $0.52_{\pm.24}$ | $0.88_{\pm.10}$ | $0.40_{\pm.14}$ | $0.80_{\pm.09}$ |
| | | Fci* | $0.52_{\pm.11}$ | $0.70_{\pm.20}$ | $0.38_{\pm.20}$ | $0.69_{\pm.22}$ | $0.56_{\pm.16}$ | $0.75_{\pm.18}$ | $0.41_{\pm.13}$ | $0.61_{\pm.14}$ |
| | | Gies* | $0.81_{\pm.12}$ | $\mathbf{1.0}_{\pm.00}$ | $0.61_{\pm.16}$ | $0.97_{\pm.06}$ | $0.70_{\pm.14}$ | $0.98_{\pm.03}$ | $0.61_{\pm.10}$ | $\mathbf{0.89}_{\pm.06}$ |
| | | Sea (Fci) | $0.98_{\pm.02}$ | $\mathbf{1.0}_{\pm.0}$ | $0.88_{\pm.09}$ | $0.96_{\pm.05}$ | $0.83_{\pm.18}$ | $\mathbf{0.99}_{\pm.02}$ | $0.62_{\pm.09}$ | $0.87_{\pm.06}$ |
| | | Sea (Gies) | $\mathbf{0.99}_{\pm.01}$ | $\mathbf{1.0}_{\pm.0}$ | $\mathbf{0.94}_{\pm.06}$ | $0.97_{\pm.05}$ | $\mathbf{0.85}_{\pm.12}$ | $0.98_{\pm.03}$ | $\mathbf{0.70}_{\pm.11}$ | $0.86_{\pm.05}$ |
| 20 | 20 | Dcdi-G | $0.59_{\pm.12}$ | $0.89_{\pm.04}$ | $0.78_{\pm.07}$ | $0.93_{\pm.07}$ | $0.36_{\pm.06}$ | $0.84_{\pm.05}$ | $0.42_{\pm.08}$ | $0.58_{\pm.10}$ |
| | | Dcdi-Dsf | $0.66_{\pm.16}$ | $0.91_{\pm.04}$ | $0.69_{\pm.18}$ | $0.93_{\pm.10}$ | $0.37_{\pm.04}$ | $0.86_{\pm.05}$ | $0.26_{\pm.08}$ | $0.55_{\pm.15}$ |
| | | DiffAn | $0.19_{\pm.09}$ | $0.47_{\pm.16}$ | $0.16_{\pm.10}$ | $0.44_{\pm.15}$ | $0.29_{\pm.11}$ | $0.67_{\pm.12}$ | $0.09_{\pm.03}$ | $0.32_{\pm.07}$ |
| | | Avici | $0.48_{\pm.17}$ | $0.84_{\pm.07}$ | $0.59_{\pm.09}$ | $0.85_{\pm.07}$ | $0.42_{\pm.13}$ | $0.78_{\pm.10}$ | $0.24_{\pm.08}$ | $0.62_{\pm.08}$ |
| | | VarSort* | $0.81_{\pm.08}$ | $0.93_{\pm.04}$ | $0.81_{\pm.15}$ | $0.88_{\pm.14}$ | $0.50_{\pm.13}$ | $0.88_{\pm.09}$ | $0.33_{\pm.13}$ | $0.76_{\pm.11}$ |
| | | Fci* | $0.66_{\pm.07}$ | $0.79_{\pm.07}$ | $0.42_{\pm.19}$ | $0.54_{\pm.22}$ | $0.56_{\pm.08}$ | $0.67_{\pm.07}$ | $0.41_{\pm.14}$ | $0.61_{\pm.13}$ |
| | | Gies* | $0.84_{\pm.08}$ | $\mathbf{0.99}_{\pm.00}$ | $0.79_{\pm.07}$ | $0.94_{\pm.04}$ | $0.71_{\pm.10}$ | $0.95_{\pm.03}$ | $0.62_{\pm.09}$ | $\mathbf{0.85}_{\pm.09}$ |
| | | Sea (Fci) | $\mathbf{0.96}_{\pm.03}$ | $\mathbf{1.0}_{\pm.00}$ | $0.91_{\pm.04}$ | $0.95_{\pm.03}$ | $\mathbf{0.85}_{\pm.09}$ | $\mathbf{0.97}_{\pm.02}$ | $\mathbf{0.69}_{\pm.09}$ | $\mathbf{0.86}_{\pm.09}$ |
| | | Sea (Gies) | $\mathbf{0.97}_{\pm.02}$ | $\mathbf{1.0}_{\pm.00}$ | $\mathbf{0.94}_{\pm.03}$ | $\mathbf{0.96}_{\pm.03}$ | $0.84_{\pm.07}$ | $0.95_{\pm.03}$ | $\mathbf{0.69}_{\pm.12}$ | $0.82_{\pm.11}$ |

Finally, we note that Avici scales very poorly to graphs significantly beyond the scope of their training set. For example, $N = 100$ is only 2× their largest training graphs, but the performance already drops dramatically.

Figure 9 depicts the model runtimes. Sea continues to run quickly on much larger graphs, while Avici runtimes increase significantly with graph size.

Dcdi learns a new generative model over each dataset, and its more powerful, deep sigmoidal flow variant seems to perform well in some (but not all) of these harder cases.

## C.5 Results on real datasets

The Sachs flow cytometry dataset (Sachs et al., 2005) measured the expression of phosphoproteins and phospholipids at the single cell level. We use the subset proposed by Wang et al. (2017). The ground truth

Table 16: Scaling to synthetic graphs, larger than those seen in training. Each setting encompasses 5 distinct Erdős-Rényi graphs. All SEA runs in this table used $T = 500$ subsets of nodes, with $b = 500$ examples per batch. For AVICI, we took $M = 2000$ samples per dataset (higher than maximum analyzed in their paper), since it performed better than $M = 1000$. Here, the mean AUC values are artificially high due to the high negative rates, as actual edges scale linearly as $N$, while the number of possible edges scales quadratically.

| $N$ | Model | Linear, $E = N$ | | | | Linear, $E = 4N$ | | | |
|---|---|---|---|---|---|---|---|---|---|
| | | mAP ↑ | AUC ↑ | SHD ↓ | OA ↑ | mAP ↑ | AUC ↑ | SHD ↓ | OA ↑ |
| 100 | INVCOV | $0.43_{\pm0.0}$ | $0.99_{\pm0.0}$ | $117_{\pm7}$ | — | $0.30_{\pm0.0}$ | $0.93_{\pm0.0}$ | $512_{\pm11}$ | — |
| | CORR | $0.42_{\pm0.0}$ | $0.99_{\pm0.0}$ | $113_{\pm5}$ | — | $0.19_{\pm0.0}$ | $0.80_{\pm0.0}$ | $579_{\pm25}$ | — |
| | AVICI | $0.03_{\pm0.0}$ | $0.43_{\pm0.1}$ | $109_{\pm6}$ | $0.49_{\pm0.0}$ | $0.11_{\pm0.0}$ | $0.55_{\pm0.1}$ | $394_{\pm14}$ | $0.58_{\pm0.0}$ |
| | SEA (FCI) | $\mathbf{0.97}_{\pm0.0}$ | $\mathbf{1.00}_{\pm0.0}$ | $\mathbf{11.6}_{\pm4.3}$ | $\mathbf{0.93}_{\pm0.0}$ | $0.88_{\pm0.0}$ | $0.98_{\pm0.0}$ | $129_{\pm10}$ | $0.94_{\pm0.0}$ |
| | SEA (GIES) | $\mathbf{0.97}_{\pm0.0}$ | $\mathbf{1.00}_{\pm0.0}$ | $12.8_{\pm4.7}$ | $0.91_{\pm0.0}$ | $\mathbf{0.91}_{\pm0.0}$ | $\mathbf{0.99}_{\pm0.0}$ | $\mathbf{105}_{\pm6}$ | $\mathbf{0.95}_{\pm0.0}$ |
| 200 | INVCOV | $0.45_{\pm0.0}$ | $1.00_{\pm0.0}$ | $218_{\pm11}$ | — | $0.33_{\pm0.0}$ | $0.96_{\pm0.0}$ | $1000_{\pm23}$ | — |
| | CORR | $0.42_{\pm0.0}$ | $0.99_{\pm0.0}$ | $223_{\pm8}$ | — | $0.18_{\pm0.0}$ | $0.86_{\pm0.0}$ | $1184_{\pm25}$ | — |
| | AVICI | $0.00_{\pm0.0}$ | $0.36_{\pm0.1}$ | $207_{\pm10}$ | $0.41_{\pm0.1}$ | $0.05_{\pm0.0}$ | $0.53_{\pm0.1}$ | $827_{\pm37}$ | $0.54_{\pm0.1}$ |
| | SEA (FCI) | $0.91_{\pm0.0}$ | $\mathbf{1.00}_{\pm0.0}$ | $49.9_{\pm5.4}$ | $0.87_{\pm0.0}$ | $0.82_{\pm0.0}$ | $0.97_{\pm0.0}$ | $327_{\pm52}$ | $0.92_{\pm0.0}$ |
| | SEA (GIES) | $\mathbf{0.95}_{\pm0.0}$ | $\mathbf{1.00}_{\pm0.0}$ | $\mathbf{35.4}_{\pm5.7}$ | $\mathbf{0.91}_{\pm0.0}$ | $\mathbf{0.86}_{\pm0.0}$ | $\mathbf{0.98}_{\pm0.0}$ | $\mathbf{272}_{\pm50}$ | $\mathbf{0.92}_{\pm0.0}$ |
| 300 | INVCOV | $0.46_{\pm0.0}$ | $\mathbf{1.00}_{\pm0.0}$ | $308_{\pm20}$ | — | $0.35_{\pm0.0}$ | $0.98_{\pm0.0}$ | $1445_{\pm56}$ | — |
| | CORR | $0.42_{\pm0.0}$ | $1.00_{\pm0.0}$ | $326_{\pm21}$ | — | $0.20_{\pm0.0}$ | $0.89_{\pm0.0}$ | $1710_{\pm82}$ | — |
| | AVICI | $0.01_{\pm0.0}$ | $0.70_{\pm0.0}$ | $298_{\pm19}$ | $0.64_{\pm0.0}$ | $0.02_{\pm0.0}$ | $0.50_{\pm0.0}$ | $1214_{\pm68}$ | $0.51_{\pm0.0}$ |
| | SEA (FCI) | $0.80_{\pm0.0}$ | $1.00_{\pm0.0}$ | $121_{\pm14}$ | $0.78_{\pm0.0}$ | $0.70_{\pm0.0}$ | $0.95_{\pm0.0}$ | $693_{\pm67}$ | $0.86_{\pm0.0}$ |
| | SEA (GIES) | $\mathbf{0.88}_{\pm0.0}$ | $1.00_{\pm0.0}$ | $\mathbf{88.9}_{\pm11.3}$ | $\mathbf{0.84}_{\pm0.0}$ | $\mathbf{0.78}_{\pm0.0}$ | $\mathbf{0.96}_{\pm0.0}$ | $\mathbf{556}_{\pm71}$ | $\mathbf{0.87}_{\pm0.0}$ |
| 400 | INVCOV | $0.47_{\pm0.0}$ | $1.00_{\pm0.0}$ | $418_{\pm7}$ | — | $0.36_{\pm0.0}$ | $0.98_{\pm0.0}$ | $1883_{\pm28}$ | — |
| | CORR | $0.42_{\pm0.0}$ | $1.00_{\pm0.0}$ | $445_{\pm14}$ | — | $0.20_{\pm0.0}$ | $0.91_{\pm0.0}$ | $2269_{\pm52}$ | — |
| | AVICI | $0.01_{\pm0.0}$ | $0.68_{\pm0.0}$ | $411_{\pm7}$ | $0.62_{\pm0.0}$ | $0.01_{\pm0.0}$ | $0.46_{\pm0.0}$ | $1614_{\pm21}$ | $0.47_{\pm0.0}$ |
| | SEA (FCI) | $0.49_{\pm0.2}$ | $0.93_{\pm0.1}$ | $314_{\pm107}$ | $0.61_{\pm0.1}$ | $0.56_{\pm0.1}$ | $0.90_{\pm0.1}$ | $1103_{\pm190}$ | $0.75_{\pm0.1}$ |
| | SEA (GIES) | $\mathbf{0.70}_{\pm0.1}$ | $\mathbf{0.99}_{\pm0.0}$ | $\mathbf{226}_{\pm57}$ | $\mathbf{0.71}_{\pm0.1}$ | $\mathbf{0.70}_{\pm0.0}$ | $\mathbf{0.94}_{\pm0.0}$ | $\mathbf{872}_{\pm44}$ | $\mathbf{0.80}_{\pm0.0}$ |
| 500 | INVCOV | $0.47_{\pm0.0}$ | $1.00_{\pm0.0}$ | $504_{\pm19}$ | — | $0.38_{\pm0.0}$ | $0.99_{\pm0.0}$ | $2300_{\pm34}$ | — |
| | CORR | $0.42_{\pm0.0}$ | $1.00_{\pm0.0}$ | $543_{\pm18}$ | — | $0.21_{\pm0.0}$ | $0.93_{\pm0.0}$ | $2790_{\pm78}$ | — |
| | AVICI | $0.00_{\pm0.0}$ | $0.70_{\pm0.0}$ | $497_{\pm19}$ | $\mathbf{0.63}_{\pm0.0}$ | $0.01_{\pm0.0}$ | $0.48_{\pm0.0}$ | $2004_{\pm25}$ | $0.48_{\pm0.0}$ |
| | SEA (FCI) | $0.27_{\pm0.1}$ | $0.90_{\pm0.1}$ | $758_{\pm297}$ | $0.51_{\pm0.0}$ | $0.29_{\pm0.1}$ | $0.86_{\pm0.1}$ | $1824_{\pm273}$ | $0.56_{\pm0.1}$ |
| | SEA (GIES) | $\mathbf{0.41}_{\pm0.2}$ | $\mathbf{0.98}_{\pm0.0}$ | $\mathbf{485}_{\pm170}$ | $0.57_{\pm0.1}$ | $\mathbf{0.48}_{\pm0.1}$ | $\mathbf{0.92}_{\pm0.0}$ | $\mathbf{1654}_{\pm505}$ | $\mathbf{0.67}_{\pm0.0}$ |

"consensus graph" consists of 11 nodes and 17 edges over 5,845 samples, of which 1,755 are observational and 4,091 are interventional. The observational data were generated by a "general perturbation" which activated signaling pathways, and the interventional data were generated by perturbations intended to target individual proteins. Despite the popularity of this dataset in causal discovery literature (due to lack of better alternatives), biological networks are known to be time-resolved and cyclic, so the validity of the ground truth "consensus" graph has been questioned by experts Mooij et al. (2020). Nonetheless, we benchmark all methods on this dataset in Table 26.

Table 17: Causal discovery ablations by setting hidden representations to zero. Each setting encompasses 5 distinct Erdős-Rényi graphs. The symbol † indicates that SEA was not pretrained on this setting. We set $T = 100$.

| $N$ | $E$ | Model | Linear | | NN add. | | NN non-add. | | Sigmoid† | | Polynomial† | |
|---|---|---|---|---|---|---|---|---|---|---|---|---|
| | | | mAP ↑ | SHD ↓ | mAP ↑ | SHD ↓ | mAP ↑ | SHD ↓ | mAP ↑ | SHD ↓ | mAP ↑ | SHD ↓ |
| 10 | 10 | SEA (FCI) | 0.97 | 1.6 | 0.95 | 2.4 | 0.92 | 2.8 | 0.83 | 3.7 | 0.69 | 6.7 |
| | | $h^\rho \leftarrow 0$ | 0.20 | 29.8 | 0.27 | 22.4 | 0.34 | 24.2 | 0.26 | 22.6 | 0.24 | 26.0 |
| | | $h^E \leftarrow 0$ | 0.61 | 24.0 | 0.71 | 28.8 | 0.76 | 27.8 | 0.49 | 26.7 | 0.51 | 26.7 |
| | | SEA (GIES) | 0.99 | 1.2 | 0.94 | 2.6 | 0.91 | 3.2 | 0.85 | 4.0 | 0.70 | 5.8 |
| | | $h^\rho \leftarrow 0$ | 0.30 | 29.2 | 0.27 | 29.4 | 0.19 | 29.0 | 0.35 | 27.4 | 0.31 | 27.1 |
| | | $h^E \leftarrow 0$ | 0.85 | 6.4 | 0.82 | 10.2 | 0.78 | 13.2 | 0.63 | 10.2 | 0.55 | 13.4 |
| 20 | 80 | SEA (FCI) | 0.86 | 29.6 | 0.55 | 73.6 | 0.72 | 51.8 | 0.77 | 42.8 | 0.61 | 61.8 |
| | | $h^\rho \leftarrow 0$ | 0.23 | 128.4 | 0.27 | 110.6 | 0.22 | 119.6 | 0.23 | 110.7 | 0.23 | 111.1 |
| | | $h^E \leftarrow 0$ | 0.79 | 50.4 | 0.52 | 99.6 | 0.71 | 76.4 | 0.58 | 93.6 | 0.59 | 87.5 |
| | | SEA (GIES) | 0.89 | 26.8 | 0.58 | 71.4 | 0.73 | 50.6 | 0.76 | 45.0 | 0.65 | 60.1 |
| | | $h^\rho \leftarrow 0$ | 0.25 | 125.2 | 0.21 | 123.0 | 0.24 | 113.6 | 0.27 | 118.0 | 0.24 | 129.8 |
| | | $h^E \leftarrow 0$ | 0.86 | 35.2 | 0.55 | 76.8 | 0.70 | 53.4 | 0.69 | 47.1 | 0.59 | 63.5 |
| 100 | 400 | SEA (FCI) | 0.90 | 122.0 | 0.28 | 361.2 | 0.60 | 273.2 | 0.69 | 226.9 | 0.38 | 327.0 |
| | | $h^\rho \leftarrow 0$ | 0.05 | 726.4 | 0.04 | 639.4 | 0.05 | 637.0 | 0.05 | 760.2 | 0.04 | 658.3 |
| | | $h^E \leftarrow 0$ | 0.82 | 167.4 | 0.04 | 403.4 | 0.51 | 352.4 | 0.64 | 263.8 | 0.33 | 366.1 |
| | | SEA (GIES) | 0.91 | 116.6 | 0.27 | 364.4 | 0.61 | 266.8 | 0.69 | 218.3 | 0.38 | 328.0 |
| | | $h^\rho \leftarrow 0$ | 0.05 | 780.0 | 0.04 | 846.0 | 0.05 | 715.4 | 0.04 | 744.4 | 0.04 | 769.8 |
| | | $h^E \leftarrow 0$ | 0.86 | 134.8 | 0.03 | 403.4 | 0.52 | 357.8 | 0.67 | 224.1 | 0.32 | 359.6 |

Table 18: Our predicted graphs are highly acyclic, on synthetic ER test sets.

| $N$ | Acyclic | Total | Proportion |
|---|---|---|---|
| 10 | 434 | 440 | 0.99 |
| 20 | 431 | 440 | 0.98 |
| 100 | 433 | 440 | 0.98 |

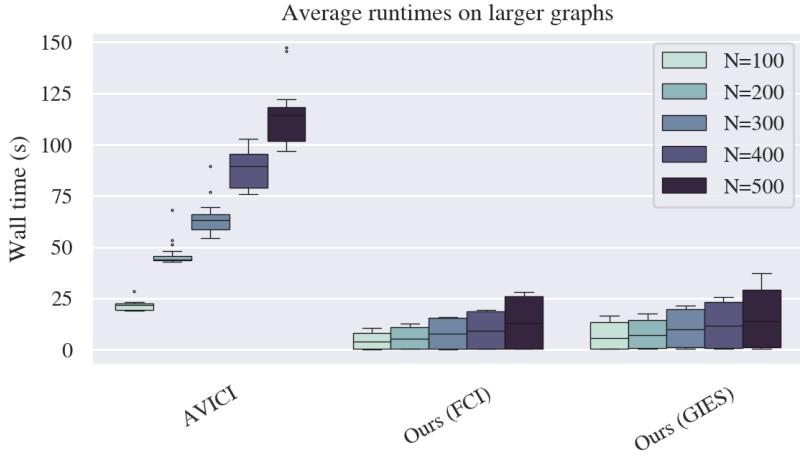

Figure 9: SEA scales very well in terms of runtime on much larger graphs, while AVICI runtimes suffer as graph sizes increase.

Table 19: Full results on synthetic datasets (continuous metrics). Mean/std over 5 distinct Erdős-Rényi graphs. † indicates o.o.d. setting. ∗ indicates non-parametric bootstrapping. All standard deviations within 0.03 (most within 0.01).

| N | E | Model | Linear | | NN add. | | NN non-add. | | Sigmoid† | | Polynomial† | |
|---|---|---|---|---|---|---|---|---|---|---|---|---|
| | | | mAP ↑ | AUC ↑ | mAP ↑ | AUC ↑ | mAP ↑ | AUC ↑ | mAP ↑ | AUC ↑ | mAP ↑ | AUC ↑ |
| 10 | 10 | DCDI-G | 0.74 | 0.88 | 0.79 | 0.91 | 0.89 | 0.95 | 0.46 | 0.72 | 0.41 | 0.68 |
| | | DCDI-DSF | 0.82 | 0.92 | 0.57 | 0.83 | 0.50 | 0.81 | 0.38 | 0.69 | 0.29 | 0.64 |
| | | DCD-FG | 0.45 | 0.68 | 0.41 | 0.67 | 0.59 | 0.79 | 0.40 | 0.64 | 0.50 | 0.72 |
| | | DIFFAN | 0.25 | 0.73 | 0.32 | 0.70 | 0.12 | 0.51 | 0.24 | 0.70 | 0.20 | 0.65 |
| | | DECI | 0.18 | 0.63 | 0.16 | 0.63 | 0.23 | 0.71 | 0.29 | 0.72 | 0.46 | 0.83 |
| | | AVICI | 0.45 | 0.80 | 0.81 | 0.97 | 0.65 | 0.89 | 0.52 | 0.81 | 0.31 | 0.70 |
| | | VARSORT* | 0.70 | 0.84 | 0.76 | 0.90 | 0.83 | 0.93 | 0.52 | 0.72 | 0.40 | 0.71 |
| | | INVCOV | 0.46 | 0.88 | 0.43 | 0.86 | 0.34 | 0.81 | 0.43 | 0.86 | 0.43 | 0.83 |
| | | FCI* | 0.52 | 0.78 | 0.38 | 0.71 | 0.40 | 0.70 | 0.56 | 0.79 | 0.41 | 0.71 |
| | | GIES* | 0.81 | 0.96 | 0.61 | 0.93 | 0.71 | 0.92 | 0.70 | 0.95 | 0.61 | 0.87 |
| | | SEA (FCI) | 0.98 | 1.00 | 0.88 | 0.98 | 0.88 | 0.97 | 0.83 | **0.97** | 0.62 | 0.88 |
| | | SEA (GIES) | **0.99** | **1.00** | **0.94** | **0.99** | **0.91** | **0.98** | **0.85** | 0.97 | **0.70** | **0.89** |
| 10 | 40 | DCDI-G | 0.65 | 0.72 | 0.65 | 0.74 | 0.84 | 0.88 | 0.54 | 0.59 | 0.56 | 0.61 |
| | | DCDI-DSF | 0.65 | 0.72 | 0.59 | 0.67 | 0.84 | 0.90 | 0.52 | 0.58 | 0.57 | 0.62 |
| | | DCD-FG | 0.51 | 0.57 | 0.57 | 0.63 | 0.53 | 0.59 | 0.65 | 0.68 | 0.65 | 0.68 |
| | | DIFFAN | 0.40 | 0.49 | 0.36 | 0.37 | 0.41 | 0.53 | 0.40 | 0.45 | 0.37 | 0.40 |
| | | DECI | 0.45 | 0.49 | 0.50 | 0.58 | 0.44 | 0.52 | 0.53 | 0.61 | 0.63 | 0.72 |
| | | AVICI | 0.46 | 0.47 | 0.63 | 0.65 | 0.79 | 0.86 | 0.49 | 0.53 | 0.47 | 0.55 |
| | | VARSORT* | 0.81 | 0.83 | 0.87 | 0.89 | 0.71 | 0.73 | 0.69 | 0.72 | 0.56 | 0.62 |
| | | INVCOV | 0.49 | 0.57 | 0.46 | 0.50 | 0.46 | 0.53 | 0.47 | 0.50 | 0.48 | 0.53 |
| | | FCI* | 0.43 | 0.50 | 0.50 | 0.53 | 0.46 | 0.52 | 0.50 | 0.51 | 0.45 | 0.50 |
| | | GIES* | 0.49 | 0.53 | 0.56 | 0.64 | 0.49 | 0.53 | 0.44 | 0.46 | 0.61 | 0.62 |
| | | SEA (FCI) | 0.83 | 0.85 | 0.85 | 0.89 | 0.86 | 0.90 | 0.74 | 0.75 | 0.69 | 0.67 |
| | | SEA (GIES) | **0.94** | **0.95** | **0.91** | **0.94** | **0.89** | **0.92** | **0.81** | **0.85** | **0.89** | **0.92** |
| 20 | 20 | DCDI-G | 0.59 | 0.87 | 0.78 | 0.94 | 0.75 | 0.91 | 0.36 | 0.81 | 0.42 | 0.74 |
| | | DCDI-DSF | 0.66 | 0.89 | 0.69 | 0.91 | 0.41 | 0.83 | 0.37 | 0.82 | 0.26 | 0.71 |
| | | DCD-FG | 0.48 | 0.85 | 0.58 | 0.91 | 0.51 | 0.87 | 0.50 | 0.78 | 0.44 | 0.76 |
| | | DIFFAN | 0.19 | 0.73 | 0.16 | 0.69 | 0.20 | 0.72 | 0.29 | 0.79 | 0.09 | 0.65 |
| | | DECI | 0.14 | 0.70 | 0.14 | 0.72 | 0.16 | 0.73 | 0.24 | 0.79 | 0.35 | 0.84 |
| | | AVICI | 0.48 | 0.87 | 0.59 | 0.91 | 0.67 | 0.90 | 0.42 | 0.84 | 0.24 | 0.69 |
| | | VARSORT* | 0.81 | 0.91 | 0.81 | 0.92 | 0.57 | 0.83 | 0.50 | 0.76 | 0.33 | 0.69 |
| | | INVCOV | 0.40 | 0.90 | 0.31 | 0.90 | 0.31 | 0.84 | 0.42 | 0.92 | 0.41 | 0.87 |
| | | FCI* | 0.66 | 0.86 | 0.42 | 0.74 | 0.40 | 0.77 | 0.56 | 0.80 | 0.41 | 0.76 |
| | | GIES* | 0.84 | 0.99 | 0.79 | 0.97 | 0.56 | 0.93 | 0.71 | 0.97 | 0.62 | 0.91 |
| | | SEA (FCI) | 0.96 | **1.00** | 0.91 | 0.99 | 0.82 | **0.97** | **0.85** | **0.98** | **0.69** | 0.91 |
| | | SEA (GIES) | **0.97** | **1.00** | **0.94** | **0.99** | **0.83** | 0.97 | 0.84 | 0.97 | 0.69 | **0.92** |
| 20 | 80 | DCDI-G | 0.46 | 0.73 | 0.41 | 0.71 | **0.82** | **0.93** | 0.48 | 0.71 | 0.37 | 0.62 |
| | | DCDI-DSF | 0.48 | 0.75 | 0.44 | 0.74 | 0.74 | 0.92 | 0.48 | 0.71 | 0.38 | 0.63 |
| | | DCD-FG | 0.32 | 0.61 | 0.33 | 0.64 | 0.41 | 0.73 | 0.47 | 0.74 | 0.49 | 0.69 |
| | | DIFFAN | 0.21 | 0.53 | 0.19 | 0.41 | 0.18 | 0.46 | 0.22 | 0.55 | 0.18 | 0.37 |
| | | DECI | 0.25 | 0.57 | 0.29 | 0.61 | 0.26 | 0.59 | 0.31 | 0.66 | 0.43 | 0.73 |
| | | AVICI | 0.34 | 0.63 | 0.46 | 0.73 | 0.49 | 0.74 | 0.34 | 0.64 | 0.30 | 0.59 |
| | | VARSORT* | 0.76 | 0.86 | 0.50 | 0.81 | 0.47 | 0.69 | 0.59 | 0.76 | 0.38 | 0.63 |
| | | INVCOV | 0.36 | 0.72 | 0.26 | 0.54 | 0.30 | 0.64 | 0.35 | 0.72 | 0.32 | 0.61 |
| | | FCI* | 0.30 | 0.59 | 0.31 | 0.57 | 0.30 | 0.59 | 0.41 | 0.66 | 0.34 | 0.61 |
| | | GIES* | 0.41 | 0.75 | 0.44 | 0.74 | 0.46 | 0.73 | 0.50 | 0.78 | 0.49 | 0.69 |
| | | SEA (FCI) | 0.80 | 0.92 | 0.55 | 0.81 | 0.70 | 0.89 | 0.74 | 0.85 | 0.55 | 0.67 |
| | | SEA (GIES) | **0.89** | **0.95** | **0.58** | **0.84** | 0.73 | 0.90 | **0.76** | **0.90** | **0.65** | **0.84** |

Table 20: Full results on synthetic datasets (discrete metrics). Mean/std over 5 distinct Erdős-Rényi graphs. † indicates o.o.d. setting. ∗ indicates non-parametric bootstrapping. All OA standard deviations within 0.2.

| N | E | Model | Linear | | NN add. | | NN non-add. | | Sigmoid† | | Polynomial† | |
|---|---|---|---|---|---|---|---|---|---|---|---|---|
| | | | OA ↑ | SHD ↓ | OA ↑ | SHD ↓ | OA ↑ | SHD ↓ | OA ↑ | SHD ↓ | OA ↑ | SHD ↓ |
| 10 | 10 | DCDI-G | 0.73 | $2.8_{\pm2}$ | 0.84 | $\mathbf{2.2}_{\pm3}$ | 0.88 | $\mathbf{1.0}_{\pm1}$ | 0.46 | $5.8_{\pm3}$ | 0.33 | $8.9_{\pm6}$ |
| | | DCDI-DSF | 0.81 | $2.0_{\pm3}$ | 0.73 | $3.0_{\pm3}$ | 0.60 | $4.2_{\pm1}$ | 0.43 | $6.3_{\pm3}$ | 0.24 | $11.2_{\pm5}$ |
| | | DCD-FG | 0.50 | $20.4_{\pm3}$ | 0.47 | $21.2_{\pm4}$ | 0.70 | $19.2_{\pm4}$ | 0.43 | $19.8_{\pm4}$ | 0.54 | $18.5_{\pm5}$ |
| | | DIFFAN | 0.61 | $14.0_{\pm5}$ | 0.55 | $13.6_{\pm14}$ | 0.26 | $21.8_{\pm8}$ | 0.53 | $12.0_{\pm5}$ | 0.47 | $15.0_{\pm6}$ |
| | | DECI | 0.50 | $19.4_{\pm5}$ | 0.43 | $13.8_{\pm6}$ | 0.62 | $16.2_{\pm3}$ | 0.63 | $13.9_{\pm7}$ | 0.75 | $7.8_{\pm4}$ |
| | | AVICI | 0.58 | $8.2_{\pm4}$ | 0.79 | $4.2_{\pm3}$ | 0.65 | $5.6_{\pm3}$ | 0.54 | $8.3_{\pm3}$ | 0.35 | $9.6_{\pm4}$ |
| | | VARSORT* | 0.70 | $6.0_{\pm2}$ | 0.74 | $4.0_{\pm2}$ | 0.90 | $4.2_{\pm3}$ | 0.52 | $7.6_{\pm3}$ | 0.48 | $9.3_{\pm3}$ |
| | | INVCOV | — | $10.6_{\pm3}$ | — | $10.2_{\pm6}$ | — | $13.6_{\pm3}$ | — | $11.1_{\pm4}$ | — | $10.4_{\pm3}$ |
| | | FCI* | 0.52 | $10.0_{\pm3}$ | 0.43 | $8.2_{\pm4}$ | 0.41 | $9.8_{\pm2}$ | 0.55 | $9.1_{\pm3}$ | 0.40 | $10.0_{\pm4}$ |
| | | GIES* | 0.76 | $3.6_{\pm2}$ | 0.49 | $6.0_{\pm5}$ | 0.69 | $4.8_{\pm2}$ | 0.67 | $5.9_{\pm3}$ | 0.63 | $7.1_{\pm3}$ |
| | | SEA (FCI) | 0.93 | $\mathbf{1.0}_{\pm1}$ | 0.82 | $3.0_{\pm4}$ | 0.90 | $3.8_{\pm2}$ | 0.73 | $\mathbf{3.9}_{\pm2}$ | 0.70 | $6.1_{\pm3}$ |
| | | SEA (GIES) | **0.94** | $1.2_{\pm1}$ | **0.88** | $2.6_{\pm4}$ | **0.93** | $3.2_{\pm1}$ | **0.84** | $4.0_{\pm3}$ | **0.79** | $\mathbf{5.8}_{\pm3}$ |
| 10 | 40 | DCDI-G | 0.50 | $19.8_{\pm2}$ | 0.64 | $17.4_{\pm3}$ | 0.82 | $8.0_{\pm2}$ | 0.25 | $30.8_{\pm4}$ | 0.23 | $31.0_{\pm2}$ |
| | | DCDI-DSF | 0.48 | $19.8_{\pm3}$ | 0.55 | $21.6_{\pm7}$ | 0.87 | $\mathbf{6.0}_{\pm3}$ | 0.25 | $31.4_{\pm3}$ | 0.25 | $30.0_{\pm3}$ |
| | | DCD-FG | 0.35 | $28.6_{\pm4}$ | 0.40 | $26.0_{\pm1}$ | 0.36 | $26.8_{\pm4}$ | 0.45 | $24.4_{\pm5}$ | 0.41 | $25.5_{\pm4}$ |
| | | DIFFAN | 0.41 | $27.6_{\pm4}$ | 0.29 | $33.4_{\pm4}$ | 0.49 | $26.4_{\pm4}$ | 0.38 | $29.4_{\pm6}$ | 0.32 | $31.8_{\pm4}$ |
| | | DECI | 0.43 | $27.6_{\pm5}$ | 0.55 | $22.4_{\pm4}$ | 0.50 | $25.4_{\pm3}$ | 0.54 | $\mathbf{21.5}_{\pm3}$ | 0.57 | $\mathbf{18.4}_{\pm3}$ |
| | | AVICI | 0.44 | $33.0_{\pm5}$ | 0.68 | $28.4_{\pm7}$ | 0.85 | $20.0_{\pm4}$ | 0.50 | $35.0_{\pm3}$ | 0.50 | $38.8_{\pm2}$ |
| | | VARSORT* | 0.76 | $21.8_{\pm5}$ | 0.81 | $15.4_{\pm4}$ | 0.61 | $24.0_{\pm4}$ | 0.67 | $33.6_{\pm5}$ | 0.46 | $37.6_{\pm2}$ |
| | | INVCOV | — | $40.2_{\pm2}$ | — | $44.6_{\pm2}$ | — | $41.0_{\pm3}$ | — | $44.2_{\pm3}$ | — | $42.3_{\pm2}$ |
| | | FCI* | 0.16 | $39.2_{\pm2}$ | 0.24 | $38.8_{\pm3}$ | 0.22 | $36.2_{\pm3}$ | 0.22 | $38.8_{\pm2}$ | 0.16 | $39.9_{\pm2}$ |
| | | GIES* | 0.44 | $33.4_{\pm4}$ | 0.60 | $33.0_{\pm5}$ | 0.51 | $32.0_{\pm3}$ | 0.38 | $36.7_{\pm2}$ | 0.63 | $34.6_{\pm3}$ |
| | | SEA (FCI) | 0.80 | $18.6_{\pm2}$ | 0.89 | $15.4_{\pm4}$ | 0.87 | $18.8_{\pm4}$ | 0.78 | $24.4_{\pm6}$ | 0.69 | $26.9_{\pm4}$ |
| | | SEA (GIES) | **0.91** | $\mathbf{12.8}_{\pm4}$ | **0.95** | $\mathbf{10.4}_{\pm6}$ | **0.89** | $17.2_{\pm3}$ | **0.87** | $24.5_{\pm3}$ | **0.93** | $29.5_{\pm3}$ |
| 20 | 20 | DCDI-G | 0.75 | $6.4_{\pm2}$ | 0.90 | $3.0_{\pm2}$ | 0.84 | $\mathbf{4.4}_{\pm2}$ | 0.39 | $42.7_{\pm6}$ | 0.48 | $10.4_{\pm3}$ |
| | | DCDI-DSF | 0.77 | $5.2_{\pm3}$ | 0.85 | $4.2_{\pm4}$ | 0.64 | $11.6_{\pm3}$ | 0.41 | $43.2_{\pm6}$ | 0.45 | $15.7_{\pm5}$ |
| | | DCD-FG | 0.76 | $51.2_{\pm12}$ | 0.88 | $177_{\pm57}$ | 0.85 | $193_{\pm27}$ | 0.65 | $251_{\pm35}$ | 0.60 | $52.7_{\pm19}$ |
| | | DIFFAN | 0.56 | $40.2_{\pm27}$ | 0.47 | $38.6_{\pm26}$ | 0.53 | $35.0_{\pm26}$ | 0.63 | $19.2_{\pm8}$ | 0.42 | $49.7_{\pm15}$ |
| | | DECI | 0.54 | $52.0_{\pm17}$ | 0.56 | $41.0_{\pm14}$ | 0.56 | $39.0_{\pm8}$ | 0.65 | $30.0_{\pm9}$ | 0.73 | $18.9_{\pm6}$ |
| | | AVICI | 0.56 | $17.2_{\pm5}$ | 0.69 | $10.8_{\pm2}$ | 0.79 | $11.2_{\pm3}$ | 0.53 | $17.2_{\pm5}$ | 0.37 | $18.4_{\pm4}$ |
| | | VARSORT* | 0.84 | $10.0_{\pm3}$ | 0.88 | $6.6_{\pm6}$ | 0.74 | $14.6_{\pm8}$ | 0.50 | $16.1_{\pm4}$ | 0.40 | $17.1_{\pm4}$ |
| | | INVCOV | — | $23.6_{\pm6}$ | — | $24.6_{\pm5}$ | — | $24.6_{\pm6}$ | — | $22.9_{\pm5}$ | — | $20.0_{\pm5}$ |
| | | FCI* | 0.70 | $19.0_{\pm5}$ | 0.45 | $17.4_{\pm2}$ | 0.50 | $19.4_{\pm6}$ | 0.57 | $18.5_{\pm4}$ | 0.44 | $18.9_{\pm5}$ |
| | | GIES* | 0.80 | $7.4_{\pm2}$ | 0.77 | $9.0_{\pm5}$ | 0.71 | $14.0_{\pm3}$ | 0.75 | $12.5_{\pm4}$ | 0.68 | $13.7_{\pm4}$ |
| | | SEA (FCI) | **0.92** | $3.2_{\pm3}$ | 0.93 | $5.0_{\pm3}$ | 0.94 | $8.8_{\pm3}$ | 0.79 | $\mathbf{6.7}_{\pm3}$ | 0.76 | $\mathbf{9.8}_{\pm4}$ |
| | | SEA (GIES) | 0.89 | $\mathbf{3.0}_{\pm1}$ | **0.95** | $3.4_{\pm2}$ | **0.94** | $7.8_{\pm3}$ | **0.83** | $8.1_{\pm3}$ | **0.78** | $10.1_{\pm4}$ |
| 20 | 80 | DCDI-G | 0.54 | $44.0_{\pm6}$ | 0.53 | $61.6_{\pm11}$ | 0.89 | $37.4_{\pm34}$ | 0.46 | $44.2_{\pm5}$ | 0.26 | $59.7_{\pm5}$ |
| | | DCDI-DSF | 0.57 | $41.2_{\pm3}$ | 0.61 | $\mathbf{60.0}_{\pm12}$ | 0.85 | $\mathbf{28.4}_{\pm26}$ | 0.47 | $43.6_{\pm6}$ | 0.30 | $57.6_{\pm5}$ |
| | | DCD-FG | 0.58 | $172_{\pm27}$ | 0.65 | $156_{\pm41}$ | 0.75 | $162_{\pm49}$ | 0.62 | $80.1_{\pm13}$ | 0.48 | $79.8_{\pm7}$ |
| | | DIFFAN | 0.46 | $127_{\pm5}$ | 0.28 | $154_{\pm10}$ | 0.36 | $145_{\pm7}$ | 0.45 | $117_{\pm21}$ | 0.21 | $157_{\pm7}$ |
| | | DECI | 0.30 | $87.2_{\pm3}$ | 0.47 | $104_{\pm7}$ | 0.35 | $79.6_{\pm9}$ | 0.48 | $71.0_{\pm7}$ | 0.57 | $58.9_{\pm11}$ |
| | | AVICI | 0.51 | $75.6_{\pm10}$ | 0.69 | $72.8_{\pm6}$ | 0.69 | $61.2_{\pm10}$ | 0.50 | $70.6_{\pm6}$ | 0.50 | $75.5_{\pm7}$ |
| | | VARSORT* | 0.82 | $44.8_{\pm4}$ | 0.84 | $73.6_{\pm13}$ | 0.61 | $65.2_{\pm10}$ | 0.67 | $63.4_{\pm5}$ | 0.39 | $75.6_{\pm5}$ |
| | | INVCOV | — | $97.6_{\pm6}$ | — | $121_{\pm4}$ | — | $104_{\pm9}$ | — | $95.6_{\pm7}$ | — | $97.8_{\pm4}$ |
| | | FCI* | 0.19 | $75.8_{\pm11}$ | 0.19 | $80.2_{\pm5}$ | 0.22 | $74.4_{\pm8}$ | 0.33 | $72.3_{\pm6}$ | 0.23 | $76.6_{\pm5}$ |
| | | GIES* | 0.56 | $70.0_{\pm11}$ | 0.73 | $75.2_{\pm4}$ | 0.59 | $67.4_{\pm7}$ | 0.62 | $65.6_{\pm7}$ | 0.61 | $68.1_{\pm5}$ |
| | | SEA (FCI) | 0.86 | $39.8_{\pm12}$ | 0.87 | $73.8_{\pm12}$ | **0.92** | $52.0_{\pm9}$ | 0.82 | $\mathbf{42.9}_{\pm6}$ | 0.69 | $\mathbf{57.0}_{\pm5}$ |
| | | SEA (GIES) | **0.92** | $\mathbf{26.8}_{\pm8}$ | **0.88** | $71.4_{\pm8}$ | **0.92** | $50.6_{\pm7}$ | **0.84** | $45.0_{\pm7}$ | **0.89** | $60.1_{\pm6}$ |

Table 21: Full results on synthetic datasets (continuous metrics). Mean over 5 distinct scale-free graphs. † indicates o.o.d setting. ∗ indicates non-parametric bootstrapping. All standard deviations were within 0.02.

| N | E | Model | Linear | | NN add. | | NN non-add. | | Sigmoid† | | Polynomial† | |
|---|---|---|---|---|---|---|---|---|---|---|---|---|
| | | | mAP ↑ | AUC ↑ | mAP ↑ | AUC ↑ | mAP ↑ | AUC ↑ | mAP ↑ | AUC ↑ | mAP ↑ | AUC ↑ |
| 10 | 10 | Dcdi-G | 0.54 | 0.90 | 0.59 | 0.88 | 0.69 | 0.89 | 0.48 | 0.77 | 0.50 | 0.73 |
| | | Dcdi-Dsf | 0.70 | 0.92 | 0.71 | 0.88 | 0.36 | 0.83 | 0.46 | 0.75 | 0.49 | 0.76 |
| | | Dcd-Fg | 0.56 | 0.76 | 0.47 | 0.72 | 0.50 | 0.73 | 0.44 | 0.68 | 0.57 | 0.75 |
| | | DiffAn | 0.25 | 0.73 | 0.15 | 0.66 | 0.16 | 0.62 | 0.31 | 0.75 | 0.24 | 0.63 |
| | | Deci | 0.17 | 0.65 | 0.17 | 0.67 | 0.20 | 0.72 | 0.27 | 0.73 | 0.49 | 0.82 |
| | | Avici | 0.51 | 0.87 | 0.55 | 0.85 | 0.76 | 0.95 | 0.44 | 0.81 | 0.27 | 0.71 |
| | | VarSort* | 0.67 | 0.84 | 0.69 | 0.86 | 0.76 | 0.88 | 0.45 | 0.69 | 0.46 | 0.73 |
| | | InvCov | 0.50 | 0.92 | 0.41 | 0.87 | 0.38 | 0.84 | 0.47 | 0.90 | 0.45 | 0.86 |
| | | Fci* | 0.56 | 0.80 | 0.51 | 0.80 | 0.43 | 0.74 | 0.60 | 0.82 | 0.34 | 0.68 |
| | | Gies* | 0.87 | 0.98 | 0.61 | 0.94 | 0.69 | 0.94 | 0.75 | 0.96 | 0.71 | **0.91** |
| | | Sea (Fci) | **0.96** | **0.99** | 0.88 | 0.98 | 0.88 | 0.97 | 0.79 | 0.96 | 0.73 | 0.90 |
| | | Sea (Gies) | 0.95 | **0.99** | **0.94** | **0.98** | **0.92** | **0.98** | **0.85** | **0.98** | **0.74** | 0.90 |
| 10 | 40 | Dcdi-G | 0.70 | 0.85 | 0.74 | 0.85 | **0.88** | **0.91** | 0.56 | 0.66 | 0.53 | 0.64 |
| | | Dcdi-Dsf | 0.74 | 0.87 | 0.73 | 0.84 | 0.71 | 0.90 | 0.56 | 0.69 | 0.51 | 0.63 |
| | | Dcd-Fg | 0.37 | 0.58 | 0.45 | 0.61 | 0.45 | 0.58 | 0.49 | 0.63 | 0.63 | 0.73 |
| | | DiffAn | 0.29 | 0.50 | 0.25 | 0.38 | 0.28 | 0.46 | 0.31 | 0.53 | 0.27 | 0.44 |
| | | Deci | 0.30 | 0.51 | 0.41 | 0.65 | 0.33 | 0.51 | 0.38 | 0.60 | 0.59 | 0.77 |
| | | Avici | 0.41 | 0.57 | 0.65 | 0.81 | 0.55 | 0.67 | 0.41 | 0.59 | 0.40 | 0.61 |
| | | VarSort* | 0.77 | 0.83 | 0.74 | 0.87 | 0.59 | 0.71 | 0.66 | 0.76 | 0.50 | 0.66 |
| | | InvCov | 0.44 | 0.71 | 0.38 | 0.59 | 0.42 | 0.62 | 0.44 | 0.67 | 0.42 | 0.62 |
| | | Fci* | 0.47 | 0.64 | 0.41 | 0.60 | 0.40 | 0.58 | 0.48 | 0.64 | 0.41 | 0.59 |
| | | Gies* | 0.43 | 0.68 | 0.43 | 0.63 | 0.44 | 0.61 | 0.49 | 0.69 | 0.59 | 0.71 |
| | | Sea (Fci) | 0.85 | 0.91 | 0.80 | 0.90 | 0.77 | 0.88 | 0.76 | 0.83 | 0.66 | 0.71 |
| | | Sea (Gies) | **0.92** | **0.96** | **0.84** | **0.93** | 0.83 | 0.90 | **0.79** | **0.88** | **0.78** | **0.87** |
| 20 | 20 | Dcdi-G | 0.41 | 0.95 | 0.50 | 0.94 | 0.69 | 0.96 | 0.37 | 0.83 | 0.37 | 0.77 |
| | | Dcdi-Dsf | 0.48 | 0.95 | 0.55 | 0.93 | 0.33 | 0.90 | 0.37 | 0.79 | 0.35 | 0.82 |
| | | Dcd-Fg | 0.51 | 0.87 | 0.39 | 0.83 | 0.48 | 0.84 | 0.56 | 0.84 | 0.50 | 0.84 |
| | | DiffAn | 0.27 | 0.80 | 0.11 | 0.65 | 0.11 | 0.66 | 0.26 | 0.77 | 0.12 | 0.69 |
| | | Deci | 0.13 | 0.69 | 0.15 | 0.71 | 0.15 | 0.73 | 0.15 | 0.71 | 0.25 | 0.79 |
| | | Avici | 0.53 | 0.88 | 0.66 | 0.89 | 0.74 | 0.92 | 0.46 | 0.87 | 0.32 | 0.77 |
| | | VarSort* | 0.67 | 0.85 | 0.84 | 0.93 | 0.59 | 0.86 | 0.45 | 0.72 | 0.44 | 0.73 |
| | | InvCov | 0.44 | 0.94 | 0.35 | 0.91 | 0.30 | 0.89 | 0.43 | 0.93 | 0.41 | 0.87 |
| | | Fci* | 0.63 | 0.84 | 0.44 | 0.78 | 0.43 | 0.79 | 0.60 | 0.86 | 0.47 | 0.78 |
| | | Gies* | 0.82 | 0.99 | 0.58 | 0.95 | 0.57 | 0.96 | 0.75 | 0.98 | 0.61 | 0.90 |
| | | Sea (Fci) | **0.94** | **1.00** | 0.87 | **0.98** | 0.83 | 0.97 | **0.82** | **0.98** | **0.72** | **0.92** |
| | | Sea (Gies) | 0.93 | **1.00** | **0.91** | 0.98 | **0.88** | **0.98** | **0.82** | 0.98 | 0.70 | 0.91 |
| 20 | 80 | Dcdi-G | 0.62 | 0.88 | 0.61 | **0.89** | **0.76** | **0.94** | 0.44 | 0.76 | 0.36 | 0.60 |
| | | Dcdi-Dsf | 0.58 | 0.87 | 0.55 | 0.86 | 0.58 | 0.92 | 0.43 | 0.78 | 0.35 | 0.66 |
| | | Dcd-Fg | 0.38 | 0.70 | 0.30 | 0.69 | 0.48 | 0.80 | 0.48 | 0.75 | 0.53 | 0.73 |
| | | DiffAn | 0.18 | 0.55 | 0.15 | 0.44 | 0.16 | 0.53 | 0.19 | 0.56 | 0.15 | 0.38 |
| | | Deci | 0.21 | 0.58 | 0.24 | 0.64 | 0.26 | 0.66 | 0.30 | 0.68 | 0.41 | 0.75 |
| | | Avici | 0.29 | 0.61 | 0.54 | 0.80 | 0.60 | 0.85 | 0.35 | 0.65 | 0.28 | 0.63 |
| | | VarSort* | 0.79 | 0.90 | 0.57 | 0.83 | 0.62 | 0.81 | 0.57 | 0.77 | 0.38 | 0.64 |
| | | InvCov | 0.38 | 0.81 | 0.22 | 0.56 | 0.32 | 0.73 | 0.38 | 0.78 | 0.33 | 0.67 |
| | | Fci* | 0.31 | 0.63 | 0.30 | 0.62 | 0.30 | 0.62 | 0.41 | 0.68 | 0.32 | 0.62 |
| | | Gies* | 0.51 | 0.87 | 0.43 | 0.78 | 0.47 | 0.81 | 0.52 | 0.82 | 0.47 | 0.73 |
| | | Sea (Fci) | 0.87 | 0.96 | 0.59 | 0.87 | 0.70 | 0.90 | 0.73 | 0.87 | 0.53 | 0.71 |
| | | Sea (Gies) | **0.92** | **0.98** | **0.63** | 0.89 | 0.73 | 0.91 | **0.77** | **0.92** | **0.62** | **0.84** |

Table 22: Full results on synthetic datasets (discrete metrics). Mean/std over 5 distinct scale-free graphs. † indicates o.o.d. setting. ∗ indicates non-parametric bootstrapping. All OA standard deviations within 0.2.

| N E | Model | Linear | | NN add. | | NN non-add. | | Sigmoid† | | Polynomial† | |
|---|---|---|---|---|---|---|---|---|---|---|---|
| | | OA ↑ | SHD ↓ | OA ↑ | SHD ↓ | OA ↑ | SHD ↓ | OA ↑ | SHD ↓ | OA ↑ | SHD ↓ |
| 10 10 | Dcdi-G | 0.51 | $16.6_{\pm1}$ | 0.61 | $17.4_{\pm2}$ | 0.71 | $16.2_{\pm2}$ | 0.59 | $16.9_{\pm5}$ | 0.63 | $16.6_{\pm3}$ |
| | Dcdi-Dsf | 0.70 | $16.2_{\pm2}$ | 0.75 | $15.4_{\pm3}$ | 0.36 | $16.8_{\pm2}$ | 0.48 | $18.1_{\pm3}$ | 0.67 | $18.0_{\pm2}$ |
| | Dcd-Fg | 0.60 | $16.4_{\pm5}$ | 0.57 | $22.2_{\pm4}$ | 0.57 | $20.0_{\pm4}$ | 0.47 | $17.9_{\pm4}$ | 0.59 | $16.8_{\pm5}$ |
| | DiffAn | 0.55 | $9.2_{\pm4}$ | 0.49 | $14.6_{\pm4}$ | 0.36 | $11.6_{\pm4}$ | 0.59 | $7.7_{\pm4}$ | 0.42 | $14.8_{\pm6}$ |
| | Deci | 0.51 | $17.4_{\pm5}$ | 0.55 | $17.4_{\pm5}$ | 0.58 | $12.0_{\pm2}$ | 0.62 | $12.6_{\pm5}$ | 0.74 | $8.2_{\pm6}$ |
| | Avici | 0.64 | $6.8_{\pm3}$ | 0.59 | $5.4_{\pm3}$ | 0.77 | $3.6_{\pm1}$ | 0.48 | $7.8_{\pm3}$ | 0.35 | $9.5_{\pm3}$ |
| | VarSort* | 0.82 | $4.4_{\pm1}$ | 0.78 | $4.8_{\pm2}$ | 0.80 | $3.4_{\pm2}$ | 0.47 | $7.3_{\pm3}$ | 0.55 | $8.1_{\pm3}$ |
| | InvCov | — | $8.8_{\pm3}$ | — | $9.2_{\pm2}$ | — | $9.8_{\pm1}$ | — | $9.7_{\pm3}$ | — | $10.2_{\pm2}$ |
| | Fci* | 0.62 | $8.4_{\pm2}$ | 0.51 | $7.8_{\pm2}$ | 0.45 | $8.0_{\pm2}$ | 0.61 | $9.2_{\pm3}$ | 0.34 | $9.6_{\pm2}$ |
| | Gies* | 0.83 | $2.2_{\pm2}$ | 0.60 | $6.0_{\pm2}$ | 0.75 | $4.2_{\pm2}$ | 0.72 | $4.9_{\pm3}$ | 0.73 | $5.7_{\pm2}$ |
| | Sea (Fci) | **0.89** | $\mathbf{1.4}_{\pm2}$ | 0.90 | $2.6_{\pm2}$ | 0.93 | $2.2_{\pm1}$ | 0.71 | $4.0_{\pm3}$ | 0.72 | $5.2_{\pm2}$ |
| | Sea (Gies) | 0.85 | $\mathbf{1.4}_{\pm1}$ | **0.96** | $\mathbf{1.8}_{\pm1}$ | **0.94** | $\mathbf{2.0}_{\pm1}$ | **0.86** | $\mathbf{3.4}_{\pm3}$ | **0.83** | $5.0_{\pm2}$ |
| 10 40 | Dcdi-G | 0.82 | $24.0_{\pm4}$ | 0.85 | $27.8_{\pm5}$ | 0.87 | $19.6_{\pm2}$ | 0.62 | $31.4_{\pm3}$ | 0.52 | $32.6_{\pm4}$ |
| | Dcdi-Dsf | 0.79 | $22.8_{\pm5}$ | 0.79 | $24.4_{\pm4}$ | 0.82 | $20.4_{\pm2}$ | 0.64 | $31.6_{\pm2}$ | 0.58 | $33.3_{\pm3}$ |
| | Dcd-Fg | 0.36 | $24.8_{\pm4}$ | 0.41 | $25.2_{\pm5}$ | 0.38 | $25.6_{\pm8}$ | 0.41 | $23.2_{\pm6}$ | 0.54 | $18.5_{\pm3}$ |
| | DiffAn | 0.40 | $29.8_{\pm8}$ | 0.28 | $37.0_{\pm2}$ | 0.38 | $32.6_{\pm2}$ | 0.45 | $28.0_{\pm6}$ | 0.33 | $32.7_{\pm5}$ |
| | Deci | 0.43 | $27.8_{\pm3}$ | 0.66 | $22.6_{\pm3}$ | 0.48 | $28.6_{\pm3}$ | 0.52 | $22.2_{\pm4}$ | 0.66 | $\mathbf{13.3}_{\pm3}$ |
| | Avici | 0.43 | $20.2_{\pm4}$ | 0.84 | $17.2_{\pm4}$ | 0.61 | $20.4_{\pm8}$ | 0.52 | $24.8_{\pm3}$ | 0.49 | $26.5_{\pm2}$ |
| | VarSort* | 0.75 | $13.4_{\pm2}$ | 0.89 | $\mathbf{12.6}_{\pm2}$ | 0.60 | $20.6_{\pm3}$ | 0.65 | $20.8_{\pm4}$ | 0.50 | $26.4_{\pm2}$ |
| | InvCov | — | $31.4_{\pm3}$ | — | $37.2_{\pm4}$ | — | $36.0_{\pm4}$ | — | $32.3_{\pm5}$ | — | $34.8_{\pm3}$ |
| | Fci* | 0.33 | $23.8_{\pm2}$ | 0.28 | $27.2_{\pm2}$ | 0.25 | $27.6_{\pm4}$ | 0.36 | $26.1_{\pm2}$ | 0.24 | $27.3_{\pm2}$ |
| | Gies* | 0.46 | $21.8_{\pm3}$ | 0.50 | $24.4_{\pm2}$ | 0.48 | $25.2_{\pm5}$ | 0.52 | $22.7_{\pm3}$ | 0.64 | $22.5_{\pm2}$ |
| | Sea (Fci) | 0.80 | $10.0_{\pm4}$ | 0.88 | $14.0_{\pm2}$ | 0.87 | $15.8_{\pm3}$ | 0.79 | $15.6_{\pm3}$ | 0.64 | $18.5_{\pm3}$ |
| | Sea (Gies) | **0.88** | $\mathbf{6.6}_{\pm3}$ | **0.98** | $14.0_{\pm5}$ | **0.88** | $\mathbf{14.4}_{\pm3}$ | **0.87** | $\mathbf{14.1}_{\pm3}$ | **0.93** | $19.1_{\pm3}$ |
| 20 20 | Dcdi-G | 0.54 | $40.4_{\pm2}$ | 0.70 | $44.8_{\pm8}$ | 0.88 | $39.8_{\pm6}$ | 0.47 | $41.1_{\pm4}$ | 0.53 | $38.4_{\pm6}$ |
| | Dcdi-Dsf | 0.64 | $40.4_{\pm3}$ | 0.65 | $42.4_{\pm8}$ | 0.40 | $42.2_{\pm8}$ | 0.37 | $41.1_{\pm4}$ | 0.45 | $49.3_{\pm18}$ |
| | Dcd-Fg | 0.68 | $252_{\pm23}$ | 0.77 | $183_{\pm52}$ | 0.78 | $181_{\pm27}$ | 0.70 | $251_{\pm45}$ | 0.69 | $278_{\pm68}$ |
| | DiffAn | 0.67 | $23.6_{\pm13}$ | 0.40 | $42.2_{\pm22}$ | 0.42 | $34.0_{\pm11}$ | 0.59 | $22.6_{\pm12}$ | 0.50 | $46.8_{\pm13}$ |
| | Deci | 0.50 | $42.0_{\pm5}$ | 0.54 | $43.0_{\pm11}$ | 0.57 | $40.0_{\pm13}$ | 0.51 | $34.7_{\pm7}$ | 0.65 | $25.3_{\pm6}$ |
| | Avici | 0.68 | $11.8_{\pm3}$ | 0.79 | $9.2_{\pm2}$ | 0.80 | $7.8_{\pm2}$ | 0.60 | $15.3_{\pm4}$ | 0.45 | $18.3_{\pm5}$ |
| | VarSort* | 0.74 | $10.4_{\pm3}$ | 0.91 | $6.2_{\pm2}$ | 0.76 | $13.0_{\pm6}$ | 0.49 | $13.7_{\pm4}$ | 0.51 | $16.7_{\pm6}$ |
| | InvCov | — | $20.2_{\pm3}$ | — | $24.4_{\pm5}$ | — | $23.8_{\pm5}$ | — | $21.2_{\pm4}$ | — | $20.8_{\pm5}$ |
| | Fci* | 0.67 | $13.8_{\pm2}$ | 0.52 | $17.4_{\pm1}$ | 0.53 | $17.8_{\pm5}$ | 0.65 | $16.7_{\pm4}$ | 0.50 | $18.9_{\pm5}$ |
| | Gies* | 0.82 | $6.4_{\pm3}$ | 0.71 | $12.6_{\pm2}$ | 0.68 | $13.2_{\pm3}$ | 0.75 | $11.4_{\pm5}$ | 0.73 | $13.4_{\pm4}$ |
| | Sea (Fci) | **0.90** | $\mathbf{2.8}_{\pm1}$ | 0.87 | $7.0_{\pm2}$ | 0.91 | $8.2_{\pm5}$ | 0.73 | $\mathbf{7.7}_{\pm3}$ | 0.71 | $\mathbf{9.5}_{\pm4}$ |
| | Sea (Gies) | 0.85 | $4.0_{\pm2}$ | **0.95** | $\mathbf{3.6}_{\pm2}$ | **0.93** | $\mathbf{6.2}_{\pm4}$ | **0.80** | $7.9_{\pm4}$ | **0.82** | $9.9_{\pm3}$ |
| 20 80 | Dcdi-G | 0.81 | $93.0_{\pm10}$ | 0.73 | $104_{\pm7}$ | **0.91** | $67.8_{\pm8}$ | 0.61 | $82.5_{\pm8}$ | 0.53 | $79.7_{\pm5}$ |
| | Dcdi-Dsf | 0.75 | $103_{\pm7}$ | 0.73 | $94.8_{\pm11}$ | 0.77 | $63.8_{\pm7}$ | 0.65 | $84.4_{\pm8}$ | 0.52 | $82.6_{\pm5}$ |
| | Dcd-Fg | 0.63 | $188_{\pm24}$ | 0.70 | $187_{\pm14}$ | 0.78 | $190_{\pm26}$ | 0.71 | $217_{\pm38}$ | 0.71 | $235_{\pm32}$ |
| | DiffAn | 0.42 | $111_{\pm18}$ | 0.30 | $145_{\pm11}$ | 0.40 | $119_{\pm13}$ | 0.41 | $100_{\pm22}$ | 0.21 | $149_{\pm11}$ |
| | Deci | 0.33 | $72.2_{\pm10}$ | 0.48 | $81.4_{\pm10}$ | 0.48 | $67.0_{\pm9}$ | 0.50 | $60.4_{\pm12}$ | 0.58 | $\mathbf{47.2}_{\pm7}$ |
| | Avici | 0.45 | $56.0_{\pm5}$ | 0.77 | $\mathbf{47.6}_{\pm8}$ | 0.76 | $42.6_{\pm3}$ | 0.50 | $54.0_{\pm6}$ | 0.47 | $62.2_{\pm5}$ |
| | VarSort* | 0.85 | $32.4_{\pm5}$ | 0.84 | $54.8_{\pm8}$ | 0.75 | $40.2_{\pm6}$ | 0.64 | $54.0_{\pm6}$ | 0.37 | $60.1_{\pm5}$ |
| | InvCov | — | $79.4_{\pm6}$ | — | $106_{\pm5}$ | — | $87.4_{\pm2}$ | — | $78.8_{\pm9}$ | — | $86.2_{\pm4}$ |
| | Fci* | 0.26 | $58.2_{\pm6}$ | 0.22 | $58.4_{\pm4}$ | 0.26 | $55.0_{\pm7}$ | 0.37 | $59.3_{\pm6}$ | 0.23 | $62.9_{\pm4}$ |
| | Gies* | 0.65 | $48.4_{\pm5}$ | 0.71 | $53.6_{\pm3}$ | 0.68 | $48.0_{\pm4}$ | 0.64 | $53.6_{\pm6}$ | 0.61 | $54.9_{\pm5}$ |
| | Sea (Fci) | 0.91 | $25.6_{\pm6}$ | 0.88 | $51.6_{\pm6}$ | 0.89 | $42.2_{\pm4}$ | 0.83 | $36.1_{\pm6}$ | 0.71 | $47.4_{\pm5}$ |
| | Sea (Gies) | **0.92** | $\mathbf{17.6}_{\pm3}$ | **0.93** | $49.2_{\pm9}$ | 0.89 | $\mathbf{37.2}_{\pm5}$ | **0.88** | $\mathbf{35.1}_{\pm7}$ | **0.89** | $48.1_{\pm5}$ |

Table 23: Causal discovery results on synthetic datasets with 100 nodes, continuous metrics. Each setting encompasses 5 distinct Erdős-Rényi graphs. The symbol † indicates that the model was not trained on this setting. All standard deviations were within 0.1.

| N | E | Model | Linear | | NN add. | | NN non-add. | | Sigmoid† | | Polynomial† | |
|---|---|---|---|---|---|---|---|---|---|---|---|---|
| | | | mAP ↑ | AUC ↑ | mAP ↑ | AUC ↑ | mAP ↑ | AUC ↑ | mAP ↑ | AUC ↑ | mAP ↑ | AUC ↑ |
| 100 | 100 | DCD-FG | 0.11 | 0.75 | 0.12 | 0.71 | 0.18 | 0.73 | 0.20 | 0.72 | 0.06 | 0.60 |
| | | INVCOV | 0.40 | 0.99 | 0.22 | 0.94 | 0.16 | 0.87 | 0.40 | 0.97 | 0.36 | 0.90 |
| | | SEA (FCI) | 0.96 | 1.00 | 0.83 | 0.97 | 0.75 | 0.97 | 0.79 | 0.97 | 0.56 | 0.88 |
| | | SEA (GIES) | 0.97 | 1.00 | 0.82 | 0.98 | 0.74 | 0.96 | 0.80 | 0.97 | 0.54 | 0.85 |
| 100 | 400 | DCD-FG | 0.05 | 0.59 | 0.07 | 0.64 | 0.10 | 0.72 | 0.13 | 0.72 | 0.12 | 0.64 |
| | | INVCOV | 0.25 | 0.91 | 0.09 | 0.62 | 0.14 | 0.77 | 0.27 | 0.86 | 0.20 | 0.67 |
| | | SEA (FCI) | 0.90 | 0.99 | 0.28 | 0.82 | 0.60 | 0.92 | 0.69 | 0.92 | 0.38 | 0.80 |
| | | SEA (GIES) | 0.91 | 0.99 | 0.27 | 0.82 | 0.61 | 0.92 | 0.69 | 0.91 | 0.38 | 0.78 |

Table 24: Causal discovery results on synthetic datasets with 100 nodes, discrete metrics. Each setting encompasses 5 distinct Erdős-Rényi graphs. The symbol † indicates that the model was not trained on this setting.

| N | E | Model | Linear | | NN add. | | NN non-add. | | Sigmoid† | | Polynomial† | |
|---|---|---|---|---|---|---|---|---|---|---|---|---|
| | | | OA ↑ | SHD ↓ | OA ↑ | SHD ↓ | OA ↑ | SHD ↓ | OA ↑ | SHD ↓ | OA ↑ | SHD ↓ |
| 100 | 100 | DCD-FG | 0.63 | 3075.8 | 0.58 | 2965.0 | 0.60 | 2544.4 | 0.59 | 3808.0 | 0.34 | 1927.9 |
| | | INVCOV | — | 124.4 | — | 130.0 | — | 158.8 | — | 112.3 | — | 106.3 |
| | | SEA (FCI) | 0.91 | 13.4 | 0.90 | 34.4 | 0.91 | 47.2 | 0.78 | 40.3 | 0.69 | 59.2 |
| | | SEA (GIES) | 0.91 | 13.6 | 0.93 | 32.8 | 0.91 | 45.8 | 0.78 | 38.6 | 0.68 | 60.3 |
| 100 | 400 | DCD-FG | 0.46 | 3068.2 | 0.60 | 3428.8 | 0.70 | 3510.8 | 0.67 | 3601.8 | 0.53 | 3316.7 |
| | | INVCOV | — | 557.0 | — | 667.8 | — | 639.0 | — | 514.7 | — | 539.4 |
| | | SEA (FCI) | 0.93 | 122.0 | 0.90 | 361.2 | 0.91 | 273.2 | 0.87 | 226.9 | 0.82 | 327.0 |
| | | SEA (GIES) | 0.94 | 116.6 | 0.91 | 364.4 | 0.92 | 266.8 | 0.87 | 218.3 | 0.84 | 328.0 |

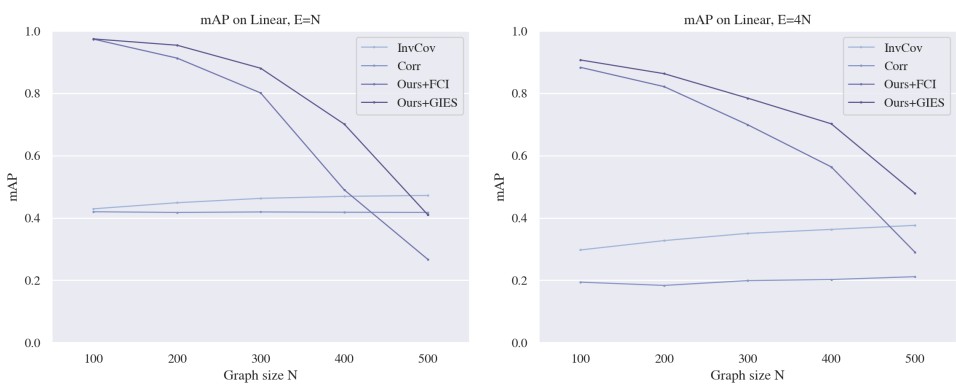

Figure 10: mAP on graphs larger than seen during training. During training, we only sampled a maximum of 100 subsets, so performance drop may be due to extrapolation beyond trained embeddings. We did not have time to finetune these embeddings for more samples. These values correspond to the numbers in Table 16.

Table 25: Swapping to estimation algorithms with significantly different assumptions (LiNGAM) leads to a larger performance drop. Interestingly, the GIES model seems to depend more on marginal estimates, while the FCI depends more on global statistics. Results on $N = 10, E = 10$ observational setting.

| Train | Inference | Linear | | NN add. | | NN non-add. | | Sigmoid[†] | | Polynomial[†] | |
|---|---|---|---|---|---|---|---|---|---|---|---|
| | | mAP ↑ | SHD ↓ | mAP ↑ | SHD ↓ | mAP ↑ | SHD ↓ | mAP ↑ | SHD ↓ | mAP ↑ | SHD ↓ |
| FCI | FCI | 0.96±.03 | 3.2±.6 | 0.91±.04 | 5.0±.8 | 0.82±.05 | 8.8±.9 | 0.85±.09 | 6.7±.1 | 0.69±.09 | 9.8±.2 |
| | PC | 0.95±.04 | 3.6±.4 | 0.91±.05 | 4.6±.6 | 0.81±.06 | 10.0±.3 | 0.84±.07 | 7.4±.2 | 0.65±.14 | 10.8±.5 |
| | GES | 0.94±.05 | 4.4±.3 | 0.91±.05 | 4.2±.6 | 0.81±.06 | 9.6±.6 | 0.81±.10 | 8.4±.9 | 0.61±.16 | 11.5±.2 |
| | GRaSP | 0.94±.05 | 4.0±.0 | 0.91±.05 | 4.4±.9 | 0.81±.06 | 10.0±.0 | 0.81±.10 | 8.5±.0 | 0.61±.16 | 11.5±.2 |
| | LiNGAM | 0.88±.08 | 8.0±.5 | 0.84±.05 | 6.0±.3 | 0.74±.05 | 10.8±.3 | 0.71±.06 | 12.5±.5 | 0.59±.12 | 14.4±.6 |
| GIES | PC | 0.96±.02 | 1.8±.7 | 0.91±.05 | 2.8±.4 | 0.89±.10 | 3.2±.2 | 0.82±.14 | 4.1±.5 | 0.58±.20 | 6.7±.3 |
| | GES | 0.95±.03 | 2.0±.9 | 0.91±.05 | 2.6±.1 | 0.88±.11 | 3.4±.5 | 0.81±.15 | 4.1±.5 | 0.57±.19 | 6.8±.1 |
| | GRaSP | 0.95±.03 | 1.8±.7 | 0.92±.05 | 3.0±.8 | 0.88±.11 | 3.2±.2 | 0.81±.15 | 4.0±.4 | 0.57±.19 | 6.9±.0 |
| | LiNGAM | 0.60±.14 | 5.6±.8 | 0.40±.23 | 9.0±.6 | 0.40±.15 | 9.4±.2 | 0.53±.15 | 7.2±.0 | 0.51±.14 | 7.9±.0 |

Table 26: Complete results on Sachs flow cytometry dataset (Sachs et al., 2005), using the subset proposed by (Wang et al., 2017).

| Model | mAP ↑ | AUC ↑ | SHD ↓ |
|---|---|---|---|
| Dcdi-G | 0.17 | 0.55 | 21 |
| Dcdi-Dsf | 0.20 | 0.59 | 20 |
| Dcd-Fg | 0.32 | 0.59 | 27 |
| DiffAn | 0.14 | 0.45 | 37 |
| Deci | 0.21 | 0.62 | 28 |
| Avici-L | 0.35 | 0.78 | 20 |
| Avici-R | 0.29 | 0.65 | 18 |
| Avici-L+R | **0.59** | **0.83** | 14 |
| Fci* | 0.27 | 0.59 | 18 |
| Gies* | 0.21 | 0.59 | 17 |
| Sea (Fci) | 0.23 | 0.54 | 24 |
| +Kci | 0.33 | 0.63 | 14 |
| +Corr | 0.41 | 0.70 | 15 |
| +Kci+Corr | 0.49 | 0.71 | **13** |
| Sea (Gies) | 0.23 | 0.60 | 14 |

