# OpenReview forum: "Sample, estimate, aggregate: A recipe for causal discovery foundation models"
_TMLR — Accepted by TMLR_

### Review · Reviewer_gjMn · 2024-12-16

**Summary Of Contributions:**

This paper introduces SEA (Sample, Estimate, Aggregate), a general framework for causal discovery foundation model that combines the traditional casual discovery methods (e.g. GES, FCI, LiNGAM…) and modern deep learning approaches. This work contributes a foundation model approach that in three steps: (1). Sample subsets from the observational data, (2). Apply classical causal discovery methods to acquire subsets’ graph estimates and acquire global statistics, (3). Aggregate local results via pretraining a neural-network-based aggregator to predict the whole graph.  The authors provide a theoretical analysis showing that the pretrained aggregator could recover the consistent union from local estimates. They conduct comprehensive experiments to demonstrate the generalizability, efficiency, and identifiability of the proposed framework.

**Audience:**

Yes

**Broader Impact Concerns:**

The results generally look good and this work is potentially helpful in casual discovery for scientific discovery. This paper would benefit from a discussion of potential misuse and this framework's limitations in sensitive domains where causal discovery could impact decision-making (e.g., healthcare).

**Claims And Evidence:**

Yes

**Requested Changes:**

The author should consider benchmarking with the classical causal discovery algorithms (PC, GES), especially in large-scale cases where optimization-based methods are usually unstable and ineffective.

**Strengths And Weaknesses:**

**Strength**

* The paper is written in a clear and concise manner, with easy-to-understand illustrative figures.
* The proposed SEA is a general framework that can be combined with classical causal discovery algorithms specializing in different assumptions. It is very efficient and easy to scale for large graphs in that it operates on the subsets of data.
* The empirical validation is very comprehensive and strong. It shows the framework (especially the pretrained aggregator) can be generalized to new data (as well as sim2real, small samples), quick adaptation with finetuning for new assumptions, and consistency with the classic identifiability theory.

**Weakness**

I don’t see critical weaknesses in this work, here are some minor issues and questions.

* The baselines selected here are all deep-learning or optimization-based methods, which are hard to learn and might be unstable. The authors should consider including some typical casual discovery algorithms that are common and stable as baselines (PC, GES).
* The tolerance of changing the subset algorithm in the inference procedure is interesting. Would this be because the algorithms swapped don’t have critical conflicts with the test data properties (LiNGAM would be an extreme one to test)? Or it is because the aggregator is robust enough that the importance of local estimators are not that high?
* The current implementation of independent test and score functions in the SEA are FCI and GIES is Fisher Z and BIC (which are relatively linear and simple), would complicated score functions/independence tests help? Or the aggregator is enough to handle the complexity and local estimators can be simple and naive.

---

> ### Author Response · Authors · 2025-02-04
> **Thank you for your review**
>
> Thank you for your review, and for your questions. Our submission has been updated accordingly, and we include a point-by-point response below.
>
> > benchmarking classical causal discovery algorithms
>
> Thank you for the recommendation. We report on GIES (interventional version of GES) in Table 2, which is our largest dataset, at $N=622$. GIES actually does outperform the deep learning baselines in terms of precision.
>
> On the remaining datasets, we include both GIES and FCI (similar to the PC algorithm) in Tables 1, 17-20. These are not optimized at all, and directly run over all variables. VarSort is also a linear regression-based approach, which is quite stable and benchmarked throughout these synthetic experiments as well.
>
> > tolerance to changing subset
>
> Both reasons are likely to contribute. The alternative algorithms in Table 7 all assume linear Gaussian, so it is likely that they make similar mistakes. In addition, the global statistic is symmetric, so without the local estimators, it's not possible to perform this well. However, "how correct" they need to be is an empirical question.
>
> To probe this, we added Table 23, in which we compare with LiNGAM. LiNGAM has a noticeable drop from the other methods, but the overall performance is still quite respectable for SEA (FCI).
>
> Interestingly, the drop is quite significant for SEA (GIES), suggesting that the latter may rely more on the marginal estimates, while the former relies more on the global statistic. Figures 7 and 8 show that the GIES estimates are much higher quality than the FCI estimates, so it is unsurprising that the GIES model learned to depend on the estimates more, while the FCI model only used them to refine its guesses.
>
> > more complex score functions
>
> We did find that more complex score functions / independence tests help in more complex cases. For example, in Tables 3 and 5, we found that swapping to a kernel-based CI test (KCI) + quadratic kernel was more suitable for real + polynomial/sigmoid data.
>
> The primary reason we did not train with these tests is that KCI scales with the number of samples per batch, so it is much slower to compute.
>
> > discussion regarding potential misuse and limitations
>
> Thank you for bringing this up. We added an extensive section regarding limitations and misuse to the Discussion on page 13.

---

### Review · Reviewer_fspD · 2025-01-01

**Summary Of Contributions:**

This paper proposes a foundation model-like framework to predict causal graphs, similar to the idea of amortized causal discovery. In summary, the proposed framework Sample, Estimate, Aggregate (SEA) trains a deep learning model using large-scale data.

SEA framework summary:
- FM-inspired goals: Train a model that outperforms individual algorithms, easily change the assumptions to incorporate the knowledge, and strong performance in few-shot learning.
- Training the model and inference on test data are performed similarly. In summary, the input is a whole dataset. *Sample* step samples batches and node subsets from the input dataset. *Estimate* step computes global statistics over all nodes (e.g., inverse covariance), and run a causal discovery algorithm on each batch/node subset to obtain marginal estimates (of the induced subgraph over each subset of nodes generated in the sample step).
- The pretraining the *aggregator* is done in a supervised way: A deep net takes the global statistics, marginal estimates and node subsets to predict the known global causal graph. Aggregator first creates embedding of the inputs, then use axial attention blocks. Finally, all features are passed to a feedforward net to predict the causal graph.
- Axial attention blocks are used for computational efficiency.
- Inference (predicting a causal graph of a given dataset), is performed similar to the training: batches and node subsets are sampled, a causal discovery algorithm (e.g., FCI) is used to create marginal estimates on these subsets, and pretrained aggregator module combines these estimates and global statistics to output a global graph estimate.

The paper presents empirical proofs of concept of SEA that the framework achieves the FM-inspired goals (up to an extent). There is also *some* effort for theoretical demonstration of SEA’s capabilities, in the sense that the proposed model can technically produce the correct global output from marginal estimates.

**Audience:**

Yes

**Claims And Evidence:**

Yes

**Requested Changes:**

**Requests**:
- Section 3.1: Overall Section 3.1 is a bit difficult to parse at this point. For instance, the meaning of the selection scores and the role of the global statistics are not immediately clear. Another example is marginal estimate. This is the first time in the paper ``marginal estimate’’ appears, so an explanation sentence would be really helpful. I know that the details become clear later in the paper (some are even in the appendix), but I think Section 3.1 needs to be more clear to improve the flow of the paper.
- Discussion of (Lorch et al., 2022): While the approaches are different, it seems to me that the study by Lorch et al. 2022 is the most closely related one to this paper. Therefore, I think it warrants a better, more detailed discussion on the similar and different aspects of the present work compared to that paper.
- I respect the decision of authors to limit the main content to 12 pages to make it a regular TMLR submission. That being said, I suggest authors to consider move some parts of the appendix to the main paper. Limitations discussion in Appendix B.7 and at the end of Appendix A.3 are helpful to assess the paper better, and can be given in the main paper. As a minor request, a very brief summary of Appendix A.2 in the main paper would help the reader -- just the main idea of how can marginal estimates be used to form the global graph theoretically, and the takeaways.

**Minor requests:**
- Section 2.1, Page 2: Please define the parent set $\pi_i$ before using it in conditional.
- Section 3.1, Page 4: Perhaps make the faithfulness statement more explicit.

**Minor questions:**
- Section 3.2, Page 4: In the aggregator explanation, an example is given for two variables with low correlation. I think the more interesting case would be to determine whether a high correlation pair is an edge (or just spurious correlation due to a confounder). Can you provide any intuition for how aggregator can handle such cases?
- Section 3.3, Page 5: ``Unique edges’’, can you specify what does unique refer to here? Is it the union of all edges over the marginal estimates?
- Appendix A.4: 2nd point. Why SEA(FCI) pretrained aggregator is more robust, this is a bit unclear to me. Can you provide any intuition behind this observation?

**Strengths And Weaknesses:**

The paper represents a valuable contribution to the active area of using pretraining for causal discovery (in the manner of amortized causal discovery).

The paper’s findings are mostly empirical. Overall, I find the experimental setup and presented results satisfactory.
- The training datasets contain sufficient diversity to assess SEA’s performance in different settings. There is also some effort to evaluate the proposed framework on standard benchmark real data (e.g., Sachs et al. protein signaling dataset)
- Multiple metrics and relevant baselines are presented to evaluate the results in a fair way. For instance, I appreciated using the naïve baseline VarSort to verify that the positive results are not an artifact of the synthetic data generation procedure, and reporting whether the predicted causal graphs obey the theoretical identifiability limits.

As expected, there are some limitations, which the authors discussed in various places in the paper. The theoretical aspect of the paper is weak, though I think the experimental aspect and the model design are convincing enough that I don’t have an issue with it.

---

> ### Author Response · Authors · 2025-02-04
> **Thank you for your review**
>
> Thank you for your detailed review, and for pointing out which areas you find confusing. Our submission has been updated accordingly, and we include a point-by-point response below.
>
> > Section 3.1
>
> We added a brief overview of the methods to the beginning of Section 3, where we define marginal estimates. We also moved some of the details from B.7 to Section 3.1, so we hope that the transition is less abrupt.
>
> > discussion of Lorch et al., 2022
>
> We have added the following comparisons to B.2, which is referenced by the related work.
>
> - Both works simulate diverse datasets for training, which differ in graph topology, causal mechanism, and type of exogenous noise. Beyond differences in mechanisms/noise, the primary difference is scale: their datasets only contain several hundred examples, while ours contain 10-100K.
> - AVICI's input is raw data, and their architecture is invariant to the ordering of samples, equivariant to the ordering of nodes. Our inputs are summary statistics, and we are equivariant to the ordering of nodes.
> - If $M$ is the number of samples and $N$ is the number of nodes, AVICI's complexity scales as $O(M^2 N) + O(N^2 M)$. Our model does not explicitly depend on the number of data samples and scales as $O(N^3 T) + O(N^2 T^2)$, as we operate over richer, pairwise features. For real data, e.g. single-cell transcriptomics, where $M \approx 10^{5-6}$, the former's dependence on $M^2$ would require significant subsampling.
> - AVICI explicitly regularizes for acyclicity. We follow the ENCO formulation (edges $i\to j$ and $j\to i$ cannot co-exist), since it is easier to optimize. We still find that empirically, our predictions are 99% acyclic (Table 16).
>
> > limitations and theoretical intuition in the main text
>
> Thank you for your advice. We added the following content to the main text:
> - limitations in B.7 and B.3 moved to Discussion, which has been significantly expanded with other information as well.
> - brief mention of theoretical intuition (A.2) and results (A.3), added to the start of Section 3.4
>
> The beginning of A.3 now provides extended context for what we are trying to show. We kept the mention of pairwise comparisons in the end of A.3, we feel it is not immediately obvious that a shallow Transformer cannot learn arbitrary functions of, e.g. 3 variables, without context of this related work.
>
> > defining $\pi_i$ and assumptions
>
> Based on Reviewer RoBV's recommendation, we explicitly added the assumption of causal sufficiency to Section 2.1, where we also define $\pi_i$.
>
> In Section 3.1, we have removed the faithfulness statement, as we realize that the ground truth graph is not relevant to or used in the inference procedure. We clarify the relationship between $D$ and $G$ in Section 3.2 instead. We have also reframed the presentation of Section 3.4 to focus on model specification, rather than correctness of results in general.
>
> > how does the aggregator handle spurious high correlation pairs?
>
> We added an illustrative example in C.2 (Figure 7), which depicts the global statistic, GIES bootstrapped, our prediction, and the ground truth.
>
> The highlighted spurious "edge" has a moderate global statistic value, of similar magnitude as several true edges. However, while true edges appear in the majority of GIES estimates (adding up both directions), this edge is missing from a significant portion of estimates, which somewhat reflects the theory in A.2 that edges missing from marginal estimates should be missing from the final skeleton. The aggregator is able to confidently reject this "edge."
>
> > what are unique edges?
>
> We operate over the marginal estimates in a "sparse" fashion, inspired by COO format for sparse matrices / how graphs are represented in some [graph neural network frameworks](https://pytorch-geometric.readthedocs.io/en/stable/modules/data.html).
>
> That is, we separate each edge's features (edge type) from the coordinates they represent, i.e. $(i, j)$.
> The number of "unique" edges is the number of unique pairs $(i,j)$. This has been clarified in the sentence below Eq. 1.
>
> > A.4, why is SEA(FCI) more robust to noise
>
> Figure 8, also in C.2, depicts the same graph as above, except for FCI. We notice that FCI tends to predict much fewer edges in general. This does not necessarily mean poor performance on metrics, as continuous metrics take a sliding window of discretization thresholds, and the prediction is discretized (SHD) with respect to the true number of edges, for "oracles" like FCI.
>
> However, because there are so few "votes" for edges, the signal to noise ratio is lower (when using FCI alone). The highlighted edge here demonstrates that, due to low global statistic, the trained model is able to correctly reject the edge.
>
> (Figures 7 and 8 aren't cherry picked; we simply plotted the first linear test graph and are happy to analyze more cases if that would be of interest.)

---

### Review · Reviewer_RoBV · 2025-01-22

**Summary Of Contributions:**

The authors propose an approach to building foundation models for causal discovery. I'll admit at the outset of this review that I'm a priori quite skeptical of this endeavor: I think in general it is more often than not dangerous to trust these models to arrive at empirical truths, for various reasons. But the approach outlined is interesting, and perhaps could serve as the basis for further discussion. I think the idea of learning subgraphs reliably along with global statistics and stitching them together is a good one for this approach.

(I should also note that my expertise is in causal discovery, graphical models, and statistics, but I have no expertise in foundation models and minimal understanding of recent deep learning advancements, so my review will mainly stick to what I know.)

**Audience:**

Yes

**Broader Impact Concerns:**

None.

**Claims And Evidence:**

No

**Requested Changes:**

The revision should address all the points raised above.

**Strengths And Weaknesses:**

There are several places where I find issues with the proposed method or surrounding discussion:

I'm going to make my comments in the order they appear rather than in order of importance. I'll mark what I take to be the most important comments with an asterisk *.

*Abstract and later: in several places, the authors repeat some version of the claim that errors of causal discovery algorithms "remain comparable across datasets." (This is said in the abstract, the introduction, Sec 3.4, and elsewhere.) I'm not sure what this means precisely or what reason there could be to believe this. It seems the suggestion is that different algorithms, which operate under different assumptions, use different statistical principles, and consider different classes of outputs (DAGs vs CPDAGs vs PAGs vs other), will suffer "similar" errors across different datasets generated from different kinds of ground truth... I don't see how this could possibly be true and certainly is not true in my experience with causal discovery algorithms. Different algorithms can behave quite differently on the same data in practice, and the same algorithm across different data sets will often have very different performance. Maybe I misunderstand what the authors intend to convey with these sentences, but if so it should be clarified -- and if it is actually essential then the claim should be backed up by evidence.

Pg. 1 "Discrete optimization algorithms": this is a potentially misleading way to describe the class of causal discovery methods discussed in the cited work, since many of these are not "optimization" algorithms in an obvious sense -- while score-based algs do try to find "optimal" scoring graphs, constraint-based algs and LiNGaM-type algs do not optimize anything, but rather use testing or other model selection ideas. I would suggest simply calling these "discrete search algorithms."

Pg. 1 "Driven by hypothesis tests": only some of these are based on hypothesis tests, others do not use hypothesis tests at all.

Pg. 1 "Necessarily impose functional assumptions": not true, there are nonparametric tests, though these have challenges e.g. no test has power for all alternatives.

Pg. 1 "the former must be trained from scratch per-dataset": this is a strange way to (negatively) summarize these continuous optimization-based algs. There's always *something* happening "per-dataset" (or else you're not using the data), it is only negative if that doesn't work well. Maybe the correct challenge to highlight for the cited methods is that they do not seem robust, they don't perform well outside of simple settings according to some recent papers.

Pg. 2 "which determine whether G is identifiable": should be "uniquely identifiable or identifiable up to an equivalence class."

Pg. 2 "allow us to perform interventions": this should be "can be used, along with other information, to compute the downstream consequences of interventions." Computing the post-intervention distribution requires more than just the graph structure.

*Pg. 2 "the goal of causal structure learning is to recover G": there is no mention of latent variables or unmeasured confounding in this set up. The implicit assumption here is that the truth is a DAG over variables only in X, a.k.a. causal sufficiency. Even if the setting with unmeasured confounders is not of interest in this paper, which is fine, this should be made explicit here.

Pg. 2 "Discrete optimization methods make atomic changes to a proposal graph until a stopping criterion is met." Again, this is misleading description of many algorithms, see above. It is also misleading to describe LiNGaM as score-based. These types of algorithms have been previously described as "semi-parametric" procedures since they exploit asymmetries implied by the semi-parametric model classes that are assumed.

Pg. 3 "amortized inference approaches": missing a reference to Petersen et al. (2023) which should be cited. (This actually came out on arxiv earlier than or around the same time as the cited work.)

Petersen, A. H., Ramsey, J., Ekstrøm, C. T., & Spirtes, P. (2023). Causal Discovery for Observational Sciences Using Supervised Machine Learning. Journal of Data Science, 21(2), 255-280.

*Pg. 4 some of the description in Sec 3.1 is unclear. What are the "selection scores \alpha? If they are correlation or precision matrices then they are not in {0,1}^{N x N}, they are real numbers. Are these actually probabilities? Why are they only over D_0? The \sum_{j \in S_t} \alpha_{i,j} is not clear, what's going on with the index i? S_t are subsets of nodes. Why the index t? Overall this description could be cleaned up.

Pg. 4 "Each input is a whole dataset": make clear here that you are generating synthetic data with known ground truth based on some parametric model. This whole section leaves it somewhat mysterious what the training data is.

*Pg. 4 use of FCI and GIES: I'm confused about these choices. FCI is for the setting with unmeasured confounding and learns a PAG, not a DAG or CPDAG. GIES is for the setting with no confounding but with interventional data. Why are you using FCI (instead of say PC), and how are you translating between the PAG output and the directed graph DG output (esp when computing statistics such as structural hamming distances, etc)? It's not straightforward, because the adjacencies in a PAG and a CPDAG may not be the same even with no unmeasured confounding. You mention using both observational data and interventional data, but how do you do use interventional data with FCI? When there is no interventional data do you use GES instead of GIES? It seems like the inference step is ultimately predicting a (possibly cyclic) directed graph, so how are you using information like "X is an ancestor of Y"? You write: "as long as their outputs are the same format" -- but they are not the same format? This part is unclear to me.

*Pg. 6 "graph prediction": it seems you are learning a (possibly cyclic) directed graph, not a DAG or CPDAG. If you were learning a DAG, at least you could transform it to the corresponding CPDAG using existing results on Markov equivalence. In your evaluations, you seem to computer performance statistics (SHD etc) wrt the true DAG (since the truth is always a DAG) not the true CPDAG, is that correct? This is misleading, since you could estimate a DG where the true CPDAG (best you can do, theoretically) has a *higher* SHD to the true DAG than the estimated DG. But it isn't because you did "better" than getting the true CPDAG, but because you got randomly lucky. These issues ought to be addressed.

(For example: say the truth is A-->B-->C. The true CPDAG is A--B--C which has an SHD of 2 from the true DAG. Perhaps you could learn this with PC or GES. But with the proposed algorithm you learn, out of luck, the DG A-->B-->C-->A, which has a SHD of 1 from the truth. You just got "lucky," the best you can do nonparametrically is the CPDAG A--B--C. To say you've done "better" than PC/GES in this case would be misleading, PC/GES was perfectly correct.)

*Theorem 3.1. I must admit I don't understand Theorem 3.1 as a formal statement, and I don't follow the proof. It is not that I doubt that the claim is roughly correct, but am not sure I understand what the claim is because of language that is to me (as a statistician) too informal. "Has the capacity to recover" is not precise to me. I'm used to claims about identifiability, what a procedure will do as sample size approaches infinity, or what will happen with high probability for some finite sample size, but this is none of those. Again, I don't really doubt that the reasoning is sound. I hope another reviewer could evaluate this claim more carefully. [Note: Appendices A.1 and A.2 seem ok to me, it is the argument in A.3 that I don't follow.]

Pg. 7 "failure modes of causal discovery algorithms may be similar across datasets and can be corrected using statistics that capture richer information": Again, it is not clear to me what this means and whether it is true (see above).

Pg. 7 "an algorithm that assumes linearity will make (predictably) asymmetric mistakes on non-linear data and underestimate the presence of edges" -- this depends on the algorithm. It is true that constraint-based methods that use linear association tests may miss edges that correspond to nonlinear dependencies. But how does LiNGaM or NOTEARS behave in the presence of nonlinearities? Less obvious.

*Pg. 7 "When all assumptions are upheld, and infinite data are available, the model has the capacity to infer a sound graph": this statement is unclear (what are the assumptions, and what is a "sound graph"?) and there is no real evidence for this claim presented specifically. Theorem 3.1 is does not refer to DGP assumptions or infinite data at all.

Pg. 7 "In practice, the model will output an orientation for all edges, but the graph can be interpreted as one member of an equivalence class": as discussed above, if the graph was acyclic then you could and probably should transform the result to a CPDAG to compare to algs that return a CPDAG. As it stands the procedure can return a cyclic DG for which equivalence theory is more complicated (but anyway the training data all comes from DAGs).

Pg. 7 "both observational and interventional datasets": how were the interventional data sets generated, what were the interventions and the ground truth here? How was this info used, if at all, with the purely observational discovery algs?

*Pg. 8 "VarSort, Fci, and Gies were run using non-parametric bootstrapping (Friedman et al., 1999), with 100 subsets of 500 examples each": how were the bootstraps aggregated into one final structure? Different ways of combining bootstrap with causal discovery algorithms can lead to very different results. Also I'm a little unclear about the sample sizes here. What was the actual sample size for each of the "inference" algorithms, n=500? (The training sample is described for the foundation model, but not the testing sample sizes for Table 1. Also, what \alpha threshold is used with FCI? This obviously affects the sparsity of the discovered graph, and in real applications this hyperparameter needs to be selected somehow.

*Sec 5.3: I find the discussion of "respecting identifiability theory" somewhat unclear. It seems to say that when the truth is only identifiable up to Markov equivalence, the proposed procedure recovers a graph that is similar in accuracy to a random graph from the MEC. Why not evaluate whether the procedure gets close to the true MEC/CPDAG instead? It is misleading to evaluate everything wrt the true DAG when that is not identifiable. Also: how is FCI being evaluated here since it returns a PAG? See above.

References: some citations are missing a lot of information, needs editing.

*Meta-point: There are a lot of important questions about using these kinds of tools in practice that are not addressed here. In my opinion, foundation models are only as good as the data they are trained on, and as it stands, this are trained on synthetic data which barely resembles real data. All one variable type, simple (even if non-linear) functional relationships, no missingness, no measurement error, etc. What assurances can there be for use on real data? How does one know the "failure modes" of this kind of procedure, which elements to trust? (I recognize these questions are also tough for classic causal discovery algorithms, but there I think we have at least imperfect answers. We know roughly when hypothesis tests perform poorly, when score-based algs get stuck in local minima, etc.)

---

> ### Author Response · Authors · 2025-02-04
> **Thank you for your review (1/3)**
>
> Thank you for your detailed review. We very much appreciate your feedback towards improving our paper. Our submission has been updated accordingly, and we include a point-by-point response below (smaller points roughly grouped with * where relevant).
>
> > * "similar" errors across different datasets generated from different kinds of ground truth
>
> We would like to clarify that these statements refer to the primary setting in this paper, where we fix a single causal discovery algorithm, and the types of errors are compared across datasets.
>
> - For constraint-based algorithms with linear tests, this may mean missing nonlinear associations.
> - For LiNGAM, this may mean making errors on variables with Gaussian-like exogenous noises.
> - We do not envision using complex algorithms like NOTEARS/DCDI within the estimation subroutine, as they require significant optimization, even on smaller graphs.
>
> We modified these statements throughout the paper, to emphasize that we primarily compare across datasets, not algorithms. We also noted in the Discussion that the cross-algorithm setting is a limitation/extension. To clarify the context of our experiments which involve other algorithms:
>
> - In Table 4 (LiNGAM on non-Gaussian), our primary goal was to show that incorporating prior knowledge can "recover" performance in genuinely out-of-distribution settings. The "mistake" of non-Gaussianity is out of distribution since we (could but) did not train on it. If we are aware that the data are actually non-Gaussian, which can be tested for, then LiNGAM would not make this mistake, and the marginal estimates would be more reliable. The practice of holding out causal mechanisms/noise types to evaluate model generalization is not uncommon for supervised causal discovery works, including AVICI (Lorch et al., 2022).
> - Table 7 (swapping algorithms) is an empirical demonstration, and we add a note for clarification. While not a primary focus of this work, it does appear that there is some commonality among the errors of different algorithms that assume linear Gaussian, as the change in performance is small.
>
> > more precise language in the exposition
>
> Thank you so much for helping us refine our presentation of concepts in causality and causal discovery. We have updated the relevant sections to reflect your recommendations, including adding Petersen et al.
>
> > * causal sufficiency
>
> We have added this explicitly to the start of Section 2.1.
>
> > * selection scores $\alpha$
>
> We added a brief overview of the methods to the beginning of Section 3, and we moved some of the details from B.7 to Section 3.1. Specifically:
>
> - We updated the language to reflect that precision matrices may be real values, but nodes are sampled with "probability proportional to [...]"
> - We computed $\alpha$ over $D_0$ to introduce stochasticity during training. During inference, stochasticity is not necessary (updated text to include $D$). Empirically, our ablations (Figure 4) indicate that performance plateaus after several hundred examples, so the difference may be minimal.
> - Subsets may be constructed to prioritize nodes that are likely to be connected to existing ones, hence the sum over $S_t$ (which refers to the subset currently under construction). We added details regarding how each set is initialized / that we downweight selected nodes in subsequent rounds of sampling.
>
> > * FCI and GIES
>
> To reiterate from above, the primary setting is one discovery algorithm, many datasets. FCI and GIES are indeed different in many aspects, which is why we chose to train two different models, one with FCI and one with GIES. Though our datasets were not generated with unobserved confounders in mind, the act of sampling nodes may induce "unobserved" confounders with respect to the subset, which is why we chose FCI over PC.
>
> The model (aggregator)'s input and output spaces may be different. The input space corresponds to the type of graph produced by the classic discovery algorithm (e.g. PAG v.s. CPDAG). Regardless of the input space, the model outputs only three edge types: "causes," "is caused by," "no edge."
>
> **Representing different graph types:** The PAG produced by FCI is input to the model with all its edge types ("learned embedding $\text{ebd}_\mathcal{E}$" above Eq. 1).
>
> For example, suppose we know "A causes B; no set d-separates A and C; and there is a common latent cause of C and D." We set "causes = 1, is caused by = 2, not d-separated = 3, and common latent cause = 4."
>
> |   | A | B | C | D |
> |---|---|---|---|---|
> | A | 0 | **1** | **3** | 0 |
> | B | **2** | 0 | 0 | 0 |
> | C | **3** | 0 | 0 | **4** |
> | D | 0 | 0 | **4** | 0 |
>
> To encode this matrix, we use a learned embedding, i.e. a look-up table where $d$-dimensional representations of each edge type are updated over the course of model training – similar to how words are represented to language models. This example would produce an embedding of size $4\times 4\times d$.

---

> > ### Author Response · Authors · 2025-02-04
> > **(2/3)**
> >
> > **observational vs interventional data:** Each graph is associated with two datasets, one observational and one interventional (observational + perfect interventions on each node). Models that take interventional data were provided full knowledge of the intervention target identities. We updated Section 4.1 to clarify, with details in B.1 and brief mention in Section 2.1.
> >
> > We only run FCI on observational datasets, and GIES on interventional datasets. We updated the caption of Table 1 to note which models were run in which setting. We do try the GIES model with GES in Table 7, which only reports on the observational setting.
> >
> > **Swapping algorithms:** We added a footnote for clarification.
> >
> > A model trained to interpret GIES outputs *cannot* be used to interpret a PAG (there are no embeddings for "not an ancestor"). On the other hand, a model trained to interpret GIES graphs *could* be used to interpret PC or GES estimates. However, since GIES and PC may make different types of mistakes, correct interpretations by the model, i.e. empirical performance, cannot be guaranteed. Table 7 suggests that the difference is minimal among alternatives with similar assumptions, while (new) Table 23 shows that swapping to LiNGAM leads to a larger gap.
> >
> > > * Model output and evaluation with respect to DAG / CPDAG / MEC + identifiability section
> >
> > During model development, we did consider that evaluating with respect to the CPDAG / essential graph may be more correct in the observational case, if a DAG can be predicted. There were several reasons we ultimately chose to evaluate with respect to the ground truth DAG, rather than the equivalence class (added to new section B.3).
> >
> > - Many continuous causal discovery baselines evaluate with respect to the single ground truth DAG, whether they consider interventional data or not (DCDI, AVICI, BaCaDI, DiffAN), perhaps because metrics that reflect uncertainty (mAP, AUROC) are difficult to translate to the equivalent CPDAG.
> >
> > - It has been reported that common simulators produce "identifiable" linear Gaussian datasets, if the data are unstandardized (hence the VarSort baseline). This would mean that the MEC is actually smaller than expected. While it is possible to normalize all the data, it is unknown whether there are additional artifacts of the data generating process that may influence the (empirical) identifiability of the graphs, and thus, the "true" size of the equivalence class.
> >
> > - Our models and baselines are evaluated on the same graphs, but not all can incorporate interventional information. For comparability across methods, we evaluated all predictions against the same ground truth, while noting that the equivalence class is larger in the observational case.
> >
> > - Empirically, most of our predictions were acyclic (Table 16, ~99%). However, not all baselines are. This includes the nonparametric bootstrapping baselines (FCI, GIES, VarSort) due to bootstrapping, as well as the symmetric summary statistics (Appendix C) that quantify the "cost" of not directing edges / of not using marginal estimates.
> >
> > - Practically, it was somewhat expensive to compute the CPDAG, especially for larger dense graphs ($N=100, E=400$ and beyond).
> >
> > > * Theorem 3.1 and A.3
> >
> > We have added a section to the beginning of A.3 to discuss: what is a "well-specified" model, in classical statistics and modern neural networks; what continuous causal discovery algorithms must consider in terms of identifiability (optimization objective; convergence; and model specification); and what Theorem 3.1 addresses (model specification).
> >
> > We survey the related work that motivates the rest of A.3. Please let us know if this provides context regarding what the proof intends to show, or if you have any remaining questions.
> >
> > > * Capacity to infer a sound graph
> >
> > More precisely, we meant that: given correct marginal estimates, the axial attention model is well-specified, i.e. there exists a setting of its parameters that can derive the correct graph from correct marginal estimates. We have modified the language regarding infinite data / assumptions, which was intended to convey the "correctness" of the marginal estimates, rather than provide guarantees on what *will* be learned by our model.

---

> > > ### Author Response · Authors · 2025-02-04
> > > **(3/3)**
> > >
> > > > * Non-parametric bootstrapping: We have updated the relevant sentences to reflect the details below.
> > >
> > > We viewed FCI and GIES as "oracles" for what can be inferred, if these estimation algorithms were run on all the variables, without subsampling nodes. We aimed to *maximize* the performance of these baselines (and VarSort), and our bootstrapping strategy was selected based on best *test* performance. This is in contrast to the remaining baselines, which are fairly evaluated on the test sets without any prior knowledge.
> > >
> > > Specifically, we ran each algorithm 100 times on all $N$ variables, with 1000 samples each time (correction). This yields 100 predicted graphs. The final prediction for each edge was its frequency of appearing as directed.
> > >
> > > Alternatives we tried (which did not perform as well) included:
> > > - Running the algorithm once, on all $1000N$ samples – FCI tends to miss edges very frequently (Figure 8)
> > > - For FCI, interpreting "B is not an ancestor of A" as "A -> B"
> > > - Instead of frequency, binarize to "does an edge appear" – GIES tends to predict very dense graphs with this aggregation scheme (Figure 7)
> > > - Counting undirected edges in the frequency computation, i.e. "A - B" counts for both "A -> B" and "B -> A"
> > >
> > > FCI used the same threshold as the marginal estimates, as the primary goal was to quantify the information content of the marginal estimates.
> > >
> > > > * practical use of these tools and failure modes
> > >
> > > We have substantially expanded the Discussion to discuss limitations and considerations about the data generation process. This work focuses primarily on the modeling framework, and we do not solve how one might faithfully simulate biology, or other real world domains. We agree that these are very important open problems, and note that any real application should start with thoroughly understanding the nature of the data and its imperfections.
> > >
> > > > references
> > >
> > > We have filled in the missing information for the references, and in the few cases where only preprints are available, ensured that an identifier is listed. Thank you for pointing this out.

---

> > > ### Comment · Reviewer_RoBV · 2025-02-21
> > > **Re: comparing wrt DAG or MEC**
> > >
> > > Thank you to the authors for their responses and changes to the paper. One point which I want to register some disagreement is on is the discussion of comparison wrt the true DAG vs MEC.
> > >
> > > When the truth is a DAG, without any additional strong assumptions (e.g., equal variances etc.) only the CPDAG is identifiable. Some model selection methods may output a single DAG anyway, but to separate the actual performance of the proposed algorithm from artifacts of the data-generating process, comparisons should be made at the level of CPDAGs (or interventional MECs in the case of interventional data). For example, say I am comparing my new proposed method to PC or GES or similar. Let's say PC/GES does as best as it theoretically could and outputs the true CPDAG (of the true DAG). My new proposed method outputs a DAG G that looks "close" to the true DAG, perhaps even closer to the true DAG than the true CPDAG output by PC/GES according to some distance metric, but if I turn G into its corresponding MEC(G) it is the wrong CPDAG. Did the proposed method do "better" than PC/GES here? No, by assumption PC/GES did as best as theoretically possible -- the discrepancy must be attributed to some artifact of the data-generating process: my new proposed algorithm that returned G just got "lucky" somehow. If instead G was a member of the true CPDAG, MEC(G) and the output of PC/GES here would be the same, and then I would say they've done equally well. For this reason I remain unconvinced that evaluating wrt the true DAG is generally a good idea, despite the fact that some baseline methods have been evaluated this way.
> > >
> > > It is a little unclear to me what  you are doing when the output of the procedure is MEC, if you're using the true DAG as the target of comparison: are you computing the performance metric between the MEC and true DAG, or the "best DAG in the MEC" compared to the true DAG?
> > >
> > > Re: varsortability and related artifacts -- I agree that there may be other artifacts of a simulation pipeline that have empirical identifiability consequences. But at the very least, the data could be standardized throughout to prevent this particular known artifact. Or better yet, the data-generating trick described in Squires et al. (2022, Sec 5.1) could be implemented; this controls the variance contributed to each node by its parents versus exogenous noise and seems to adequately address the known varsortability issue.
> > >
> > > Squires, Yun, Nichani, Agrawal, Uhler (2022) "Causal Structure Discovery between Clusters of Nodes Induced by Latent Factors" in CLeaR.

---

> > > > ### Author Response · Authors · 2025-02-24
> > > > **Thank you for your response!**
> > > >
> > > > Thank you for explaining further, and your examples make a lot of sense. We've made additional changes in magenta, with a summary below. Please let us know if you have any additional questions or concerns!
> > > >
> > > > - We updated the identifiability discussion in 3.4 and the metrics overview in 4.2.
> > > > - We added Table 14 with SHD with respect to the CPDAG, for the observational models.
> > > > - We added Table 15 with mAP with respect to the undirected skeleton, for all models.
> > > > - Table 1 clarifies that its results compare to the true DAG, with references to 14 and 15 for the CPDAG/skeleton.
> > > >
> > > > In the observational setting (DiffAN, FCI, VarSort, SEA - FCI), we convert the predicted DAG into the equivalent CPDAG, over which we evaluate the SHD (as implemented in [the CDT package](https://fentechsolutions.github.io/CausalDiscoveryToolbox/html/_modules/cdt/metrics.html)). We find that the SHD varies slightly when evaluating w.r.t. the DAG / converted CPDAG, but the difference is minimal, compared to the overall gaps in performance between methods.
> > > >
> > > > For uncertainty quantification (continuous discovery algorithms), we report the mAP of the predicted undirected skeleton ($P(i - j) = P(i \to j) + P(j \leftarrow i)$), compared to the true undirected skeleton. Here, our method still performs the best, and GIES is very competitive. This may suggest that other continuous causal discovery models with access to intervention targets are not leveraging this information to the maximum potential.
> > > >
> > > > When the output is an MEC: if you have any recommendations with regard to best practices, we are happy to implement them. Currently, the evaluation looks like this.
> > > >
> > > > - In the new Table 14, we evaluate at the level of CPDAGs.
> > > >
> > > > - In Table 1, we compare between the MEC and true DAG (option 1).
> > > >
> > > > - In Table 6, row 1 is the same as above. For row 2, we compare *each DAG* in the MEC to the ground truth DAG and compute the expected value of each metric (mean, instead of min/max), i.e. if we sample any DAG uniformly at random from the MEC.

---

### Author Response · Authors · 2025-02-04
**Thank you for your reviews!**

Dear Reviewers,

Thank you very much for taking the time to read and provide feedback on our work. Your comments are very helpful for improving the clarity and presentation of the paper.

Our submission has been updated to address your reviews, with changes in blue. In particular, we have prepared a much-expanded Discussion section, which includes limitations and future directions.

We hope that these updates and individual responses adequately address your concerns. Please let us know if there is anything else you find confusing.

Best regards,
Authors

---

### Decision · Action_Editor_bWNF · 2025-03-19

**Recommendation:** Accept as is

**Comment:**

Some concerns about the experimental setup were raised by reviewers which were satisfactorily addressed by the authors. Various clarity and completeness concerns were also addressed in the revision. After the revision, all reviewers agreed to accept the paper as-is.

**Audience:**

The topic is clearly relevant to TMLR's audience.

**Claims And Evidence:**

The authors propose SEA (sample estimate aggregate), a new foundation model-inspired approach for amortized causal discovery. This is a supervised model trained on large amounts of synthetically generated data to predict causal graphs from summary statistics. This approach to causal discovery has grown in recent years, and it appears the authors provide a useful contribution to this developing field.